# Single hidden layer diffusion models provably learn simple low-dimensional structure

**Nicholas M. Boffi**
Carnegie Mellon University
nboffi@andrew.cmu.edu

**Arthur Jacot**
New York University
arthur.jacot@nyu.edu

**Stephen Tu**
University of Southern California
stephen.tu@usc.edu

**Ingvar Ziemann**
University of Pennsylvania
ingvarz@seas.upenn.edu

## Abstract

Diffusion-based generative models provide a powerful framework for learning to sample from a complex target distribution. The remarkable empirical success of these models applied to high-dimensional signals, including images and video, stands in stark contrast to classical results highlighting the curse of dimensionality for distribution recovery. In this work, we take a step towards understanding this gap through a careful analysis of learning diffusion models over the Barron space of single hidden layer neural networks. In particular, we show that these shallow models provably adapt to simple forms of low-dimensional structure, such as an unknown linear subspace or hidden independence, thereby avoiding the curse of dimensionality. We combine our results with recent analyses of sampling with diffusions to provide an end-to-end sample complexity bound for learning to sample from structured distributions. Importantly, our results do not require specialized architectures tailored to particular latent structures, and instead rely on the low-index structure of the Barron space to adapt to the underlying distribution.

## 1 Introduction

Generative models learn to sample from a target probability distribution given a dataset of examples. Applications are pervasive, and include language modeling (Li et al., 2022), high-fidelity image generation (Rombach et al., 2022), *de-novo* drug design (Watson et al., 2023), and molecular dynamics (Arts et al., 2023). Recent years have witnessed extremely rapid advancements in the field of generative modeling, particularly with the development of models based on dynamical transport of measure (Santambrogio, 2015), such as diffusion-based generative models (Ho et al., 2020; Song et al., 2021), stochastic interpolants (Albergo et al., 2023), flow matching (Lipman et al., 2023), and rectified flow (Liu et al., 2023) approaches. Yet, despite their strong empirical performance and well-grounded mathematical formulation, a theoretical understanding of *how* and *why* these large-scale generative models work is still in its infancy.

A promising line of recent research has shown that the problem of sampling from an arbitrarily complex distribution can be reduced to unsupervised learning: for diffusion models, if an accurate velocity or score field can be estimated from data, then high-quality samples can be generated via numerical simulation (Chen et al., 2023d; Lee et al., 2023). While deeply insightful, these works leave open the difficulty of statistical estimation, and therefore raise the possibility that the sampling problem's true difficulty is hidden in the complexity of learning the score.

In this work, we address this fundamental challenge by presenting an end-to-end analysis of sampling with score-based diffusion models. To balance tractability of the analysis with empirical relevance, we study the Barron space of single hidden layer neural networks (E et al., 2019; Bach, 2017). This space contains important features of models used in practice – most importantly, parametric nonlinearity – while retaining well-studied theoretical properties that we can adapt to the generative modeling problem. As paradigmatic examples of the widely-held belief that real-world datasets contain hidden low-dimensional structure (Tenenbaum et al., 2000; Weinberger & Saul, 2006), we focus on two

idealized settings in which the target data density is concentrated on an unknown low-dimensional linear manifold, or consists of groups of independent low-dimensional variables. We show that for learning to sample from a target distribution with either of these structures, diffusion models backed by Barron networks enjoy a sample complexity bound that only depends exponentially on the *intrinsic* dimension of the structure rather than on the ambient dimension. Our results highlight that diffusion models based on one hidden layer networks without specific architectural modifications can adapt to hidden structure and sidestep the curse of dimensionality; in this way, they give insight into the empirical performance of more complex architectures on real-world high-dimensional datasets.

## 2 RELATED WORK

**Sampling bounds for diffusion models.** Much of the recent work on analyzing diffusion models has focused on the accuracy of sampling from a discretized backwards diffusion process, assuming access to $L_2$ accurate score functions. This includes work on discretized stochastic differential equations (Lee et al., 2022; Chen et al., 2023d; Lee et al., 2023; Chen et al., 2023a; Benton et al., 2024) and more recently, discretized probability flows (Chen et al., 2023e;c; Li et al., 2024; Liang et al., 2024; Gao & Zhu, 2024). Recent work by Li & Yan (2024) shows that the DDPM sampler (Ho et al., 2020) can be tuned so that in the presence of low dimensional structure, the discretization error only depends polynomially on the intrinsic dimension (in addition to the score error). However, these works assume the existence of an $\varepsilon$-accurate score function in $L_2$, and thereby leave open the question of the sample complexity of learning such a model.

**Sample complexity of score matching.** Several works have considered the statistical complexity of learning a score function. Block et al. (2020) and Koehler et al. (2023) use the standard Rademacher complexity framework to bound the error of empirical risk minimization (ERM) for learning the score, but leave open the question of which function class to learn over. Han et al. (2024) and Wang et al. (2024) consider optimizing the denoising score matching loss over neural network models, and show that gradient descent on overparameterized models finds quality ERM solutions. Wibisono et al. (2024), Zhang et al. (2024), Oko et al. (2023), and Dou et al. (2024) study the minimax optimality of diffusion modeling under various functional assumptions regarding the target density and its corresponding score function. While together these works show that diffusion modeling is both nearly optimal for learning to sample and also computationally efficient, they simultaneously highlight the curse of dimensionality that arises without more structure. To address this issue, both Oko et al. (2023) and Chen et al. (2023b) impose a latent subspace assumption on the data distribution, which they show avoids exponential dependence on the ambient dimension. Our work is directly inspired by these results, and we defer a detailed comparison to Section 3. Another related work is Bortoli (2022), which studies diffusion models under a manifold hypothesis, proving a bound that depends exponentially on the diameter of the manifold. Finally, Cole & Lu (2024) show that under the assumption that the target log-relative density (w.r.t. a standard Gaussian) can be approximated by a network with low path norm, score estimation can be performed with a sample complexity bound that does not depend explicitly on the ambient dimension.[1] However their absolute continuity assumption rules out examples such as target distributions supported on low dimensional manifolds.

**Learning in Barron spaces.** Even though the implicit bias of deep networks remains largely an open question, there is now strong consensus that the implicit bias of shallow networks with large width is accurately captured by the Barron norm (E et al., 2019) or variational $\mathcal{F}_1$ norm (Bach, 2017). Such networks can avoid the curse of dimensionality by capturing *low-index* structure (i.e., when the target $f(x) = f(Px)$ for a low-dimensional projection $P \in \mathbb{R}^{d \times D}$), leading to generalization bounds that depend on the intrinsic dimension $d$ rather than the ambient dimension $D$ (Bach, 2017). Several recent results have also studied the (sometimes modified) gradient descent dynamics of shallow networks, and how this low-index structure emerges in the network (Abbe et al., 2022; Bietti et al., 2022; Ben Arous et al., 2022; Glasgow, 2024; Lee et al., 2024). While most of this literature focuses on supervised training problems, some work has shown that this type of analysis can be extended to the unsupervised case, in particular to learn energy-based models (Domingo-Enrich et al., 2021).

---

[1] However, there are still $O(1)^D$ pre-factors in the final rate.

## 3 PROBLEM FORMULATION AND MAIN RESULTS

Our goal in this work is to study the statistical complexity of learning to sample from a target probability measure $p_0(x_0)$ defined on $\mathbb{R}^D$ given a dataset of $n$ iid examples $x_0^i \sim p_0$ for $i = 1, \ldots, n$. In particular, we consider the use of a diffusion model (Sohl-Dickstein et al., 2015; Song et al., 2021; Ho et al., 2020) to learn a stochastic process that maps random noise to a new sample from the data distribution. We assume that the target $p_0$ contains hidden latent structure – either a low-dimensional subspace or independent components – and our primary goal will be to show that a single hidden layer network can learn this hidden structure efficiently, in the sense that the statistical rates are governed primarily by the underlying latent dimension $d$, as opposed to the ambient dimension $D \gg d$.

**Diffusion models.** We consider diffusion-based generative models based on stochastic differential equations (Song et al., 2021). These models construct a path in the space of measures between the target $p_0$ and a standard Gaussian $\mathsf{N}(0, I_D)$ by defining a *forward process* that converges to Gaussian data over an infinite horizon. For simplicity, we consider the Ornstein-Uhlenbeck (OU) process,

$$\mathrm{d}x_t = -x_t \mathrm{d}t + \sqrt{2}\mathrm{d}B_t, \quad t \in [0, T], \tag{3.1}$$

though our results straightforwardly generalize to the time-scaled OU processes commonly used in practice (Song et al., 2021). In (3.1), $(B_t)_{t \geq 0}$ denotes a standard Brownian motion on $\mathbb{R}^D$. Due to the linear nature of the OU process, for $w \sim \mathsf{N}(0, I_D)$ drawn independently of $x_0$, $x_t$ is equivalent in distribution to the stochastic interpolant (Albergo et al., 2023)

$$x_t \overset{\mathsf{d}}{=} m_t x_0 + \sigma_t w, \quad m_t := \exp(-t), \quad \sigma_t := \sqrt{1 - \exp(-2t)}. \tag{3.2}$$

Let the marginal distributions of $x_t$ be denoted as $(p_t)_{t \in [0, T]}$. The *reverse process* is the process of $y_t := x_{T-t}$ for $t \in [0, T]$. A classic result (Anderson, 1982) shows that the reverse process satisfies

$$\mathrm{d}y_t = (y_t + 2\nabla \log p_{T-t}(y_t))\mathrm{d}t + \sqrt{2}\mathrm{d}B_t, \quad y_0 \sim p_T(\cdot), \quad t \in [0, T]. \tag{3.3}$$

Thus, assuming knowledge of the time-dependent score function $\nabla \log p_t$, sampling from $p_0(\cdot)$ can be accomplished by (a) setting $T$ large enough so that $p_T(\cdot)$ is approximately an isotropic Gaussian, (b) sampling $y_0 \sim \mathsf{N}(0, I_D)$, and (c) running the reverse SDE (3.3) until time $T$.

To implement this scheme in practice, the score function must be learned, and the reverse process must be discretized. Assuming access to a learned score function $\hat{s} \approx \nabla \log p$, we now consider discretizing (3.3). In this work we make use of the *exponential integrator* (EI), which fixes a sequence (to be specified) of reverse process timesteps $0 = \tau_0 < \tau_1 < \cdots < \tau_N = T$ and implements

$$\mathrm{d}\tilde{y}_t = (\tilde{y}_t + 2\hat{s}_{T-\tau_k}(\tilde{y}_{\tau_k}))\mathrm{d}t + \sqrt{2}\mathrm{d}B_t, \quad t \in [\tau_k, \tau_{k+1}], \quad k \in \{0, \ldots, N-1\}. \tag{3.4}$$

Recently, building off of the works by Chen et al. (2023d) and Lee et al. (2023), Benton et al. (2024) showed that it suffices to control the score approximation error in $L_2(p_t)$ to guarantee that the process (3.4) yields a high quality sample from $p_0$.[2]

**Score function estimation.** To estimate the score function $\nabla \log p_t$ over the interval $[0, T]$, one would ideally minimize the least-squares objective over a model $\hat{s}$,

$$\mathcal{R}(\hat{s}) := \int_0^T \mathcal{R}_t(\hat{s}_t)\,\mathrm{d}t, \qquad \mathcal{R}_t(\hat{s}_t) := \mathbb{E}_{x_t}\|\hat{s}_t(x_t) - \nabla \log p_t(x_t)\|^2. \tag{3.5}$$

While direct minimization is not possible because $\nabla \log p_t$ is not observed, minimizing $\mathcal{R}(s)$ is equivalent to minimizing the following denoising score matching (DSM) loss (Vincent, 2011)

$$\mathcal{L}(\hat{s}) := \int_0^T \mathcal{L}_t(\hat{s}_t)\,\mathrm{d}t, \qquad \mathcal{L}_t(\hat{s}_t) := \mathbb{E}_{(w, x_t)}\|\hat{s}_t(x_t) + w/\sigma_t\|^2, \tag{3.6}$$

as can be shown by Tweedie's identity $\nabla \log p_t(x) = -\frac{1}{\sigma_t}\mathbb{E}[w \mid x_t = x]$ (Efron, 2011). In practice, (3.6) is typically approximated via Monte-Carlo by generating samples $x_{t^i}^i = m_{t^i} x_0^i + \sigma_{t^i} w^i$ using the dataset of samples from $p_0$, iid random draws of Gaussian noise $w^i$, and random time points

---

[2]Technically, Benton et al. (2024) guarantees a high quality sample from $p_{T-\tau_{N-1}}(\cdot)$ instead of $p_0(\cdot)$.

$t^i$ drawn from $[0, T]$. This empirical risk can then be minimized to estimate a time-dependent score function $\hat{s} : [0, T] \times \mathbb{R}^D \to \mathbb{R}^D$.

In this work, to simplify the mathematical analysis, we consider a stylized variant in which we fix a sequence of timesteps $0 < t_0 < \cdots < t_{N-1} = T$ and estimate $N$ time-independent score functions $\{\hat{s}_{t_i}\}_{i=0}^{N-1}$ of the form $\hat{s}_{t_i} : \mathbb{R}^D \to \mathbb{R}^D$ from a family of function classes $\mathscr{F}_t$:

$$\hat{s}_t \in \arg\min_{s_t \in \mathscr{F}_t} \hat{\mathcal{L}}_t(s_t), \quad \hat{\mathcal{L}}_t(s_t) := \frac{1}{n} \sum_{i=1}^n \| s_t(x_t^i) + w^i/\sigma_t \|^2, \quad t \in \{t_i\}_{i=0}^{N-1}. \tag{3.7}$$

In the sequel, we will let $\mathcal{D}_t := \{(x_t^i, w^i)\}_{i=1}^n$ denote the training data used in (3.7) for timestep $t$.

From Wibisono et al. (2024), we know that if the true score $\nabla \log p_t$ is Lipschitz continuous and $p_t$ is sub-Gaussian, then the minimax rate for estimating $\nabla \log p_t$ is given by $n^{-2/(D+4)}$.[3] Unfortunately, this type of bound ignores all latent structure, raising the question of whether or not diffusion models can learn latent structure in a sample efficient way.

**The Barron space $\mathcal{F}_1$.** Consider a single hidden layer network $f_m$ with $m$ neurons and mean-field scaling, $f_m(x) = \frac{1}{m} \sum_{i=1}^m u_i \sigma(\langle x, v_i \rangle)$. In the limit as $m$ tends to infinity, the summation may be replaced by integration $f_\infty(x) = \int u\sigma(\langle x, v \rangle) \, \mathrm{d}\mu(u, v)$ against a signed Radon measure $\mu$ over the neuron parameters $(u, v)$. This leads to the Barron space $\mathcal{F}_1$ (Bach, 2017; Mhaskar, 2004; Rotskoff & Vanden-Eijnden, 2019; Mei et al., 2018; Sirignano & Spiliopoulos, 2020), which models single hidden layer networks in the infinite width limit and feature learning regime (Chizat et al., 2020).

Concretely, given a Radon measure $\mu$ on a measurable space $\mathcal{V}$, recall that the *total variation norm* (TV) is defined as $\|\mu\|_{\mathrm{tv}} := \sup_g \int_{\mathcal{V}} g(v) \, \mathrm{d}\mu(v)$, where the supremum is over continuous functions $g : \mathcal{V} \mapsto [-1, 1]$. Given a basis function $\varphi_v(x)$, the TV-norm induces the space of functions $\mathcal{F}_1 := \{ f(x) = \int_{\mathcal{V}} \varphi_v(x) \, \mathrm{d}\mu(v) \mid \|\mu\|_{\mathrm{tv}} < \infty \}$. For a function $f \in \mathcal{F}_1$, its $\mathcal{F}_1$-norm is the infimum over all TV-norms of measures that can represent $f$, i.e., $\|f\|_{\mathcal{F}_1} := \inf\{ \|\mu\|_{\mathrm{tv}} \mid f(x) = \int_{\mathcal{V}} \varphi_v(x) \, \mathrm{d}\mu(v) \}$.

In this work, we will consider the special case $\mathcal{V} = \mathbb{S}^{p-1} \times \mathbb{S}^{p-1}$ for $p \in \mathbb{N}_+$ and $\varphi_v(x) = u\sigma(\langle x, v \rangle)$, where $(u, v) \in \mathcal{V}$ and $\sigma(\cdot) = \max\{0, \cdot\}$ is the ReLU activation. Hence, the induced class $\mathcal{F}_1$ consists of vector-valued maps from $\mathbb{R}^p \to \mathbb{R}^p$ that satisfy $\|f(x)\| \leqslant \|f\|_{\mathcal{F}_1} \|x\|$. In learning the score functions $\hat{s}_{t_i}$ via DSM (3.7), we will utilize norm-ball subsets of $\mathcal{F}_1$ to model the true score functions $\nabla \log p_{t_i}(x)$. This allows us to leverage the low-index structure of $\mathcal{F}_1$ (Bach, 2017) and obtain bounds for score estimation that scale with the *intrinsic*, rather than ambient, dimension of the problem.

Finally, while we consider $\mathcal{F}_1$ in this paper, we note that any $f \in \mathcal{F}_1$ can be $\varepsilon$-approximated in $L_2$ by a finite width $f_m$ with $m \leqslant O(\|f\|_{\mathcal{F}_1}^2 / \varepsilon^2)$ neurons (Bach, 2017, Section 2). Since we will show that we can approximate the underlying score function with $\mathcal{F}_1$-norm polynomial in $D$, our results easily extend to finite width one hidden layer networks with $\mathrm{poly}(D)$ width.

**Flow matching and interpolant models.** Flow matching models (Lipman et al., 2023; Liu et al., 2023) and stochastic interpolants (Albergo et al., 2023) have also been used recently in the context of generative modeling. In Appendix B, we relate the Barron norm of the drift vector field in these ODE based models with the Barron norm of the score function, which implies that our proof techniques can also be used to prove similar results for learning flow matching and interpolant drift terms.

**Notation.** We briefly review the (relatively standard) notation used in this work. For a $d$-dimensional vector $x$, the $\ell_p$ norm is denoted $\|x\|_p$; the notation $\|x\|$ is reserved for the Euclidean ($p = 2$) case. The (closed) $\ell_2$-ball of radius $r$ in $d$-dimension is denoted by $B_2(r, d)$; when $r = 1$, we use the shorthand $B_2(d)$. The unit sphere in $\mathbb{R}^d$ is denoted $\mathbb{S}^{d-1}$. The notation $O_d(\cdot)$ hides both universal constants and constants that depend arbitrarily on the variable $d$. Similarly, $\tilde{O}_d(\cdot)$ hides universal constants, constants that depend on $d$, and terms that may depend poly-logarithmically on $d$, i.e., terms of the form $\log^{O(d)}(\cdot)$. The notation $\mathrm{poly}(\cdot)$ indicates a polynomial dependence on the arguments, whereas $\mathrm{poly}_d(\cdot)$ indicates that the polynomial degrees are allowed to depend arbitrarily on $d$. The notation $a \lesssim b$ (resp. $a \gtrsim b$) indicates that there exists a universal positive constant $c$ such that $a \leqslant cb$ (resp. $a \geqslant cb$), and $a \asymp b$ means that $a \lesssim b$ and $a \gtrsim b$. Finally, we use $a \vee b$ to denote $\max\{a, b\}$.

---

[3] Interestingly, this rate is slower than the $n^{-2/(D+2)}$ rate for learning Lipschitz functions (Tsybakov, 2008).

### 3.1 LEARNING LATENT SUBSPACE STRUCTURE

We first consider a setting in which $x_0$ is supported on a $d$-dimensional ($d \ll D$) linear subspace:

$$x_0 = Uz_0, \quad z_0 \sim \pi_0(\cdot), \quad U \in O(D, d), \tag{3.8}$$

where $z_0$ is a $d$-dimensional random vector and $O(D, d) := \{U \in \mathbb{R}^{D \times d} \mid U^{\mathsf{T}}U = I_d\}$ denotes the $d$-dimensional orthogonal group in $\mathbb{R}^D$. Note that both the subspace dimension $d$ and the embedding matrix $U$ are unknown to the learner.

Recently, both Chen et al. (2023b) and Oko et al. (2023) consider learning diffusion models under the subspace structure (3.8). The main takeaway from both works is that the latent subspace dimension $d$, rather than the ambient dimension $D$, can govern the complexity of learning to sample from $p_0$ if the network used for learning satisfies various architectural assumptions. While insightful, these architectural assumption are difficult to satisfy in practice. Chen et al. (2023b) utilize a function class that is specifically tailored to the linear structure (3.8), in the sense that a linear autoencoder with prior knowledge of the latent dimension $d$ is used to reduce the learning problem to the latent space. The situation is improved in Oko et al. (2023), which considers fully-connected neural networks with bounded weight *sparsity*. Though closer to real-world architectures, optimizing networks with bounded sparsity constraints is computationally challenging in practice.

Our first result further closes the gap between theory and practice: we show that by optimizing over $\mathcal{F}_1$, the space of infinite-width single hidden layer networks, the latent subspace is adaptively learned without requiring prior latent dimension knowledge or difficult to impose sparsity constraints. We note that from a computational perspective, sufficiently wide shallow networks trained with gradient descent (GD) and weight decay will converge to the minimal $\mathcal{F}_1$-norm solution (Chizat & Bach, 2018). While control on the number of neurons required in the worst case is not available (optimizing over $\mathcal{F}_1$ is unfortunately NP-hard (Bach, 2017)), recent results have shown that under various assumptions on the data and task, this hardness can be avoided and GD can indeed learn low-index functions with a number of neurons that scales polynomially with an exponent that depends only on the intrinsic dimension (Abbe et al., 2022; Dandi et al., 2023; Lee et al., 2024). The fact that an exponential width might be required for GD to converge to the optimal solution is not in contradiction with the fact that a polynomial width is sufficient to approximate the optimal solution (cf. beginning of Section 3), since approximation is a weaker guarantee than convergence of GD.

Towards stating our first result, we begin with a few standard regularity assumptions.

**Assumption 3.1.** *The latent variable $z_0 \sim \pi_0(\cdot)$ is $\beta$-sub-Gaussian[4] (with $\beta \geqslant 1$), i.e.,*

$$\mathbb{E}\exp(\lambda(\|z_0\| - \mathbb{E}\|z_0\|)) \leqslant \exp(\lambda^2\beta^2/2), \quad \forall \lambda \in \mathbb{R}.$$

Our next assumption concerns the regularity of the score function of the latent distribution $\pi_t$, which denotes the marginal distribution of $z_t := U^{\mathsf{T}}x_t$.

**Assumption 3.2.** *The latent score $\nabla \log \pi_t$ is $L$-Lipschitz (with $L \geqslant 1$) on $\mathbb{R}^d$ for all $t \geqslant 0$.*

We emphasize that Assumption 3.2 concerns the Lipschitz regularity of the latent measure $\pi_t$ and *not* the ambient measure $p_t$; due to the subspace structure (3.8), the Lipschitz constant of the ambient score diverges as $t \to 0$. Finally, we define some shorthand notation for $t \in [0, T]$,

$$\mu_{t,x} := \mu_0 \vee \sigma_t\sqrt{D}, \quad \mu_{t,z} := \mu_0 \vee \sigma_t\sqrt{d}, \quad \bar{L} := cL(\mu_0 + \sqrt{d} + \|\nabla \log \pi_0(0)\|),$$

where $\mu_0 := (\mathbb{E}\|z_0\|^2)^{1/2}$ and $c \geqslant 1$ is a universal constant. Since $\sigma_t \to 1$ as $t \to \infty$, these constants are uniformly bounded above. Hence, we define $\mu_x := \lim_{t\to\infty} \mu_{t,x}$ and $\mu_z := \lim_{t\to\infty} \mu_{t,z}$.

**Theorem 3.3.** *Suppose that $p_0$ follows the latent structure (3.8), and that both Assumption 3.1 and Assumption 3.2 hold. Fix a $t > 0$ and define*

$$\mathcal{F}_t := \{s : \mathbb{R}^D \mapsto \mathbb{R}^D \mid \|s\|_{\mathcal{F}_1} \leqslant R_t\}, \quad R_t := \bar{R}_t n^{\frac{d+1}{2(d+5)}} + \frac{D}{\sigma_t^2}, \tag{3.9}$$

*where $\bar{R}_t$ does not depend on $n$.[5] Suppose that $n$ satisfies*

$$n \geqslant n_0(t) := \mathrm{poly}(D, 1/\sigma_t, \mu_{t,x} \vee \beta) \cdot \mathrm{poly}_d(\bar{L}, \mu_{t,z} \vee \beta). \tag{3.10}$$

---

[4]This definition is also referred to as *norm*-sub-Gaussian in the literature (see e.g. Jin et al., 2019).

[5]The explicit dependence of $\bar{R}_t$ on the other problem parameters is detailed in the proof.

*Then, the empirical risk minimizer $\hat{s}_t \in \arg\min_{s \in \mathscr{F}_t} \hat{\mathcal{L}}_t(s)$ satisfies:*

$$\mathbb{E}_{\mathcal{D}_t}[\mathcal{R}_t(\hat{s}_t)] \leqslant \tilde{O}_d(1) \left[ \frac{D^2}{\sigma_t^2 n} (\bar{L}(\mu_{t,z} \vee \beta))^{d+3} (\mu_{t,x} \vee \beta)^2 \right]^{\frac{2}{d+5}} + \tilde{O}_d(1) \sqrt{\frac{D^3}{\sigma_t^6 n} (\mu_{t,x} \vee \beta)^2}.$$

Focusing on the factors $n, d, D$, Theorem 3.3 states that the score matching risk scales as $\mathbb{E}_{\mathcal{D}_t}[\mathcal{R}_t(\hat{s}_t)] \leqslant \tilde{O}_d(1)(\text{poly}(D)/n)^{2/(d+5)}$, demonstrating that exponential dependence enters only through the latent subspace dimension $d$ and not the ambient dimension $D$. Some further remarks regarding Theorem 3.3 are in order. First, we can obtain a high-probability bound with minor modifications to the proof; we omit these details in the interest of brevity. Second, our $n^{-2/(d+5)}$ rate nearly matches the minimax optimal rate of score matching from Wibisono et al. (2024), but with the subspace dimension $d$ replacing the ambient dimension $D$; we leave showing a $n^{-2/(d+4)}$ rate to future work. Last, as noted before, Chen et al. (2023b, Theorem 2) proves a related result for learning score functions under the subspace structure (3.8). However, Theorem 3.3 substantially improves their result in the following ways. First, as already mentioned, our result does not require any specialized architectures, but instead applies to learning directly in the Barron space $\mathcal{F}_1$ of single hidden layer neural networks. Additionally, our result also provides several technical improvements: (a) our leading dependence on $n$ is improved to $n^{-2/(d+5)}$ from $n^{-(2-o(1))/(d+5)}$,[6] and (b) our dependence on $\sigma_t$ is improved to $\sigma_t^{-4/(d+5)}$ instead of $\sigma_t^{-2}$ in the regime as $t \to 0$.

We now use Theorem 3.3 to provide an end-to-end sample complexity bound for sampling from $p_0$.

**Corollary 3.4.** *Fix $\varepsilon, \zeta \in (0,1)$. Suppose that $p_0$ follows the latent structure (3.8), and that both Assumption 3.1 and Assumption 3.2 hold. Consider the exponential integrator (3.4) with:*

$$T = c_0 \log\left( \frac{\sqrt{D} \vee \mu_0}{\varepsilon} \right), \quad N = 2 \left\lceil c_1 \frac{D \vee \mu_0^2}{\varepsilon^2} \left[ \log^2\left( \frac{\sqrt{D} \vee \mu_0}{\varepsilon} \right) + \log^2\left( \frac{1}{\zeta} \right) \right] \right\rceil, \quad (3.11)$$

*and reverse process discretization timesteps $\{\tau_i\}_{i=0}^{N}$ defined as:*

$$\tau_i = \begin{cases} 2(T-1)\frac{i}{N} & \text{if } i \in \{0, \ldots, N/2\}, \\ T - \zeta^{2i/N - 1} & \text{if } i \in \{N/2 + 1, \ldots, N\}. \end{cases} \quad (3.12)$$

*Next, define the forward process timesteps $\{t_i\}_{i=0}^{N-1}$ by $t_i := T - \tau_{N-i}$. Suppose the exponential integration scheme is run with score functions $\{\hat{s}_{t_i}\}_{i=0}^{N-1}$, where $\hat{s}_{t_i} \in \arg\min_{s \in \mathscr{F}_{t_i}} \hat{\mathcal{L}}_{t_i}(s)$ with $\mathscr{F}_t$ as defined in (3.9). Suppose furthermore that $n$ satisfies:*

$$n \geqslant \tilde{O}_d(1) \max\left\{ \frac{D^2}{\zeta} (\bar{L}(\mu_z \vee \beta))^{d+3} (\mu_x \vee \beta)^2 \cdot \varepsilon^{-(d+5)}, \frac{D^3}{\zeta^3} (\mu_x \vee \beta)^2 \cdot \varepsilon^{-4}, n_0(\zeta) \right\},$$

*where $n_0(\cdot)$ is defined in (3.10). With constant probability (over the randomness of the training datasets $\{\mathcal{D}_{t_i}\}_{i=0}^{N-1}$), we have that $\text{KL}(p_\zeta \parallel \text{Law}(\hat{y}_{T-\zeta})) \leqslant \varepsilon^2$, where $\text{Law}(\hat{y}_{T-\zeta})$ refers to the distribution of the random vector $\hat{y}_{T-\zeta}$.*

Treating $\bar{L}$ and $\beta$ as constants, Corollary 3.4 prescribes a rate of $n \geqslant \tilde{O}_d(1)\frac{\text{poly}(D)}{\zeta}\varepsilon^{-(d+5)}$ (after a burn-in on $n$) to obtain a sampler that satisfies $\text{KL}(p_\zeta \parallel \text{Law}(\hat{y}_{T-\zeta})) \leqslant \varepsilon^2$. To the best of our knowledge, this is the first end-to-end sample complexity bound for learning a diffusion model over *single hidden layer* neural networks that adapts to the intrinsic dimensionality of the problem. Note that as Corollary 3.4 controls the KL-divergence between the true data distribution $p_\zeta$ and the distribution $\text{Law}(\hat{y}_{T-\zeta})$ of the final iterate of the exponential integrator (3.4), by Pinsker's inequality this also implies control on the TV-distance $\|p_\zeta - \text{Law}(\hat{y}_{T-\zeta})\|_{\text{tv}}$. Furthermore, we can upgrade Corollary 3.4 to a high probability guarantee by utilizing a high probability variant of Theorem 3.3.

The parameter $\zeta > 0$ is the early stopping parameter often found in practical implementations of diffusion models (cf. Karras et al. (2022)). Note that this is necessary since $p_0$ is supported on a lower-dimensional manifold and hence $\nabla \log p_0$ is not smooth on all of $\mathbb{R}^D$. We remark that bounds comparing the original $p_0$ to $\text{Law}(\hat{y}_{T-\zeta})$ are possible in Wasserstein distance by adopting the techniques from e.g. Chen et al. (2023d, Section 3.2); we omit these calculations.

---

[6]However, their logarithmic dependence on $n$ is only through $\log^{O(1)}(n)$ terms instead of our $\log^{O(d)}(n)$.

Compared to Chen et al. (2023b, Theorem 3), who obtain a $n \geqslant \tilde{O}_d(1)(\varepsilon\sqrt{\zeta})^{-(d+5)/(1-o(1))}$ rate in the case of a latent subspace-aware architecture, we see that our bound also improves the dependency on $\zeta$. This is important, because ultimately $\zeta$ will be chosen to decay to zero as $n \to \infty$. We do remark, however, that Chen et al. (2023b, Theorem 3) only depends polylogarithmically on the ambient dimension $D$ instead of polynomially. This can be traced back in their analysis to imposing the constraint that their score functions are uniformly bounded, i.e., $\sup_{z,t}\|s(z,t)\| \leqslant K$, which allows truncation arguments to avoid picking up extra $\text{poly}(D)$ factors. We choose to not impose such constraints in our model class, as this adds another hyperparameter that must be tuned in practice. We leave open the question of whether or not these $\text{poly}(D)$ pre-factors in the sample complexity can be removed without further modifications (e.g., clipping) of the $\mathcal{F}_1$ hypothesis class.

Compared to Oko et al. (2023, Theorem 6.4), Corollary 3.4 also relaxes a few technical assumptions, including a uniformly lower bounded density $\pi_0$, and a requirement that $\pi_0$ be $C^\infty$ near the boundary $[-1,1]^d$. On the other hand, their work obtains a sharper rate on $W_1(p_0, \text{Law}(\hat{y}_{T-\zeta})) \lesssim n^{-(3-\delta)/(d+4)}$ for any $\delta > 0$. We also leave open the question of whether these extra assumptions can be used to strengthen our guarantees for learning over $\mathcal{F}_1$.

## 3.2 LEARNING LATENT INDEPENDENCE

We now consider a different kind of latent structure – here generated by independence – as opposed to the low-dimensional subspace setting just studied. Specifically, we suppose that for some $K \in [D]$,

$$x_0 = Uz_0, \quad z_0 \sim (z_0^{(1)}, \ldots, z_0^{(K)}), \quad z_0^{(i)} \sim \pi_0^{(i)}(\cdot), \tag{3.13}$$

where $U \in O(D) := \{U \in \mathbb{R}^{D \times D} \mid U^\mathsf{T}U = I_D\}$, $z_0^{(i)} \in \mathbb{R}^{d_i}$ with $\sum_{i=1}^K d_i = D$, and where $z_0^{(i)}$ is independent of $z_0^{(j)}$ for $i \neq j$. This setting differs from the previous case in that the latent structure considered is *probabilistic* and not geometric; it is possible for $z_0$ to satisfy (3.13) but still be fully supported on $\mathbb{R}^D$. Similar to the linear subspace setting, we assume that the orthonormal matrix $U$, the number of components $K$, and the dimensionality of each component $\{d_i\}_{i=1}^K$ are all unknown to the learner, and we study the adaptive properties of $\mathcal{F}_1$ in the presence of this latent structure. We begin by imposing a similar set of assumptions as in the subspace case (cf. Section 3.1).

**Assumption 3.5.** *For all $i \in [K]$, we have that $\pi_0^{(i)}$ is $\beta_i$-sub-Gaussian (for $\beta_i \geqslant 1$), i.e.,*

$$\mathbb{E}\exp(\lambda(\|z_0^{(i)}\| - \mathbb{E}\|z_0^{(i)}\|)) \leqslant \exp(\lambda^2\beta_i^2/2), \quad \lambda \in \mathbb{R}.$$

Our next assumption again deals with the latent measure $\pi_t$, defined as the marginal distribution of $z_t := U^\mathsf{T}x_t$. We decompose $z_t = (z_t^{(1)}, \ldots, z_t^{(K)})$ into coordinate groups as for $z_0$, and we define $\pi_t^{(i)}$ as the marginal distribution of $z_t^{(i)}$.

**Assumption 3.6.** *For all $i \in [K]$, $\nabla \log \pi_t^{(i)}$ is $L_i$-Lipschitz (for $L_i \geqslant 1$) on $\mathbb{R}^{d_i}$ for all $t \geqslant 0$.*

As before, we define some shorthand notation for $t \in [0,T]$ and $i \in [K]$:

$$\mu_{t,x}^{(i)} := \mu_0^{(i)} \vee \sigma_t\sqrt{d_i}, \quad \bar{L}_i := cL_i(\mu_0^{(i)} + \sqrt{d_i} + \|\nabla\log\pi_0^{(i)}(0)\|),$$

where $\mu_0^{(i)} := (\mathbb{E}\|z_0^{(i)}\|^2)^{1/2}$ and $c \geqslant 1$ is a universal constant. Furthermore, we combine the individual constants together as $\mu_0 := \sqrt{\sum_{i=1}^K (\mu_0^{(i)})^2}$, $\beta := \sqrt{\sum_{i=1}^K \beta_i^2}$, and $\mu_{t,x} := \mu_0 \vee \sigma_t\sqrt{D}$. Finally, as before, we let $\mu_x^{(i)} := \lim_{t\to\infty}\mu_{t,x}^{(i)}$ and $\mu_x := \lim_{t\to\infty}\mu_{t,x}$. Mirroring Theorem 3.3, we first bound the error on the learned score functions under the latent independent structure (3.13).

**Theorem 3.7.** *Suppose that $p_0$ follows the latent structure* (3.13)*, and that both Assumption 3.5 and Assumption 3.6 hold. Fix a $t > 0$ and define*

$$\mathscr{F}_t := \{s : \mathbb{R}^D \mapsto \mathbb{R}^D \mid \|s\|_{\mathcal{F}_1} \leqslant R_t\}, \quad R_t := \sum_{i=1}^K \bar{R}_t^{(i)} n^{\frac{d_i+1}{2(d_i+5)}}, \tag{3.14}$$

*where $\bar{R}_t^{(i)}$ does not depend on $n$. Suppose that $n$ satisfies*

$$n \geqslant n_0(t) := \text{poly}(D, 1/\sigma_t, \mu_{t,x} \vee \beta) \cdot \max_{i\in[K]} \text{poly}_{d_i}(\bar{L}_i, \mu_{t,x}^{(i)} \vee \beta^{(i)}). \tag{3.15}$$

*Then, the empirical risk minimizer* $\hat{s}_t \in \arg\min_{s \in \mathscr{F}_t} \hat{\mathcal{L}}_t(s)$ *satisfies:*

$$\mathbb{E}_{\mathcal{D}_t}[\mathcal{R}_t(\hat{s}_t)] \leqslant \sum_{i=1}^{K} \tilde{O}_{d_i}(1) \left[ \frac{D^2 K}{\sigma_t^2 n} (\bar{L}_i(\mu_{t,x}^{(i)} \vee \beta_i))^{d_i+3} (\mu_{t,x} \vee \beta)^2 \right]^{\frac{2}{d_i+5}} + \tilde{O}(1) \sqrt{\frac{D^2}{\sigma_t^4 n} (\mu_{t,x} \vee \beta)^2}.$$

Ignoring all parameters other than $K$ and $n$, Theorem 3.7 states that the score matching risk $\mathbb{E}_{\mathcal{D}_t}[\mathcal{R}_t(\hat{s}_t)]$ scales as $\mathbb{E}_{\mathcal{D}_t}[\mathcal{R}_t(\hat{s}_t)] \lesssim \sum_{i=1}^{K} (K/n)^{2/(d_i+5)}$, which again captures the intrinsic dimensionality of the problem. To the best of our knowledge, this is the first result establishing a score function error bound in the setting (3.13) that depends primarily on the latent $d_i$'s. Our final result mirrors that of Corollary 3.4 and establishes an end-to-end sampling guarantee for this setting.

**Corollary 3.8.** *Fix $\varepsilon, \zeta \in (0,1)$. Suppose that $p_0$ follows the latent structure (3.13), and that both Assumption 3.5 and Assumption 3.6 hold. Consider the exponential integrator (3.4) with $(N,T)$ as in (3.11) and reverse process discretization timesteps $\{\tau_i\}_{i=0}^{N}$ defined as in (3.12). Next, define the forward process timesteps $\{t_i\}_{i=0}^{N-1}$ by $t_i := T - \tau_{N-i}$. Suppose the exponential integration scheme is run with score functions $\{\hat{s}_{t_i}\}_{i=0}^{N-1}$, where $\hat{s}_{t_i} \in \arg\min_{s \in \mathscr{F}_{t_i}} \hat{\mathcal{L}}_{t_i}(s)$ with $\mathscr{F}_t$ as defined in (3.14). Suppose that $n \geqslant n_0(\zeta)$ satisfies:*

$$n \geqslant (\mu_x \vee \beta)^2 \max \left\{ \max_{i \in [K]} \left\{ \frac{\tilde{O}_{d_i}(1) D^2}{\zeta} K^{(d_i+7)/2} (\bar{L}_i(\mu_x^{(i)} \vee \beta))^{d_i+3} \cdot \varepsilon^{-(d_i+5)} \right\}, \frac{\tilde{O}(1) D^2}{\zeta^2} \cdot \varepsilon^{-4} \right\}.$$

*where $n_0(\cdot)$ is defined in (3.15). With constant probability (over the randomness of the training datasets $\{\mathcal{D}_{t_i}\}_{i=0}^{N-1}$), we have that $\mathrm{KL}(p_\zeta \,\|\, \mathrm{Law}(\hat{y}_{T-\zeta})) \leqslant \varepsilon^2$.*

Corollary 3.8 states that $n \geqslant \mathrm{poly}(D)/\zeta \cdot \max_{i \in [K]} \mathrm{poly}_{d_i}(K) \varepsilon^{-(d_i+5)}$ samples (ignoring $\mu_x, \beta, L_i$) suffice to ensure that $\mathrm{KL}(p_\zeta \,\|\, \mathrm{Law}(\hat{y}_{T-\zeta})) \leqslant \varepsilon^2$. We note that, unlike the latent subspace setting of Section 3.1, given Assumption 3.6 it is possible to prove a bound on $\mathrm{KL}(p_0 \,\|\, \mathrm{Law}(\hat{y}_{T-\zeta}))$ directly, since $\nabla \log p_t$ is uniformly Lipschitz for all $t \geqslant 0$. This can be done by using Chen et al. (2023d, Theorem 2) to analyze the backwards exponential integrator process (3.4) instead of the results of Benton et al. (2024). We elect to utilize the latter's analysis in the interest of consistency with Section 3.1 where it is required.

### 3.2.1 NON-ORTHOGONAL INDEPENDENT COMPONENTS

Section 3.2 shows that a diffusion model based on a single hidden layer network can adapt to hidden independent component structure. A natural question is whether this extends to the non-orthogonal case, similar to independent component analysis (ICA) (Herault et al., 1985). Here, we explain why a direct extension of our argument works at $t = 0$ case, but breaks whenever $t > 0$ since the addition of noise breaks the independence structure. We then use data whitening to address the issue.

Recall from (3.13) that $z_0 = (z_0^{(1)}, \ldots, z_0^{(K)})$ where $x^{(i)} \in \mathbb{R}^{d_i}$ for $d_1 + \cdots + d_K = D$, and where each of the $K$ components are sampled independently. Now, let us assume that $x_0 = Az_0$, where $A \in \mathbb{R}^{D \times D}$ is invertible, but not necessarily orthonormal. Note that for $t = 0$, the score function $\nabla \log p_0$ can be expressed as $\nabla \log p_0(x) = \sum_{i=1}^{K} A^{-\mathsf{T}} P_i^{\mathsf{T}} \nabla \log \pi_0^{(i)}(P_i A^{-1} x)$, where $P_i \in \mathbb{R}^{d_i \times D}$ selects the $d_i$ coordinates that correspond to the $i$-th variable group. Hence, the score function has the structure of a sum of low-index functions, and the $\mathcal{F}_1$ norm of the score can be bounded in terms of the sum of $\mathcal{F}_1$ norms of each component. However, once isotropic noise is added, this structure is lost in general because the isotropy is not preserved in the latent space ($z_t = A^{-1}(m_t x_0 + \sigma_t w) = m_t z_0 + \sigma_t A^{-1} w$).

A possible fix for this issue is to first whiten the data, and then to apply the results of Section 3.2 to the whitened data. Specifically, write $\mathrm{Cov}(x_0) = A\Sigma A^{\mathsf{T}}$, where $\Sigma$ is a block diagonal matrix with $K$ blocks $\Sigma^{(1)}, \ldots, \Sigma^{(K)}$. Now consider the transform $\bar{x}_0 := \Sigma^{-\frac{1}{2}} A^{-1} x_0$, which orthogonalizes the components and whitens each of them independently. Indeed, $\bar{x}_0$ follows the latent structure (3.13), and hence the results from Section 3.2 directly apply to $\bar{x}_0$. However, one caveat with this approach is that the whitening factor $\Sigma^{-1/2} A^{-1}$ must be learned from the data samples $x_0^i \sim p_0(\cdot)$. Standard results in covariance estimation (see e.g., Wainwright, 2019, Chapter 6) allow us to learn $\Sigma^{-1/2} A^{-1}$ (modulo rotation) up to $n^{-1/2}$ accuracy. It is an interesting question that we leave to future work to study how the associated estimation error propagates through both the learning and sampling procedures when obtaining a final sample complexity bound.

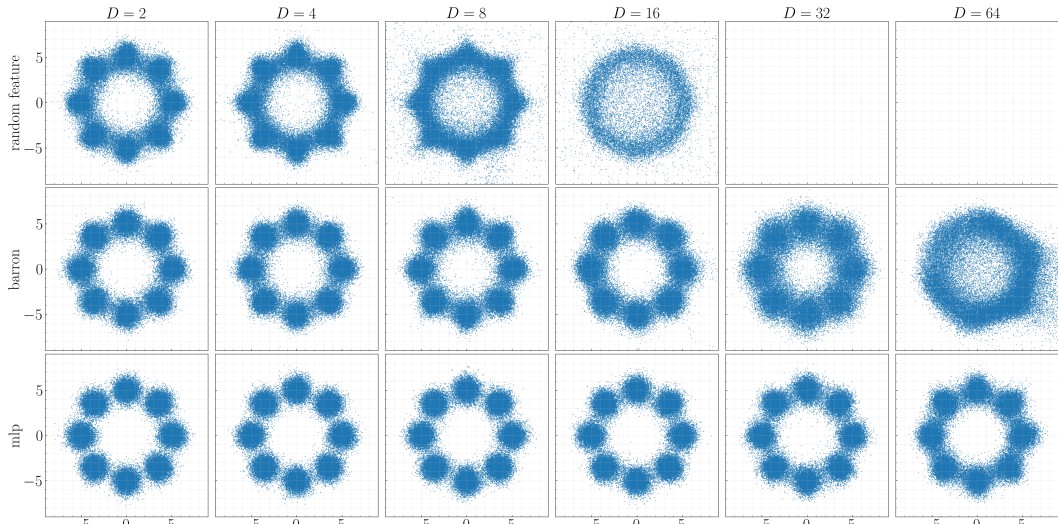

**Figure 1: Hidden linear structure.** Experimental results comparing the performance of a single hidden layer neural network, a two hidden layer neural network, and a random feature model in the presence of a hidden linear structure. The target is an eight-mode Gaussian mixture model in $d = 2$ dimensions embedded into a $D$ dimensional space via an unknown orthogonal transformation; this transformation is used to project generated samples down to $\mathbb{R}^2$. The Barron model continues to perform well in moderately high dimension which is consistent with our theory, whereas the random feature model struggles to identify the orthogonal transform and performs significantly worse than the other two models.

## 4 EXPERIMENTAL RESULTS

In this section, we present experimental results that verify our theoretical claims. To this end, we compare the performance of a single hidden layer (Barron) neural network with a random feature network (Bach, 2015; Rahimi & Recht, 2007) and a two hidden layer neural network as a function of the ambient dimension $D$ on two synthetic tasks that isolate the hidden structures we analyzed. To make the comparison as fair as possible, all networks are trained with gradient descent for $2.5 \times 10^5$ steps. We use a cosine decay schedule that initializes the learning rate from $5 \times 10^{-4}$ and ends at 0. The single hidden layer neural network has a hidden layer width of 1024 neurons; to keep the number of parameters fixed, the corresponding random feature model has 2048 neurons. To avoid an unnatural reduction in width for the two hidden-layer neural network, we keep the width fixed at 1024 rather than fixing the number of parameters. We sample from each learned diffusion model using 1024 evenly-spaced timesteps on the horizon $t \in [0, 5]$.

**Hidden linear structure.** We consider data points $x \in \mathbb{R}^D$ given by $x = Uz$ where $z \in \mathbb{R}^d$ with $d = 2$ is drawn from the Gaussian mixture model $\rho = \frac{1}{N} \sum_{i=1}^{N} \mathsf{N}(\mu_i, \Sigma_i)$ with $N = 8$. Here, $U \in \mathbb{R}^{D \times d}$ is a random orthogonal matrix that we keep fixed across all networks for constant $D$. We take $\Sigma_i = \frac{1}{2}I$ for all $i$ and $\mu_i = r \left( \cos \left( \frac{2\pi i}{N} \right), \sin \left( \frac{2\pi i}{N} \right) \right)^\top$, which leads to an eight-mode, narrowly-spaced mixture evenly distributed on the circle of radius $r = 5$. We study the performance of each network architecture as a function of $D$ ranging from $D = 2$ to $D = 64$ in powers of two. Results are shown in Figure 1, where we visualize $5 \times 10^4$ samples $\hat{z} = U^\top \hat{x}$ from each model. In each case, performance degrades as $D$ increases, reflective of the increasing difficulty of the learning problem. Notably, performance degrades significantly more rapidly for the random feature model than for the Barron model despite the fact that both models have the same number of parameters; in particular, for $D = 32$ and $D = 64$, samples do not lie within the visualized frame because the model becomes unstable. This is consistent with our theory, which predicts that sample complexity guarantees for Barron will depend exponentially on $d = 2$ rather than $D$, whereas the random feature model does not adapt to the latent structure Bach (2017). Unsurprisingly, given its extra parameters and additional nonlinear layer, the two-layer network maintains the highest performance for all $D$.

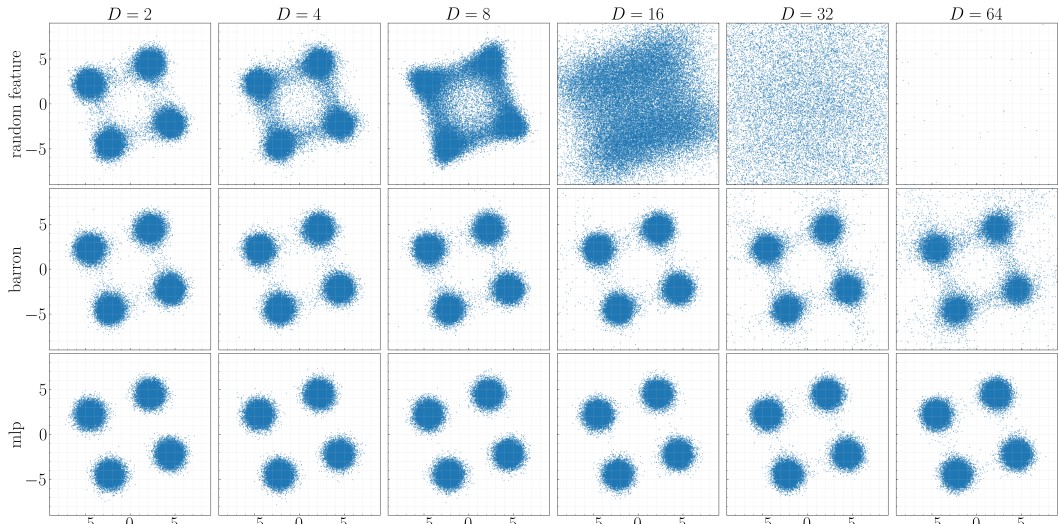

**Figure 2: Product of hidden independent components.** Experimental results comparing the performance of a random feature model, a single hidden layer network, and a two hidden layer network for a dataset with hidden independence structure. The target is a product of $K$ four mode Gaussian mixtures in $\mathbb{R}^2$ forming a randomly rotated square, so that $D = 2K$; each mixture has a different random rotation. Approximate samples are generated in $\mathbb{R}^D$ and only the first mixture is visualized. Similar to Figure 1, the single hidden layer (and two hidden layer) network is able to detect the hidden independence structure, while the performance of the random feature model rapidly breaks down.

**Hidden independent components.** We now consider data points $x = (x_1, x_2, \ldots, x_K)^\top \in \mathbb{R}^D$ where $D = Kd$ and each $x_k \in \mathbb{R}^d$. We again take $d = 2$ for visualization, and we draw each $x_k$ from the Gaussian mixture model $\rho_k = \frac{1}{N} \sum_{i=1}^{N} \mathsf{N}(\mu_i^k, \Sigma_i)$ with $N = 4$. Similar to the previous experiment, we take $\Sigma_i = \frac{1}{2}I$ for all $i$ and $\mu_i^k = r\left(\cos\left(\frac{2\pi i}{N} + \phi_k\right), \sin\left(\frac{2\pi i}{N} + \phi_k\right)\right)^\top$ where $\phi_k \sim \mathsf{Unif}(0, 2\pi)$ is fixed across all network architectures for fixed $D$. This construction ensures that each block $x_k \in \mathbb{R}^d$ is drawn from a Gaussian mixture with four modes placed on the corners of a square of side length $\sqrt{2}r$ that has been randomly rotated by an angle $\phi_k$ in the plane. Results are shown in Figure 2, where we visualize $5 \times 10^4$ samples of $\hat{x}_1 \in \mathbb{R}^2$. Similar to the previous experiment, Barron networks significantly outperform random feature models, while the two hidden-layer network outperforms Barron.

## 5 CONCLUSION

In this work, we showed that diffusion models based on single hidden layer neural networks applied to data from distributions that contain low dimensional structure – specifically, linear subspace and hidden independent component structure – exhibit favorable sample complexity bounds that primarily depend on the latent dimensionality of the problem, thereby avoiding the curse of dimensionality. We accomplish this by leveraging the low-index structure of the Barron space, which allows us to avoid specific latent-aware architectural modifications and computationally intractable sparsity constraints, both of which have been used to obtain similar results in earlier work.

Several exciting future research threads arise directly from our study. The most pertinent direction is to increase the scope of the latent structures covered by our analysis, to include, for example, non-linear manifolds. Another related question is whether or not favorable results that avoid the curse of dimensionality can be shown for latent diffusion models (Rombach et al., 2022), which first learn an autoencoder before learning a diffusion model in the autoencoder's latent space. On the algorithmic front, an interesting open question is whether or not gradient-based optimization algorithms can efficiently learn the low-index structure associated with the latent models studied in this paper. Finally, improving our rates to match the minimax optimal score estimation rates of Wibisono et al. (2024) – with the latent dimension playing the role of the ambient dimension – is another exciting area for future work.

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

# A    PROOF IDEAS

Here, we outline the key proof ideas behind the results in Section 3.1 and Section 3.2. We focus our discussion exclusively on bounding the error of the score function estimate, as translating score error into sample quality bounds is already well-established in the literature (cf. Section 2). For this discussion, we fix a specific value of $t > 0$, noting from our discussion in Section 3 (specifically Equation (3.7)) that we learn separate score models for a fixed sequence of forward process timesteps $\{t_i\}_{i=0}^{N-1}$.

## A.1    BASIC INEQUALITY

Recall that $s_t \in \arg\min_{s \in \mathscr{F}_t} \hat{\mathcal{L}}_t(s)$ is the empirical risk minimizer (ERM) of the empirical denoising loss $\hat{\mathcal{L}}_t$ over the function class $\mathscr{F}_t = \{s : \mathbb{R}^D \mapsto \mathbb{R}^D \mid \|s\|_{\mathcal{F}_1} \leqslant R_t\}$, where the norm bound $R_t$ will be determined. Our first step uses the link between the $L_2(p_t)$ score error $\mathcal{R}_t(s)$ and the DSM loss $\mathcal{L}_t(s)$ in addition to standard arguments from the analysis of ERM to show the following basic inequality for all $\varepsilon \geqslant 0$:

$$\mathbb{E}_{\mathcal{D}_t}[\mathcal{R}_t(\hat{s}_t)] \leqslant (1+\varepsilon) \inf_{s \in \mathscr{F}_t} \mathcal{R}_t(s) + \mathbb{E}_{\mathcal{D}_t} \sup_{s \in \mathscr{F}_t} [\mathcal{L}_t(s) - (1+\varepsilon)\hat{\mathcal{L}}_t(s)] + \varepsilon \cdot C_t, \tag{A.1}$$

where $C_t := \mathbb{E}\,\mathrm{tr}\,\mathrm{Cov}(\sigma_t^{-2}(m_t x_0 - x_t) \mid x_t)$. The basic inequality (A.1) contains three key terms: an *approximation-theoretic* term $T_1 := (1+\varepsilon) \inf_{s \in \mathscr{F}_t} \mathcal{R}_t(s)$ which measures how well the $\mathcal{F}_1$-norm bounded subset $\mathscr{F}_t$ approximates the true score, a *uniform convergence* term $T_2 := \mathbb{E}_{\mathcal{D}_t} \sup_{s \in \mathscr{F}_t} [\mathcal{L}_t(s) - (1+\varepsilon)\hat{\mathcal{L}}_t(s)]$ over the function class $\mathscr{F}_t$, and a third *offset* term $T_3 := \varepsilon \cdot C_t$ which trades off a fast rate for $T_2$ (controlled via $\varepsilon$) with the constant offset $C_t$ between the score error and the DSM loss. In addition, the first two terms are in tension with each other, and must be carefully balanced to achieve the desired rate.

## A.2    APPROXIMATION OF STRUCTURED MODELS WITH $\mathcal{F}_1$

Our analysis is based on careful control of the approximation error term $T_1$ in (A.1) of the structured models we consider in a way such that the requisite $\mathcal{F}_1$-norm $R_t$ does not depend exponentially on the ambient dimension $D$. This is accomplished by first understanding the low-dimensional structure present in the score functions $\nabla \log p_t$, and then arguing that this low-dimensional structure can be approximated with a norm bound $R_t$ that depends reasonably on $D$.

**Subspace structure.**    We first consider the low dimensional subspace from Section 3.1. Under this model, we have the following expression relating the score $\nabla \log p_t$ to the score $\nabla \log \pi_t$ of the latent $z_t$.

**Proposition A.1** (see e.g., Chen et al. (2023b, Lemma 1)). *The following decomposition holds under the subspace model* (3.8)*:*

$$\nabla \log p_t(x) = U \nabla \log \pi_t(U^\mathsf{T} x) - \frac{1}{\sigma_t^2}(I - UU^\mathsf{T})x. \tag{A.2}$$

**Independent components.**    We next consider the independent structure from Section 3.2. Under this model, we have the following decomposition for the score.

**Proposition A.2.** *The following holds under the independent components model* (3.13)*:*

$$\nabla \log p_t(x) = \sum_{i=1}^{K} U P_i^\mathsf{T} \nabla \log \pi_t^{(i)}(P_i U^\mathsf{T} x), \tag{A.3}$$

*where $P_i \in \mathbb{R}^{d_i \times D}$ selects the coordinates corresponding to the $i$-th variable group.*

Note that the proof of Proposition A.2 follows directly from the standard change of variables formula, and the fact that the distribution of $w$ is unchanged when pre-multiplied by $U^\mathsf{T}$.

The score function decompositions (A.2) and (A.3) both exhibit similar structure, where latent score functions are embedded into a score function in the ambient space via a linear encoding/decoding process. Fortunately, this embedding preserves the $\mathcal{F}_1$-norm of the underlying function.

**Fact A.3.** *Let $f : \mathbb{R}^d \mapsto \mathbb{R}^d$ have bounded $\mathcal{F}_1$-norm, and let $U \in O(D, k)$. Consider $g : \mathbb{R}^D \mapsto \mathbb{R}^D$ defined as $g(x) = Uf(U^\mathsf{T}x)$. We have that $\|g\|_{\mathcal{F}_1} = \|f\|_{\mathcal{F}_1}$.*

Hence, if the latent function $f : \mathbb{R}^d \to \mathbb{R}^d$ can be approximated well in $\mathcal{F}_1$, then the embedded function is also approximated well with the *same* $\mathcal{F}_1$ norm. This is the key observation that enables our results. Concretely, suppose that $\hat{f} : \mathbb{R}^d \mapsto \mathbb{R}^d$ has bounded $\mathcal{F}_1$ norm and approximates $f : \mathbb{R}^d \to \mathbb{R}^d$ via $\sup_{z \in B_2(d,M)}\|f(z) - \hat{f}(z)\| \leqslant \varepsilon$. Then, $\hat{g}(x) = U\hat{f}(U^\mathsf{T}x)$ approximates $g(x) = Uf(U^\mathsf{T}x)$ via $\sup_{x \in B_2(D,M)}\|g(x) - \hat{g}(x)\| \leqslant \varepsilon$, and $\|\hat{g}\|_{\mathcal{F}_1} = \|\hat{f}\|_{\mathcal{F}_1}$.

It remains to argue that $\mathcal{F}_1$ can approximate low dimensional functions well. Fortunately, the approximation properties of $\mathcal{F}_1$ functions over various function classes is well-understood (Bach, 2017; Jacot et al., 2024). In particular, we utilize the following $\mathcal{F}_1$ approximation result for Lipschitz continuous functions, adopted from Bach (2017, Proposition 6).

**Lemma A.4.** *Let $f : \mathbb{R}^d \to \mathbb{R}^d$ be L-Lipschitz and B-bounded on $B_2(d, M)$. Define $K := B \vee LM$. For any $\varepsilon \in (0, K/2)$, there exists an $f_\varepsilon \in \mathcal{F}_1$ such that $\sup_{x \in B_2(d,M)}\|f(x) - f_\varepsilon(x)\| \leqslant \varepsilon$ and:*

$$\|f_\varepsilon\|_{\mathcal{F}_1} \leqslant O_d(1) K \left(\frac{K}{\varepsilon}\right)^{(d+1)/2} \log^{(d+1)/2}\left(\frac{K}{\varepsilon}\right).$$

## A.3 UNIFORM CONVERGENCE OF THE DSM LOSS

With the approximation result in place, we turn to the analysis of uniform convergence term $T_2$ in (A.1). Since the DSM loss is a least-squares regression problem, we can utilize existing results for analyzing generalization error with smooth losses (Srebro et al., 2010). However, the main technical hurdle here is dealing with the fact that the data tuples $(x_0, x_t)$ are not uniformly bounded, which is a technical assumption needed in many of these arguments.[7] While this can be handled straightforwardly for a fixed time $t$ via standard truncation arguments, one challenge is ensuring that the resulting high probability bounds degrade nicely as $t \to 0$. The reason this is necessary is because the smallest timescale $t_0$ used will ultimately scale with the number of datapoints $n$.

To highlight the class of issues that arise in our truncation arguments, consider the latent score function $\nabla \log \pi_t$ arising from the subspace structure setting (3.8). In order to apply a truncation argument for analyzing $T_2$, we need to argue that $\|\nabla \log \pi_t(z)\|$ is bounded uniformly over a high-probability truncation set $z \in B_2(d, M)$. By the continuity of $\nabla \log \pi_t$, we know that $\sup_{z \in B_2(d,M)}\|\nabla \log \pi_t(z)\| = A_t < \infty$. However, we need to control the behavior of $A_t$ as $t \to 0$. By leveraging the perturbation analysis of Lee et al. (2023), we show that for all $t \geqslant 0$, under our assumptions, the inequality $\|\nabla \log \pi_t(z)\| \leqslant \bar{L}(1 + \|z\|)$ holds for all $z \in \mathbb{R}^d$. Hence, we can bound $A_t \leqslant \bar{L}(1 + M)$ for all $t \geqslant 0$.

## B EXTENSION TO STOCHASTIC INTERPOLANTS

In this section, we describe how our approach can be extended to learning the drift field entering in flow matching models (Lipman et al., 2023; Liu et al., 2023), stochastic interpolants (Albergo et al., 2023), and probability flow ODE-based samplers in diffusion models. To this end, we observe that every method can be viewed as building a stochastic interpolant with linear coefficients

$$
\begin{aligned}
x_t &= \alpha(t)x_0 + \beta(t)x_1, \\
\alpha(0) &= 1, \ \ \alpha(1) = 0, \\
\beta(0) &= 0, \ \ \beta(1) = 1,
\end{aligned}
\tag{B.1}
$$

where $x_0 \sim \mathsf{N}(0, I)$ and $x_1 \sim \rho^*$ is drawn from the target data density. A common choice is given by $\alpha(t) = 1 - t$ and $\beta(t) = t$. In this section, we use the convention of interpolants and flow matching models, which exactly interpolate on $[0, 1]$ rather than doing so on $[0, \infty)$ like diffusions. Moreover, we fix the base density at $t = 0$ and the target density at $t = 1$, which is opposite from the convention

---

[7]As an alternative to boundedness, one could also rely on small-ball arguments (Mendelson, 2015). In the interest of keeping our assumptions minimal as possible, we do not pursue this approach.

considered in the main text; see (3.2) for how this construction works in the diffusion model setup. Given (B.1), the drift field entering the probability flow equation is given by

$$b(x, t) = \mathbb{E}\left[\dot{x}_t \mid x_t = x\right], \tag{B.2}$$

while the score field is given by

$$s(x, t) = -\frac{1}{\alpha(t)}\mathbb{E}\left[x_0 \mid x_t = x\right]. \tag{B.3}$$

The following argument is also given in Albergo et al. (2023). We first observe that $\dot{x}_t = \dot{\alpha}(t)x_0 + \dot{\beta}(t)x_1$, and then solve (B.1) for $x_1$ to find that

$$x_1 = \frac{1}{\beta(t)}\left(x_t - \alpha(t)x_0\right). \tag{B.4}$$

Hence, plugging in to (B.2), we find that

$$\begin{aligned}
b(x, t) &= \mathbb{E}\left[\dot{\alpha}(t)x_0 + \dot{\beta}(t)x_1 \mid x_t = x\right], \\
&= \mathbb{E}\left[\dot{\alpha}(t)x_0 + \dot{\beta}(t)\left(\frac{1}{\beta(t)}\left(x_t - \alpha(t)x_0\right)\right) \mid x_t = x\right], \\
&= \mathbb{E}\left[\left(\dot{\alpha}(t) - \frac{\dot{\beta}(t)\alpha(t)}{\beta(t)}\right)x_0 + \frac{\dot{\beta}(t)}{\beta(t)}x_t \mid x_t = x\right], \\
&= \left(\dot{\alpha}(t) - \frac{\dot{\beta}(t)\alpha(t)}{\beta(t)}\right)\mathbb{E}\left[x_0 \mid x_t = x\right] + \frac{\dot{\beta}(t)}{\beta(t)}x, \\
&= \alpha(t)\left(\frac{\dot{\beta}(t)\alpha(t)}{\beta(t)} - \dot{\alpha}(t)\right)s(x, t) + \frac{\dot{\beta}(t)}{\beta(t)}x,
\end{aligned} \tag{B.5}$$

so that $b(x, t)$ can be entirely re-written in terms of the score $s(x, t)$. This shows that the $\mathcal{F}_1$-norm of $b(x, t)$ is the same as the $\mathcal{F}_1$-norm of $s(x, t)$ up to time-scaling and an additive $O(D)$ term coming from the linear term. Hence, similar sample complexity bounds which we derive in Section 3 for learning score models can also be shown for learning the drift field $b(x, t)$.

## C  PRELIMINARY RESULTS

We first make explicit the relation between the score matching error $\mathcal{R}_t$ and the denoising loss $\mathcal{L}_t$. In the sequel, we will make frequent use of Tweedie's formula (see e.g. Efron, 2011):

$$\nabla \log p_t(x_t) = \mathbb{E}\left[\nabla \log q_t(x_t \mid x_0) \mid x_t\right], \quad \nabla \log q_t(x_t \mid x_0) := \frac{m_t x_0 - x_t}{\sigma_t^2}.$$

**Fact C.1** (DSM loss minimizes score). *We have that for all sufficiently regular* $s : \mathbb{R}^D \mapsto \mathbb{R}^D$*:*

$$\mathcal{R}_t(s) = \mathcal{L}_t(s) - C_t, \tag{C.1}$$

*where the offset constant* $C_t$ *is given by:*

$$C_t := \mathbb{E}\operatorname{tr}\operatorname{Cov}\left(\frac{m_t x_0 - x_t}{\sigma_t^2} \mid x_t\right).$$

*Furthermore, we can bound* $C_t$ *by:*

$$C_t \leqslant D/\sigma_t^2.$$

*Proof.* The standard proof for denoising score matching (see e.g. Vincent, 2011, Section 4.2) shows that for all $s$:

$$\mathcal{R}_t(f) = \mathcal{L}_t(f) + \mathbb{E}_{(x_0, x_t)}[\|\nabla \log p_t(x_t)\|^2 - \|\nabla \log q_t(x_t \mid x_0)\|^2].$$

Hence,

$$\mathbb{E}_{(x_0,x_t)}[\|\nabla \log p_t(x_t)\|^2 - \|\nabla \log q_t(x_t \mid x_0)\|^2]$$

$$= \mathbb{E}_{(x_0,x_t)} \left[ \left\| \mathbb{E} \left[ \frac{m_t x_0 - x_t}{\sigma_t^2} \mid x_t \right] \right\|^2 - \left\| \frac{m_t x_0 - x_t}{\sigma_t^2} \right\|^2 \right]$$

$$= -\mathbb{E}\mathbb{E} \left[ \left\| \frac{m_t x_0 - x_t}{\sigma_t^2} \right\|^2 - \left\| \mathbb{E} \left[ \frac{m_t x_0 - x_t}{\sigma_t^2} \mid x_t \right] \right\|^2 \mid x_t \right]$$

$$= -\mathbb{E} \operatorname{tr} \operatorname{Cov} \left( \frac{m_t x_0 - x_t}{\sigma_t^2} \mid x_t \right)$$

$$= -C_t.$$

To bound $C_t$, we observe:

$$C_t = \mathbb{E} \operatorname{tr} \operatorname{Cov} \left( \frac{m_t x_0 - x_t}{\sigma_t^2} \mid x_t \right) = \mathbb{E} \operatorname{tr} \operatorname{Cov} \left( \frac{w}{\sigma_t} \mid x_t \right) \leqslant \mathbb{E}\|w/\sigma_t\|^2 = D/\sigma_t^2.$$

$\square$

We next state a result from Benton et al. (2024) regarding the quality of the samples generated via the exponential integrator scheme (3.4).

**Lemma C.2** (Sampler quality from $L_2$ score bounds (Benton et al., 2024, Theorem 2))**.** *Fix a $T \geqslant 1$ and $\zeta \in (0,1)$. Also fix an $N \in \mathbb{N}_+$ which is even and satisfies $N \geqslant 2\log(1/\zeta)$. Define a sequence of strictly increasing backwards process times $\{\tau_i\}_{i=0}^N$:*

$$\tau_i = \begin{cases} 2(T-1)\frac{i}{N} & \text{if } i \in \{0, \dots, N/2\}, \\ T - \zeta^{2i/N-1} & \text{if } i \in \{N/2+1, \dots, N\}. \end{cases} \tag{C.2}$$

*Let $\gamma_i := \tau_{i+1} - \tau_i$ for $i \in \{0, \dots, N-1\}$. Suppose we have $N$ score functions $\hat{s}_t(x)$ for $t \in \{T - \tau_i\}_{i=0}^{N-1}$ which satisfy:*

$$\sum_{i=0}^{N-1} \gamma_i \mathbb{E}_{p_{T-\tau_i}} \|\hat{s}_{T-\tau_i} - \nabla \log p_{T-\tau_i}\|^2 \leqslant \varepsilon_{\text{score}}^2. \tag{C.3}$$

*Then, we have the following guarantee for the exponential integrator (3.4):*

$$\text{KL}(p_\zeta \parallel \text{Law}(\hat{y}_{T-\zeta})) \lesssim \varepsilon_{\text{score}}^2 + \kappa^2 DN + \kappa(DT + \mu_0^2) + (D + \mu_0^2)e^{-2T},$$

*where $\kappa, \mu_0$ are defined as:*

$$\kappa := \frac{2(T-1) + 4\log(1/\zeta)}{N}, \quad \mu_0 := \mathbb{E}\|x_0\|^2.$$

*Proof.* In order to apply Benton et al. (2024, Theorem 2), we need to compute a $\kappa$ such that:

$$\gamma_k \leqslant \kappa \min\{1, T - t_{k+1}\}, \quad \forall k \in \{0, \dots, N-1\}.$$

To do this, we follow the proof of Benton et al. (2024, Corollary 1). First, for $k \in \{0, \dots, N/2-1\}$, we have that $t_{k+1} \leqslant T - 1$, and hence $T - t_{k+1} \geqslant T - (T-1) = 1$. Hence, we can simply take $\kappa \geqslant 2(T-1)/N$. Now, for $k \in \{N/2, \dots, N-1\}$, we have that $t_{k+1} \geqslant T - 1$, and therefore $T - t_{k+1} \leqslant T - (T-1) = 1$. Hence we need a $\kappa$ such that $\gamma_k \leqslant \kappa(T - t_{k+1})$. We therefore compute:

$$\zeta^{2k/N-1} - \zeta^{2(k+1)/N-1} = \gamma_k \leqslant \kappa(T - t_{k+1}) = \kappa\zeta^{2(k+1)/N-1}.$$

From this we see that $\kappa \geqslant \zeta^{-2/N} - 1$ is required. A sufficient condition takes:

$$\zeta^{-2/N} - 1 = \exp\left(\frac{2}{N}\log(1/\zeta)\right) - 1$$

$$\leqslant 1 + (e-1)\frac{2}{N}\log(1/\zeta) - 1 \qquad \text{since } e^x \leqslant 1 + (e-1)x \text{ for } x \in [0,1]$$

$$\leqslant \frac{4\log(1/\zeta)}{N}.$$

Hence, in total we can set:

$$\kappa = \frac{2(T-1) + 4\log(1/\zeta)}{N}.$$

The result now follows from invoking Benton et al. (2024, Theorem 2). □

Note that an immediate consequence of Lemma C.2 is the following observation: for any $\varepsilon \in (0,1)$, setting

$$T = c_0 \log\left(\frac{\sqrt{D} \vee \mu_0}{\varepsilon}\right), \quad N = 2\left\lceil c_1 \frac{D \vee \mu_0^2}{\varepsilon^2}\left[\log^2\left(\frac{\sqrt{D} \vee \mu_0}{\varepsilon}\right) + \log^2\left(\frac{1}{\zeta}\right)\right]\right\rceil,$$

for some universal positive constants $c_0, c_1$, we have that

$$\mathrm{KL}(p_\zeta \,\|\, \mathrm{Law}(\hat{y}_{T-\zeta})) \lesssim \varepsilon_{\mathrm{score}}^2 + \varepsilon^2.$$

Next, we state a technical lemma regarding a perturbation result for Gaussian mollifications at small scale.

**Lemma C.3** (Score function perturbation (Lee et al., 2023, Lemma C.12))**.** *Suppose that $p(x)$ is a density on $\mathbb{R}^d$ such that $\nabla \log p(x)$ is $L$-Lipschitz on $\mathbb{R}^d$. For $\alpha \geqslant 1$, define the corresponding density $p_\alpha(x) := \alpha^d p(\alpha x)$. Let $\gamma_{\sigma^2}$ denote an $\mathsf{N}(0, \sigma^2 I_d)$ distribution. If $L \leqslant 1/(2\alpha^2\sigma^2)$, then we have the following bound for all $x \in \mathbb{R}^d$:*

$$\|\nabla \log p(x) - \nabla \log p_\alpha * \gamma_{\sigma^2}(x)\|$$
$$\leqslant 6\alpha^2 L\sigma\sqrt{d} + (\alpha + 2\alpha^3 L\sigma^2)(\alpha-1)L\|x\| + (\alpha - 1 + 2\alpha^3 L\sigma^2)\|\nabla \log p(x)\|.$$

The previous mollification lemma is next used to bound the score functions uniformly over time.

**Proposition C.4.** *Consider a forward diffusion process $z_t \stackrel{\mathsf{d}}{=} m_t z_0 + \sigma_t w$ on $\mathbb{R}^d$. Let $\pi_t$ denote the marginal distribution of $z_t$ for all $t \geqslant 0$. Suppose that $\nabla \log \pi_t$ is $L$-Lipschitz (for $L \geqslant 1$) on $\mathbb{R}^d$ for all $t \geqslant 0$. Defining $\bar{L} := cL(\mathbb{E}\|z_0\| + \sqrt{d} + \|\nabla \log \pi_0(0)\|)$ where $c \geqslant 1$ is an universal constant, we have that for all $t \geqslant 0$ and $z \in \mathbb{R}^d$,*

$$\|\nabla \log \pi_t(z)\| \leqslant \bar{L}(1 + \|z\|).$$

*Proof.* Define $\bar{\pi}_t(z) := m_t^{-d}\pi_0(m_t^{-1}z)$, which is the density of the random variable $m_t z_0$. Hence, we have that $\pi_t = \bar{\pi}_t * \gamma_{\sigma_t^2}$. By Lemma C.3, whenever $L \leqslant m_t^2/(2\sigma_t^2)$, we have that:

$$\|\nabla \log \pi_t(0)\| \leqslant 6m_t^{-2}L\sigma_t\sqrt{d} + (m_t^{-1} + 2m_t^{-3}L\sigma_t^2)\|\nabla \log \pi_0(0)\|.$$

We now compute the range of $t$'s for which $L \leqslant m_t^2/(2\sigma_t^2)$ holds. Using the specific form of $m_t, \sigma_t$,

$$2L \leqslant \frac{m_t^2}{\sigma_t^2} = \frac{\exp(-2t)}{1 - \exp(-2t)} = \frac{1}{\exp(2t) - 1} \iff t \leqslant \frac{1}{2}\log\left(1 + \frac{1}{2L}\right) =: t_\star.$$

First, note that since we assume $L \geqslant 1$, then we have that $t_\star$ is bounded by a universal constant. Hence, for $t \leqslant t_\star$,

$$\|\nabla \log \pi_t(0)\| \lesssim L(\sqrt{d} + \|\nabla \log \pi_0(0)\|).$$

Hence for any $z$,

$$\|\nabla \log \pi_t(z)\| \leqslant L\|z\| + \|\nabla \log \pi_t(0)\|$$
$$\leqslant L\|z\| + cL(\sqrt{d} + \|\nabla \log \pi_0(0)\|)$$
$$\leqslant cL(\sqrt{d} + \|\nabla \log \pi_0(0)\|)(1 + \|z\|).$$

On the other hand, when $t \geqslant t_\star$, more work is needed. When $t \geqslant t_\star$, we first bound:

$$\sigma_t^{-1} \leqslant \sigma_{t_\star}^{-1} = \sqrt{2L + 1}.$$

Define two events:

$$\mathcal{E}_1 := \left\{ \mathbb{E}[\|w\|/\sigma_t \mid z_t] \geqslant 4\sqrt{d}/\sigma_t \right\}, \quad \mathcal{E}_2 := \left\{ \|z_t\| \geqslant 4(\mathbb{E}\|z_0\| + \sigma_t\sqrt{d}) \right\}$$

By Markov's inequality we have that:

$$\mathbb{P}_{z_t} \{\mathcal{E}_1\} \leqslant 1/4, \quad \mathbb{P}_{z_t} \{\mathcal{E}_2\} \leqslant 1/4.$$

Now suppose that $\mathcal{E}_1^c \subseteq \mathcal{E}_2$. Then, we have a contradiction, since:

$$1 - 1/4 \leqslant \mathbb{P}_{z_t}\{\mathcal{E}_1^c\} \leqslant \mathbb{P}_{z_t}\{\mathcal{E}_2\} \leqslant 1/4.$$

Hence, there must exists an $\omega \in \mathcal{E}_1^c$ such that $\omega \notin \mathcal{E}_2$. Hence, there exists a $\bar{z}_t$ satisfying:

$$\|\nabla \log \pi_t(\bar{z}_t)\| \leqslant \mathbb{E}[\|w\|/\sigma_t \mid z_t = \bar{z}_t] \leqslant 4\sqrt{d}/\sigma_t, \quad \|\bar{z}_t\| \leqslant 4(\mathbb{E}\|z_0\| + \sigma_t\sqrt{d}).$$

Hence for any $z$,

$$\begin{aligned}
\|\nabla \log \pi_t(z)\| &\leqslant L\|z - \bar{z}_t\| + \|\nabla \log \pi_t(\bar{z}_t)\| \\
&\leqslant L\|z\| + 4L(\mathbb{E}\|z_0\| + \sigma_t\sqrt{d}) + 4\sqrt{d}/\sigma_t \\
&\leqslant L\|z\| + 4L(\mathbb{E}\|z_0\| + \sqrt{d}) + 4\sqrt{(2L+1)d} \\
&\leqslant c'L(\mathbb{E}\|z_0\| + \sqrt{d})(1 + \|z\|).
\end{aligned}$$

$\square$

Next, we state a $L_\infty$ approximation result for scalar-valued Lipschitz function from Bach (2017).

**Lemma C.5** ($L_\infty$ approximation of scalar Lipschitz functions (Bach, 2017, Proposition 6))**.** *Let* $f : \mathbb{R}^d \mapsto \mathbb{R}$. *Suppose that* $f$ *is* $L$-*Lipschitz and* $B$-*bounded on* $B_2(d, M)$. *Set* $K := B \vee LM$. *For any* $\gamma \geqslant O_d(1) \cdot K$ *there exists an* $f_\gamma \in \mathcal{F}_1$ *such that* $\|f_\gamma\|_{\mathcal{F}_1} \leqslant \gamma$ *and:*

$$\sup_{x \in B_2(d,M)} |f(x) - f_\gamma(x)| \leqslant O_d(1)K \left( \frac{K}{\gamma} \right)^{2/(d+1)} \log \left( \frac{\gamma}{K} \right).$$

Our next result is a simple technical fact which we will utilize in our truncation analysis.

**Proposition C.6.** *Let* $S, \check{S}$ *be two* $\mathsf{X}$-*valued random variables over the same probability space. Let* $f : \mathsf{X} \mapsto \mathbb{R}$ *be a measurable function. We have:*

$$\mathbb{E}[f(S)] \leqslant \mathbb{E}[f(\check{S})] + \left( \sqrt{\mathbb{E}[f^2(S)]} + \sqrt{\mathbb{E}[f^2(\check{S})]} \right) \sqrt{\mathbb{P}\{S \neq \check{S}\}}.$$

*Note if* $f$ *is non-negative, then we have the simpler bound:*

$$\mathbb{E}[f(S)] \leqslant \mathbb{E}[f(\check{S})] + \sqrt{\mathbb{E}[f^2(S)]\mathbb{P}\{S \neq \check{S}\}}.$$

*Proof.* We have:

$$\begin{aligned}
\mathbb{E}[f(S)] &= \mathbb{E}[f(S)\mathbf{1}\{S = \check{S}\}] + \mathbb{E}[f(S)\mathbf{1}\{S \neq \check{S}\}] \\
&= \mathbb{E}[f(\check{S})\mathbf{1}\{S = \check{S}\}] + \mathbb{E}[f(S)\mathbf{1}\{S \neq \check{S}\}] \\
&= \mathbb{E}[f(\check{S})] + \mathbb{E}[f(S)\mathbf{1}\{S \neq \check{S}\}] - \mathbb{E}[f(\check{S})\mathbf{1}\{S \neq \check{S}\}] \\
&\leqslant \mathbb{E}[f(\check{S})] + \sqrt{\mathbb{E}[f^2(S)]\mathbb{P}\{S \neq \check{S}\}} + \sqrt{\mathbb{E}[f^2(\check{S})]\mathbb{P}\{S \neq \check{S}\}}.
\end{aligned}$$

$\square$

The next result is a simple algebraic fact which will be useful for solving for implicit inequalities involving logarithms.

**Proposition C.7** (Log dominance rule, (see e.g. Du et al., 2021, Lemma F.2))**.** *Let* $a, b, \nu$ *be positive scalars. Put* $\bar{\nu} := (1 + \nu)^\nu$. *Then,*

$$m \geqslant \bar{\nu} a \log^\nu(\bar{\nu}ab) \implies m \geqslant a \log^\nu(bm).$$

Next, we have an intermediate result to bound the Rademacher complexity of $\mathcal{F}_1$-norm bounded functions.

**Proposition C.8.** *Let $\|x_i\| \leqslant 1$ for $i \in [n]$. We have:*

$$\mathbb{E}_{\{\varepsilon_i\}} \sup_{u,v \in B_2(D)} \left| \frac{1}{n} \sum_{i=1}^n \langle u, \varepsilon_i \rangle \sigma(\langle v, x \rangle) \right| \leqslant c\sqrt{\frac{D}{n}},$$

*where the $\varepsilon_i \in \{\pm 1\}^d$ are independent Rademacher random vectors[8] and $c > 0$ is a universal constant.*

*Proof.* Define $X_{u,v} := \frac{1}{n} \sum_{i=1}^n \langle u, \varepsilon_i \rangle \sigma(\langle v, x \rangle)$. Observe that for $u_i, v_i \in B_2(D)$ for $i \in \{1, 2\}$,

$$X_{u_1,v_1} - X_{u_2,v_2} = \frac{1}{n} \sum_{i=1}^n [\sigma(\langle v_1, x_i \rangle) - \sigma(\langle v_2, x_i \rangle)] \langle u_1, \varepsilon_i \rangle + \frac{1}{n} \sum_{i=1}^n \sigma(\langle v_2, x_i \rangle) \langle u_1 - u_2, \varepsilon_i \rangle$$

$$=: T_1 + T_2.$$

First, we recall that a Rademacher random variable is 1-sub-Gaussian, and therefore $\langle u_i, \varepsilon_i \rangle$ is also 1-sub-Gaussian since $\|u_i\| \leqslant 1$. Using the fact that ReLU is 1-Lipschitz followed by Cauchy-Schwarz and the assumption that $\|x_i\| \leqslant 1$,

$$|\sigma(\langle v_1, x_i \rangle) - \sigma(\langle v_2, x_i \rangle)| \leqslant |\langle v_1 - v_2, x_i \rangle| \leqslant \|v_1 - v_2\|.$$

Hence, $[\sigma(\langle v_1, x_i \rangle) - \sigma(\langle v_2, x_i \rangle)] \langle u_1, \varepsilon_i \rangle$ is $\|v_1 - v_2\|$-sub-Gaussian. Consequently, $T_1$ is $\|v_1 - v_2\|/\sqrt{n}$-sub-Gaussian. Similarly, since $|\sigma(\langle v_2, x_i \rangle)| \leqslant 1$, we also have that $T_2$ is $\|u_1 - u_2\|/\sqrt{n}$-sub-Gaussian. Hence, the sum $T_1 + T_2$ is sub-Gaussian with constant:

$$\sqrt{2(\|v_1 - v_2\|^2 + \|u_1 - u_2\|^2)}/\sqrt{n}.$$

Letting $\omega = (u, v)$, we consider the following metric on $\Omega := B_2(D) \times B_2(D)$:

$$d((u_1, v_1), (u_2, v_2)) = \sqrt{\|u_1 - u_2\|^2 + \|v_1 - v_2\|^2}.$$

Hence for any $\omega_1, \omega_2 \in \Omega$, the difference $X_{\omega_1} - X_{\omega_2}$ is $\sqrt{2/n} \cdot d(\omega_1, \omega_2)$-sub-Gaussian. Therefore we can use Dudley's inequality (see e.g. Vershynin, 2018, Chapter 8) to bound:

$$\mathbb{E} \sup_{\omega \in \Omega} X_\omega \leqslant cn^{-1/2} \int_0^\infty \sqrt{\log N(\varepsilon; \Omega, d)} \, d\varepsilon = cn^{-1/2} \int_0^{\sqrt{2}} \sqrt{\log N(\varepsilon; \Omega, d)} \, d\varepsilon.$$

Next, fix an $\varepsilon > 0$ and let $\{u_i\}, \{v_i\}$ be $\varepsilon/\sqrt{2}$-covers of $B_2(D)$. Let $[u]$ (resp. $[v]$) denote the closest point in the cover to $u$ (resp. $v$). Given $(u, v) \in \Omega$, we have:

$$d((u, v), ([u], [v])) = \sqrt{\|u - [u]\|^2 + \|v - [v]^2\|} \leqslant \varepsilon.$$

Using the standard volume estimate of the covering of $B_2(D)$ (see e.g. Vershynin, 2018, Chapter 4),

$$\log N(\varepsilon; \Omega, d) \leqslant 2D \log(1 + 2\sqrt{2}/\varepsilon).$$

Consequently,

$$\mathbb{E} \sup_{\omega \in \Omega} X_w \leqslant cn^{-1/2}\sqrt{2D} \int_0^{\sqrt{2}} \sqrt{\log(1 + 2\sqrt{2}/\varepsilon)} \, d\varepsilon = c'\sqrt{D/n}.$$

$\square$

Our final preliminary result translates the previous bound Proposition C.8 to a bound on the Rademacher complexity of $\mathcal{F}_1$-norm balls.

**Proposition C.9.** *Let $\mathscr{F} = \{s : \mathbb{R}^d \mapsto \mathbb{R}^d \mid \|s\|_{\mathcal{F}_1} \leqslant R\}$. For any $\check{x}_i \in B_2(D, M)$, $i \in [n]$, we have:*

$$\mathbb{E}_{\{\varepsilon_i\}} \sup_{f \in \mathscr{F}} n^{-1} \left| \sum_{i=1}^n \langle \varepsilon_i, f(\check{x}_i) \rangle \right| \leqslant cRM\sqrt{\frac{D}{n}},$$

*where the $\varepsilon_i \in \{\pm 1\}^d$ are independent Rademacher random vectors and $c > 0$ is a universal constant.*

---

[8]That is, each coordinate of $\varepsilon_i \in \{\pm 1\}^d$ is an independent Rademacher random variable.

*Proof.* For any $\{\varepsilon_i\}$ and $f \in \mathscr{F}$, observe that by the definition of $\mathcal{F}_1$:

$$n^{-1}\left|\sum_{i=1}^n \langle \varepsilon_i, f(\check{x}_i)\rangle\right| = n^{-1}\left|\sum_{i=1}^n \left\langle \varepsilon_i, \int u\sigma(\langle v, \check{x}_i\rangle)\,\mathrm{d}\mu(u,v)\right\rangle\right|$$

$$= \left|\int \left[\frac{1}{n}\sum_{i=1}^n \langle \varepsilon_i, u\rangle \sigma(\langle v, \check{x}_i\rangle)\right]\mathrm{d}\mu(u,v)\right|$$

$$\leqslant R \sup_{u,v\in\mathbb{S}^{D-1}} \left|\frac{1}{n}\sum_{i=1}^n \langle \varepsilon_i, u\rangle \sigma(\langle v, \check{x}_i\rangle)\right|.$$

Hence,

$$\mathbb{E}_{\{\varepsilon_i\}} \sup_{f\in\mathscr{F}} n^{-1}\left|\sum_{i=1}^n \langle \varepsilon_i, f(\check{x}_i)\rangle\right| \leqslant R \cdot \mathbb{E}_{\{\varepsilon_i\}} \sup_{u,v\in\mathbb{S}^{D-1}} \left|\frac{1}{n}\sum_{i=1}^n \langle \varepsilon_i, u\rangle \sigma(\langle v, \check{x}_i\rangle)\right|,$$

from which the claim follows by Proposition C.8 and using homogeneity of ReLU to scale the data points $\check{x}_i$ to $B_2(D)$. □

## D $\mathcal{F}_1$ APPROXIMATION THEORY FOR LIPSCHITZ CONTINUOUS FUNCTIONS

Here we develop the necessary results to establish that $\mathcal{F}_1$ functions can approximate structured score functions in an efficient way. Our first result is a preliminary result that allows us to translate $L_\infty$ approximation bounds to $L_2$ bounds. For what follows, let the notation $\mathcal{P}(\mathsf{X})$ denote the set of subsets of $\mathsf{X}$.

**Proposition D.1.** *Let $M : (0,1) \mapsto \mathcal{P}(\mathbb{R}^D)$ be such that:*

$$\forall\, \delta \in (0,1), \quad \mathbb{P}_{x_t\sim p_t}\{x_t \in M(\delta)\} \geqslant 1-\delta.$$

*Suppose that $R(\varepsilon,\delta)$ satisfies the following condition: for any positive $\varepsilon > 0$ and $\delta \in (0,1)$ there exists a function $\hat{s} : \mathbb{R}^D \mapsto \mathbb{R}^D$ such that*

$$\|\hat{s}\|_{\mathcal{F}_1} \leqslant R(\varepsilon,\delta), \qquad \sup_{x\in M(\delta)} \|\hat{s}(x) - \nabla \log p_t(x)\| \leqslant \varepsilon. \tag{D.1}$$

*Suppose there exists a $\delta \in (0,1)$ satisfying:*

$$R^4(\varepsilon/2,\delta)\cdot\delta \leqslant c_0\varepsilon^4/\mathbb{E}\|x_t\|^4, \quad \delta \leqslant c_1(\varepsilon\sigma_t)^4/D^2. \tag{D.2}$$

*Above, both $c_0, c_1$ are universal positive constants. Then, there exists an $\hat{s} : \mathbb{R}^D \mapsto \mathbb{R}^D$ such that:*

$$\|\hat{s}\|_{\mathcal{F}_1} \leqslant R(\varepsilon/2,\delta), \quad \|\hat{s} - \nabla \log p_t\|_{L_2(p_t)} \leqslant \varepsilon. \tag{D.3}$$

*Proof.* Let $\mathcal{E}_G := \{x_t \in M(\delta)\}$. By assumption, we have that $\mathbb{P}(\mathcal{E}_G) \geqslant 1-\delta$. Put $s_\star := \nabla \log p_t$ and let $\hat{s} : \mathbb{R}^D \mapsto \mathbb{R}^D$ be as guaranteed by the assumption such that:

$$\|\hat{s}\|_{\mathcal{F}_1} \leqslant R(\varepsilon/2,\delta), \qquad \sup_{x\in M(\delta)} \|\hat{s}(x) - s_\star(x)\| \leqslant \varepsilon/2.$$

Hence,

$$\mathbb{E}_{x_t}\|\hat{s} - s_\star\|^2 = \mathbb{E}_{x_t}\|\hat{s} - s_\star\|^2\mathbf{1}\{\mathcal{E}_G\} + \mathbb{E}_{x_t}\|\hat{s} - s_\star\|^2\mathbf{1}\{\mathcal{E}_G^c\}$$

$$\leqslant \sup_{x\in M(\delta)}\|\hat{s}(x) - s_\star(x)\|^2 + \sqrt{\mathbb{E}_{x_t}\|\hat{s} - s_\star\|^4}\cdot\sqrt{\delta}$$

$$\leqslant (\varepsilon/2)^2 + \sqrt{\mathbb{E}_{x_t}\|\hat{s} - s_\star\|^4}\cdot\sqrt{\delta}.$$

Consequently, by taking square root of both sides:

$$\|\hat{s} - s_\star\|_{L_2(p_t)} \leqslant \varepsilon/2 + \|\hat{s} - s_\star\|_{L_4(p_t)}\cdot\delta^{1/4}.$$

Let us now control $\|\hat{s} - s_\star\|_{L_4(p_t)}$. By triangle inequality and $\|\hat{s}(x)\| \leqslant \|\hat{s}\|_{\mathcal{F}_1}\|x\|$ for all $x$:

$$\|\hat{s} - s_\star\|_{L_4(p_t)} \leqslant \|\hat{s}\|_{L_4(p_t)} + \|s_\star\|_{L_4(p_t)}$$
$$\leqslant \|\hat{s}\|_{\mathcal{F}_1}\|x_t\|_{L_4(p_t)} + \|s_\star\|_{L_4(p_t)}$$
$$\leqslant R(\varepsilon/2, \delta)\|x_t\|_{L_4(p_t)} + \|s_\star\|_{L_4(p_t)}.$$

To control $\|s_\star\|_{L_4(p_t)}$, we use Tweedie's formula:

$$\nabla \log p_t(x_t) = \mathbb{E}\left[\frac{m_t x_0 - x_t}{\sigma_t^2} \,\Big|\, x_t\right] = -\frac{\mathbb{E}[w \mid x_t]}{\sigma_t}.$$

Hence by Jensen's inequality and the tower property,

$$\|s_\star\|_{L_4(p_t)}^4 = \mathbb{E}\|\nabla \log p_t(x_t)\|^4 = \frac{1}{\sigma_t^4}\mathbb{E}\|\mathbb{E}[w \mid x_t]\|^4 \leqslant \frac{1}{\sigma_t^4}\mathbb{E}\|w\|^4 \leqslant \frac{3D^2}{\sigma_t^4}.$$

Combining these calculations,

$$\|\hat{s} - s_\star\|_{L_4(p_t)} \leqslant R(\varepsilon/2, \delta)\|x_t\|_{L_4(p_t)} + \frac{3^{1/4}\sqrt{D}}{\sigma_t}.$$

Hence,

$$\|\hat{s} - s_\star\|_{L_2(p_t)} \leqslant \varepsilon/2 + \left[R(\varepsilon/2, \delta)\|x_t\|_{L_4(p_t)} + \frac{3^{1/4}\sqrt{D}}{\sigma_t}\right] \cdot \delta^{1/4}$$

Hence, if we set $\delta$ such that:

$$R(\varepsilon/2, \delta)\|x_t\|_{L_4(p_t)} \cdot \delta^{1/4} \leqslant \varepsilon/4, \qquad \frac{3^{1/4}\sqrt{D}}{\sigma_t} \cdot \delta^{1/4} \leqslant \varepsilon/4,$$

then we conclude that $\|\hat{s} - s_\star\|_{L_2(p_t)} \leqslant \varepsilon$. $\qquad\square$

We next turn to our main $L_\infty$ approximation result for Lipschitz functions. We proceed in two steps. First, we extend Lemma C.5 to vector-valued Lipschitz functions in a straightforward way. Then, we use the log dominance rule to invert the result. For the first step, we have the following result.

**Proposition D.2.** *Let $f : \mathbb{R}^d \mapsto \mathbb{R}^d$. Suppose that $f$ is $L$-Lipschitz and $B$-bounded on $B_2(d, M)$. Set $K_d := d \cdot (B \vee LM)$. For any $\gamma \geqslant O_d(1) \cdot K_d$, there exists an $f_\gamma \in \mathcal{F}_1$ such that $\|f_\gamma\|_{\mathcal{F}_1} \leqslant \gamma$ and:*

$$\sup_{x \in B_2(d,M)} \|f(x) - f_\gamma(x)\| \leqslant O_d(1)K_d\left(\frac{K_d}{\gamma}\right)^{2/(d+1)} \log\left(\frac{\gamma}{K_d}\right). \tag{D.4}$$

*Proof.* For $i \in [d]$, let $f_i(x) := \langle e_i, f(x)\rangle$, where $e_i \in \mathbb{R}^d$ is the $i$-th standard basis vector. We will apply Lemma C.5 to each of the $f_i$'s. Note that each $f_i$ is also $L$-Lipschitz and $B$-bounded on $B_2(d, M)$. Hence, for every $i \in [d]$ there exists an $f_{\gamma,i} \in \mathcal{F}_1$ with $\|f_{\gamma,i}\|_{\mathcal{F}_1} \leqslant \gamma/d$ and:

$$\sup_{x \in B_2(d,M)} |f_i(x) - f_{\gamma,i}(x)| \leqslant O_d(1)\frac{K_d}{d}\left(\frac{K_d}{\gamma}\right)^{2/(d+1)} \log\left(\frac{\gamma}{K_d}\right) =: \zeta.$$

Choosing $f_\gamma := (f_{\gamma,1}, \ldots, f_{\gamma,d})$ yields:

$$\sup_{x \in B_2(d,M)} \|f(x) - f_\gamma(x)\| = \sup_{x \in B_2(d,M)} \sqrt{\sum_{i=1}^{d} |f_i(x) - f_{\gamma,i}(x)|^2}$$
$$\leqslant \sqrt{\sum_{i=1}^{d} \sup_{x \in B_2(d,M)} |f_i(x) - f_{\gamma,i}(x)|^2} \leqslant \sqrt{d}\zeta.$$

To finish the claim, we bound the $\mathcal{F}_1$ norm of $f_\gamma$. Since

$$f_\gamma(x) = \sum_{i=1}^{d} e_i f_{\gamma,i}(x),$$

by triangle inequality $\|f_\gamma\|_{\mathcal{F}_1} \leqslant \sum_{i=1}^{d} \|f_{\gamma,i}\|_{\mathcal{F}_1} \leqslant d \cdot (\gamma/d) = \gamma$. $\qquad\square$

We now execute the second step, where we invert the RHS of Proposition D.2 and solve for $\gamma$.

**Lemma A.4.** *Let $f : \mathbb{R}^d \to \mathbb{R}^d$ be $L$-Lipschitz and $B$-bounded on $B_2(d, M)$. Define $K := B \vee LM$. For any $\varepsilon \in (0, K/2)$, there exists an $f_\varepsilon \in \mathcal{F}_1$ such that $\sup_{x \in B_2(d, M)} \|f(x) - f_\varepsilon(x)\| \leqslant \varepsilon$ and:*

$$\|f_\varepsilon\|_{\mathcal{F}_1} \leqslant O_d(1) K \left( \frac{K}{\varepsilon} \right)^{(d+1)/2} \log^{(d+1)/2} \left( \frac{K}{\varepsilon} \right).$$

*Proof.* Setting the RHS of (D.4) from Proposition D.2 to $\varepsilon$ and rearranging terms, we need the following condition to hold (recall $K_d := d \cdot K$):

$$\frac{\gamma}{K_d} \geqslant \left( \frac{O_d(1) K_d}{\varepsilon} \right)^{(d+1)/2} \log^{(d+1)/2} \left( \frac{\gamma}{K_d} \right).$$

Using Proposition C.7, a sufficient condition is:

$$\frac{\gamma}{K_d} \geqslant O_d(1) \left( \frac{O_d(1) K_d}{\varepsilon} \right)^{(d+1)/2} \log^{(d+1)/2} \left( O_d(1) \left( \frac{O_d(1) K_d}{\varepsilon} \right)^{(d+1)/2} \right).$$

The claim now follows by simplifying these expressions with our assumptions. $\square$

# E UNIFORM CONVERGENCE FOR THE DSM LOSS

Our first step is to establish the claimed basic inequality (A.1).

**Proposition E.1** (Basic generalization inequality). *Let $\mathscr{F}$ be any set of functions mapping $\mathbb{R}^D \mapsto \mathbb{R}^D$. Let $\hat{f}_t \in \mathscr{F}$ denote the DSM empirical risk minimizer:*

$$\hat{f}_t \in \arg \min_{f \in \mathscr{F}} \hat{\mathcal{L}}_t(f).$$

*Then, we have for any $\varepsilon \geqslant 0$:*

$$\mathbb{E}_{\mathcal{D}_t}[\mathcal{R}_t(\hat{f}_t)] \leqslant (1 + \varepsilon) \inf_{f \in \mathscr{F}} \mathcal{R}_t(f) + \mathbb{E}_{\mathcal{D}_t} \sup_{f \in \mathscr{F}} [\mathcal{L}_t(f) - (1 + \varepsilon)\hat{\mathcal{L}}_t(f)] + \varepsilon \cdot C_t.$$

*Proof.* For any $\varepsilon \geqslant 0$ and any $f \in \mathscr{F}$,

$$
\begin{aligned}
\mathbb{E}_{\mathcal{D}_t}[\mathcal{R}_t(\hat{f}_t)] &= \mathbb{E}_{\mathcal{D}_t}[\mathcal{L}_t(\hat{f}_t) - C_t] && \text{using Fact C.1} \\
&= \mathbb{E}_{\mathcal{D}_t}[\mathcal{L}_t(\hat{f}_t) - (1 + \varepsilon)\hat{\mathcal{L}}_t(\hat{f}_t) + (1 + \varepsilon)\hat{\mathcal{L}}_t(\hat{f}_t) - C_t] && \\
&\leqslant \mathbb{E}_{\mathcal{D}_t}[\mathcal{L}_t(\hat{f}_t) - (1 + \varepsilon)\hat{\mathcal{L}}_t(\hat{f}_t) + (1 + \varepsilon)\hat{\mathcal{L}}_t(f) - C_t] && \text{since } \hat{f}_t \text{ is an ERM} \\
&= \mathbb{E}_{\mathcal{D}_t}[\mathcal{L}_t(\hat{f}_t) - (1 + \varepsilon)\hat{\mathcal{L}}_t(\hat{f}_t)] + (1 + \varepsilon)\mathcal{L}_t(f) - C_t && \text{since } \mathbb{E}_{\mathcal{D}_t}[\hat{\mathcal{L}}_t(f)] = \mathcal{L}_t(f) \\
&= \mathbb{E}_{\mathcal{D}_t}[\mathcal{L}_t(\hat{f}_t) - (1 + \varepsilon)\hat{\mathcal{L}}_t(\hat{f}_t)] + (1 + \varepsilon)\mathcal{R}_t(f) + \varepsilon \cdot C_t && \text{using Fact C.1} \\
&\leqslant \mathbb{E}_{\mathcal{D}_t} \sup_{f \in \mathscr{F}} [\mathcal{L}_t(f) - (1 + \varepsilon)\hat{\mathcal{L}}_t(f)] + (1 + \varepsilon)\mathcal{R}_t(f) + \varepsilon \cdot C_t.
\end{aligned}
$$

The claim now follows by taking the infimum of the RHS over $f \in \mathscr{F}$. $\square$

The rest of this section will focus on the uniform convergence term in the basic inequality (A.1). We first define some notation which we will use in our analysis. Let $\nu_\delta(\check{x}_0, \check{x}_t)$ denote a distribution over pairs of *truncated* vectors, parameterized by $\delta \in (0, 1)$, defined as follows:

$$\nu_\delta := \text{Law}((x_0, x_t) \cdot \mathbf{1}\{\mathcal{E}_x(\delta)\}), \quad \mathbb{P}\{\mathcal{E}_x(\delta)\} \geqslant 1 - \delta. \tag{E.1}$$

Note that in the above definition, the event $\mathcal{E}_x(\delta)$ lives in the *joint* probability space of $(x_0, x_t)$. The specifics of the event $\mathcal{E}_x(\delta)$ are left unspecified for now, as they depend on the underlying details of our latent structure. However, we will require the following properties to hold almost surely for some $\check{\mu}_{t,x}(\delta)$ and $\check{\mu}_{t,q}(\delta)$:

$$(\check{x}_0, \check{x}_t) \sim \nu_\delta \implies \|\check{x}_t\| \leqslant \check{\mu}_{t,x}(\delta) \text{ and } \|\nabla \log q_t(\check{x}_t \mid \check{x}_0)\| \leqslant \check{\mu}_{t,q}(\delta). \tag{E.2}$$

Next, define the population denoising loss over $\nu_\delta$ as:

$$\check{\mathcal{L}}_t(f;\delta) := \mathbb{E}_{(\check{x}_0,\check{x}_t)\sim\nu_\delta}\|f(\check{x}_t) - \nabla\log q_t(\check{x}_t \mid \check{x}_0)\|^2. \tag{E.3}$$

Furthermore, given a dataset $\bar{\mathcal{D}}_t = \{(\bar{x}_0^i, \bar{x}_t^i)\}_{i=1}^n$, the *generalized empirical loss* is defined as

$$\hat{\mathcal{L}}_t(f;\bar{\mathcal{D}}_t) := \frac{1}{n}\sum_{i=1}^n\|f(\bar{x}_t^i) - \nabla\log q_t(\bar{x}_t^i \mid \bar{x}_0^i)\|^2. \tag{E.4}$$

Note that the above definitions are used only in our truncation argument, and do not appear in the actual learning procedure.

The main result of this section is the following bound on the uniform convergence term.

**Lemma E.2.** *For $R_t \geq 1$, define $\mathscr{F}_t := \{s : \mathbb{R}^D \mapsto \mathbb{R}^D \mid \|s\|_{\mathcal{F}_1} \leq R_t\}$. For $\varepsilon \in (0,1]$, we have:*

$$\mathbb{E}\sup_{f\in\mathscr{F}_t}[\mathcal{L}_t(f) - (1+\varepsilon)\hat{\mathcal{L}}_t(f)]$$

$$\leq \tilde{O}(1)(1+\varepsilon^{-1})\left[\frac{R_t^2\check{\mu}_{t,x}^2(n^{-5})D + \check{\mu}_{t,q}^2(n^{-5})}{n}\right] + O(1)\frac{R_t^2\|x_t\|_{L_4(p_t)}^2 + D/\sigma_t^2}{n^2}.$$

The proof of Lemma E.2 follows immediately from the following two results (invoking them both with $\delta = n^{-4}$). The first result applies a truncation argument so that it suffices to prove uniform convergence over truncated data.

**Proposition E.3.** *Fix a $\delta \in (0,1)$. Define the truncated random pair $(\check{x}_0, \check{x}_t) \sim \nu_{\delta/n}$ (cf. (E.1)). Let the truncated dataset $\check{\mathcal{D}}_t := \{(\check{x}_0^i, \check{x}_t^i)\}_{i=1}^n$ be $n$ iid copies of $(\check{x}_0, \check{x}_t)$, i.e., $\check{\mathcal{D}}_t \sim \nu_{\delta/n}^{\otimes n}$. For some $R_t \geq 1$, define $\mathscr{F}_t := \{s : \mathbb{R}^D \mapsto \mathbb{R}^D \mid \|s\|_{\mathcal{F}_1} \leq R_t\}$. For all $\varepsilon \in [0,1]$, we have that:*

$$\mathbb{E}\sup_{f\in\mathscr{F}_t}[\mathcal{L}_t(f) - (1+\varepsilon)\hat{\mathcal{L}}_t(f)]$$

$$\leq \mathbb{E}\sup_{f\in\mathscr{F}_t}[\check{\mathcal{L}}_t(f;\delta/n) - (1+\varepsilon)\hat{\mathcal{L}}_t(f;\check{\mathcal{D}}_t)] + c(R_t^2\|x_t\|_{L_4(p_t)}^2 + \sigma_t^{-2}D)\cdot\delta^{1/2},$$

*where $c > 0$ is a universal constant.*

*Proof.* Let $\mathcal{E}_G$ denote the event $\mathcal{E}_G := \{\mathcal{D}_t = \check{\mathcal{D}}_t\}$. By a union bound, $\mathbb{P}(\mathcal{E}_G) \geq 1 - \delta$. Next we define for a dataset $\bar{\mathcal{D}}_t$ the random variable:

$$\psi(\bar{\mathcal{D}}_t) := \sup_{f\in\mathscr{F}_t}[\mathcal{L}_t(f) - (1+\varepsilon)\hat{\mathcal{L}}_t(f;\bar{\mathcal{D}}_t)]. \tag{E.5}$$

Applying Proposition C.6:

$$\mathbb{E}[\psi(\mathcal{D}_t)] \leq \mathbb{E}[\psi(\check{\mathcal{D}}_n)] + (\sqrt{\mathbb{E}[\psi^2(\mathcal{D}_t)]} + \sqrt{\mathbb{E}[\psi^2(\check{\mathcal{D}}_n)]})\cdot\delta^{1/2}. \tag{E.6}$$

We next need to upper bound both:

$$\mathbb{E}[\psi^2(\mathcal{D}_t)], \quad \mathbb{E}[\psi^2(\check{\mathcal{D}}_n)].$$

To do this, we first derive a few intermediate bounds. We start with:

$$\begin{aligned}
\mathcal{L}_t(f) &= \mathbb{E}_{(x_0,x_t)}\|f(x_t) - \nabla\log q_t(x_t \mid x_0)\|^2 \\
&\leq 2\mathbb{E}\|f(x_t)\|^2 + 2\mathbb{E}\|\nabla\log q_t(x_t \mid x_0)\|^2 && \text{since } (a+b)^2 \leq 2(a^2+b^2) \\
&\leq 2R_t^2\mathbb{E}\|x_t\|^2 + 2\mathbb{E}\|(x_t - m_t x_0)/\sigma_t^2\|^2 && \text{since } \|f\|_{\mathcal{F}_1} \leq R_t \\
&= 2R_t^2\mathbb{E}\|x_t\|^2 + 2\mathbb{E}\|w/\sigma_t\|^2 \\
&= 2R_t^2\mathbb{E}\|x_t\|^2 + 2D/\sigma_t^2.
\end{aligned}$$

Next, we have:

$$\mathbb{E}\sup_{f\in\mathscr{F}_t}\hat{\mathcal{L}}_t^2(f) = \mathbb{E}\sup_{f\in\mathscr{F}_t}\left(\frac{1}{n}\sum_{i=1}^n\|f(x_t^i)-\nabla\log q_t(x_t^i\mid x_0^i)\|^2\right)^2$$

$$\leqslant \mathbb{E}\left(\frac{2R_t^2}{n}\sum_{i=1}^n\|x_t^i\|^2+\frac{2}{n\sigma_t^2}\sum_{i=1}^n\|w^i\|^2\right)^2 \qquad \text{since } \|f\|_{\mathcal{F}_1}\leqslant R_t$$

$$\leqslant \frac{4R_t^4}{n}\sum_{i=1}^n\mathbb{E}\|x_t^i\|^4+\frac{4}{n\sigma_t^4}\sum_{i=1}^n\mathbb{E}\|w^i\|^4 \qquad \text{Cauchy-Schwarz}$$

$$\leqslant 4R_t^4\mathbb{E}\|x_t\|^4+12D^2/\sigma_t^4.$$

Hence,

$$\mathbb{E}[\psi^2(\mathcal{D}_t)] \leqslant \mathbb{E}\sup_{f\in\mathscr{F}_t}[\mathcal{L}_t(f)-(1+\varepsilon)\hat{\mathcal{L}}_t(f)]^2$$

$$\lesssim \sup_{f\in\mathscr{F}_t}\mathcal{L}_t^2(f)+\mathbb{E}\sup_{f\in\mathscr{F}_t}\hat{\mathcal{L}}_t^2(f)$$

$$\lesssim R_t^4\mathbb{E}\|x_t\|^4+D^2/\sigma_t^4.$$

Now we move on to bounding $\mathbb{E}[\psi^2(\check{\mathcal{D}}_n)]$. Defining $\mathcal{E}_x := \mathcal{E}_x(\delta/n)$,

$$\check{x}_t-m_t\check{x}_0 = (x_t-m_tx_0)\cdot\mathbf{1}\{\mathcal{E}_x\} = \sigma_tw\cdot\mathbf{1}\{\mathcal{E}_x\},$$

and therefore:

$$\mathbb{E}\|(\check{x}_t-m_t\check{x}_0)/\sigma_t^2\|^4 = \mathbb{E}\|w/\sigma_t\|^4\mathbf{1}\{\mathcal{E}_x\} \leqslant \mathbb{E}\|w/\sigma_t^4\| \leqslant 3D^2/\sigma_t^4.$$

Hence:

$$\mathbb{E}\sup_{f\in\mathscr{F}_t}\hat{\mathcal{L}}_t^2(f;\check{\mathcal{D}}_t) = \mathbb{E}\sup_{f\in\mathscr{F}_t}\left(\frac{1}{n}\sum_{i=1}^n\|f(\check{x}_t^i)-\nabla\log q_t(\check{x}_t^i\mid\check{x}_0^i)\|^2\right)^2$$

$$\leqslant \mathbb{E}\left(\frac{2R_t^2}{n}\sum_{i=1}^n\|\check{x}_t^i\|^2+\frac{2}{n\sigma_t^2}\sum_{i=1}^n\|(\check{x}_t^i-m_t\check{x}_0^i)/\sigma_t^2\|^2\right)^2 \quad \text{since } \|f\|_{\mathcal{F}_1}\leqslant R_t$$

$$\leqslant \frac{4R_t^4}{n}\sum_{i=1}^n\mathbb{E}\|\check{x}_t^i\|^4+\frac{4}{n\sigma_t^4}\sum_{i=1}^n\mathbb{E}\|(\check{x}_t^i-m_t\check{x}_0^i)/\sigma_t^2\|^4 \qquad \text{Cauchy-Schwarz}$$

$$\leqslant \frac{4R_t^4}{n}\sum_{i=1}^n\mathbb{E}\|x_t^i\|^4+\frac{4}{n\sigma_t^4}\sum_{i=1}^n\mathbb{E}\|(x_t^i-m_tx_0^i)/\sigma_t^2\|^4$$

$$= \frac{4R_t^4}{n}\sum_{i=1}^n\mathbb{E}\|x_t^i\|^4+\frac{4}{n\sigma_t^4}\sum_{i=1}^n\mathbb{E}\|w^i/\sigma_t\|^4$$

$$\leqslant 4R_t^4\mathbb{E}\|x_t\|^4+12D^2/\sigma_t^4.$$

Therefore, we conclude that:

$$\max\{\mathbb{E}[\psi^2(\mathcal{D}_t)],\mathbb{E}[\psi^2(\check{\mathcal{D}}_t)]\} \lesssim R_t^4\mathbb{E}\|x_t\|^4+\sigma_t^{-4}D^2. \tag{E.7}$$

Plugging (E.7) into (E.6),

$$\mathbb{E}[\psi(\mathcal{D}_t)] \leqslant \mathbb{E}[\psi(\check{\mathcal{D}}_t)]+c(R_t^2\|x_t\|_{L_4(p_t)}^2+\sigma_t^{-2}D)\cdot\delta^{1/2}$$

$$\leqslant \mathbb{E}\sup_{f\in\mathscr{F}_t}[\mathcal{L}_t(f)-(1+\varepsilon)\hat{\mathcal{L}}_t(f;\check{\mathcal{D}}_t)]+c(R_t^2\|x_t\|_{L_4(p_t)}^2+\sigma_t^{-2}D)\cdot\delta^{1/2}$$

$$\leqslant \mathbb{E}\sup_{f\in\mathscr{F}_t}[\check{\mathcal{L}}_t(f;\delta/n)-(1+\varepsilon)\hat{\mathcal{L}}_t(f;\check{\mathcal{D}}_t)]+\sup_{f\in\mathscr{F}_t}[\mathcal{L}_t(f)-\check{\mathcal{L}}_t(f;\delta/n)]$$

$$+c(R_t^2\|x_t\|_{L_4(p_t)}^2+\sigma_t^{-2}D)\cdot\delta^{1/2}.$$

Next, define:

$$V_f((x, \bar{x})) := \|f(\bar{x}) - \nabla \log q_t(\bar{x} \mid x)\|^2.$$

Observe by Jensen's inequality we can bound

$$\mathbb{E}[V_f^2((x_0, x_t))] \leqslant \mathbb{E} \sup_{f \in \mathscr{F}_t} \hat{\mathcal{L}}_t^2(f) \lesssim R_t^4 \mathbb{E}\|x_t\|^4 + D^2/\sigma_t^4.$$

Therefore, by application of Proposition C.6,

$$\mathbb{E}[V_f((x_0, x_t))] \leqslant \mathbb{E}[V_f((\check{x}_0, \check{x}_t))] + \sqrt{\mathbb{E}[V_f^2((x_0, x_t))]} \cdot \sqrt{\delta/n}$$
$$\leqslant \mathbb{E}[V_f((\check{x}_0, \check{x}_t))] + c'(R_t^2\|x_t\|_{L_4(p_t)}^2 + \sigma_t^{-2}D) \cdot \sqrt{\delta/n}.$$

Hence,

$$\sup_{f \in \mathscr{F}_t}[\mathcal{L}_t(f) - \check{\mathcal{L}}_t(f; \delta/n)] = \sup_{f \in \mathscr{F}_t}[\mathbb{E}[V_f((x_0, x_t))] - \mathbb{E}[V_f((\check{x}_0, \check{x}_t))]]$$
$$\leqslant c'(R_t^2\|x_t\|_{L_4(p_t)}^2 + \sigma_t^{-2}D) \cdot \sqrt{\delta/n},$$

from which the claim follows. $\qquad\square$

The second result proves uniform convergence over truncated inputs.

**Proposition E.4.** *Fix $\delta \in (0, 1)$. Define the truncated random vectors $(\check{x}_0, \check{x}_t) \sim \nu_{\delta/n}$ (cf. (E.1)), and let $\check{D}_t \sim \nu_{\delta/n}^{\otimes n}$. For some $R_t \geqslant 1$, define $\mathscr{F}_t := \{s : \mathbb{R}^D \mapsto \mathbb{R}^D \mid \|s\|_{\mathscr{F}_1} \leqslant R_t\}$. For all $\varepsilon \in (0, 1]$, we have that:*

$$\mathbb{E} \sup_{f \in \mathscr{F}_t} [\check{\mathcal{L}}_t(f; \delta/n) - (1 + \varepsilon)\hat{\mathcal{L}}_t(f; \check{D}_t)]$$
$$\leqslant \tilde{O}(1)(1 + \varepsilon^{-1}) \left[ \frac{R_t^2 \check{\mu}_{t,x}^2(\delta/n)D + \check{\mu}_{t,q}^2(\delta/n)}{n} \right] + O(1) \frac{R_t^2\|x_t\|_{L_4(p_t)}^2 + D/\sigma_t^2}{n^2}.$$

*Proof.* Let $\check{\mu}_{t,x} := \check{\mu}_{t,x}(\delta/n)$ and similarly $\check{\mu}_{t,q} := \check{\mu}_{t,q}(\delta/n)$. We first observe that the following holds almost surely:

$$\|f(\check{x}_t) - \nabla \log q_t(\check{x}_t \mid \check{x}_0)\| \leqslant R_t\|\check{x}_t\| + \|\nabla \log q_t(\check{x}_t \mid \check{x}_0)\|$$
$$\leqslant R_t\check{\mu}_{t,x} + \check{\mu}_{t,q} =: B_{\mathcal{H}}.$$

We consider the hypothesis of functions:

$$\mathcal{H} := \{(x, \bar{x}) \mapsto \|f(\bar{x}) - \nabla \log q_t(\bar{x} \mid x)\| \mid f \in \mathscr{F}_t\},$$

defined over the support $\check{\mathcal{Z}} := \text{supp}((\check{x}_0, \check{x}_t))$, and coupled with the loss function $\phi(z) = z^2$, which is 2-smooth. From Srebro et al. (2010, Theorem 1), we have with probability at least $1 - \delta$ over $\check{D}_t$, for all $h \in \mathcal{H}$:[9]

$$\mathbb{E}[\phi(h(\check{x}_0, \check{x}_t))] \leqslant (1 + \varepsilon)\frac{1}{n}\sum_{i=1}^{n}\phi(h(\check{x}_0^i, \check{x}_t^i)) + (1 + \varepsilon^{-1})c\left[\log^3 n \cdot \mathfrak{R}_n^2(\mathcal{H}) + \frac{B_{\mathcal{H}}^2 \log(1/\delta)}{n}\right], \tag{E.8}$$

where $c > 0$ is a universal constant, and $\mathfrak{R}_n(\mathcal{H})$ denotes the Rademacher complexity of $\mathcal{H}$:

$$\mathfrak{R}_n(\mathcal{H}) := \sup_{z_{1:n} \subset \check{\mathcal{Z}}} \mathbb{E}_\varepsilon \sup_{h \in \mathcal{H}} \frac{1}{n} \left|\sum_{i=1}^{n} h(z_i)\varepsilon_i\right|.$$

We now bound this Rademacher complexity term. Let $\mathcal{G}$ denote the shifted function class:

$$\mathcal{G} := \{(x, \bar{x}) \mapsto f(\bar{x}) - \nabla \log q_t(\bar{x} \mid x) \mid f \in \mathscr{F}_t\}.$$

---

[9]Note that we ignore the labels $y$ in the setup of Srebro et al. (2010, Theorem 1), as they are immaterial.

Letting $g_0 \in \mathcal{G}$ and $z_i = (x_i, \bar{x}_i) \in \check{Z}$ for $i \in [n]$ be arbitrary, we have:

$$\mathbb{E}_\varepsilon \sup_{h \in \mathcal{H}} \left| \sum_{i=1}^n h(z_i)\varepsilon_i \right|$$

$$= \mathbb{E}_\varepsilon \sup_{g \in \mathcal{G}} \left| \sum_{i=1}^n \varepsilon_i \|g(z_i)\| \right|$$

$$\leqslant \mathbb{E}_\varepsilon \sup_{g \in \mathcal{G}} \left| \sum_{i=1}^n \varepsilon_i(\|g(z_i)\| - \|g_0(z_i)\|) \right| + \mathbb{E} \left| \sum_{i=1}^n \varepsilon_i \|g_0(z_i)\| \right|$$

$$\leqslant \mathbb{E}_\varepsilon \sup_{g \in \mathcal{G}} \left| \sum_{i=1}^n \varepsilon_i(\|g(z_i)\| - \|g_0(z_i)\|) \right| + \sqrt{\sum_{i=1}^n \|g_0(z_i)\|^2} \qquad \text{Jensen's inequality}$$

$$\leqslant \mathbb{E}_\varepsilon \sup_{g,g' \in \mathcal{G}} \left| \sum_{i=1}^n \varepsilon_i(\|g(z_i)\| - \|g'(z_i)\|) \right| + \sqrt{\sum_{i=1}^n \|g_0(z_i)\|^2} \qquad \text{since } g_0 \in \mathcal{G}$$

$$= \mathbb{E}_\varepsilon \sup_{g,g' \in \mathcal{G}} \sum_{i=1}^n \varepsilon_i(\|g(z_i)\| - \|g'(z_i)\|) + \sqrt{\sum_{i=1}^n \|g_0(z_i)\|^2} \qquad \text{since } \mathcal{G} - \mathcal{G} \text{ is symmetric}$$

$$\leqslant 2\mathbb{E}_\varepsilon \sup_{g \in \mathcal{G}} \sum_{i=1}^n \varepsilon_i \|g(z_i)\| + \sqrt{n} B_\mathcal{H}.$$

Next, we proceed with Maurer (2016, Corollary 4), which allows to bound, for Rademacher random *vectors* $\gamma_i \in \{\pm 1\}^D$,

$$\mathbb{E}_\varepsilon \sup_{g \in \mathcal{G}} \sum_{i=1}^n \varepsilon_i \|g(z_i)\| \leqslant \sqrt{2} \mathbb{E}_\gamma \sup_{g \in \mathcal{G}} \sum_{i=1}^n \langle \gamma_i, g(z_i) \rangle$$

$$= \sqrt{2} \mathbb{E}_\gamma \sup_{f \in \mathscr{F}_t} \sum_{i=1}^n \langle \gamma_i, f(\bar{x}_i) \rangle$$

$$\lesssim R_t \check{\mu}_{t,x} \sqrt{Dn},$$

where the last inequality uses Proposition C.9. Putting the terms together,

$$\mathfrak{R}_n(\mathcal{H}) \lesssim (R_t \check{\mu}_{t,x} \sqrt{D} + B_\mathcal{H}) \frac{1}{\sqrt{n}} \lesssim (R_t \check{\mu}_{t,x} \sqrt{D} + \check{\mu}_{t,q}) \frac{1}{\sqrt{n}}.$$

From (E.8), with probability at least $1 - 1/n^4$, for all $h \in \mathcal{H}$:

$$\mathbb{E}[\phi(h(\check{x}_0, \check{x}_t))] \leqslant (1+\varepsilon) \frac{1}{n} \sum_{i=1}^n \phi(h(\check{x}_0^i, \check{x}_t^i)) + (1 + \varepsilon^{-1}) c \log^3 n \left[ \frac{R_t^2 \check{\mu}_{t,x}^2 D + \check{\mu}_{t,q}^2}{n} \right].$$

That is, with probability at least $1 - 1/n^4$,

$$\sup_{f \in \mathscr{F}_t} [\check{\mathcal{L}}_t(f; \delta/n) - (1+\varepsilon)\hat{\mathcal{L}}_t(f; \check{\mathcal{D}}_t)] \leqslant \tilde{O}(1)(1 + \varepsilon^{-1}) \left[ \frac{R_t^2 \check{\mu}_{t,x}^2 D + \check{\mu}_{t,q}^2}{n} \right].$$

Call this event $\mathcal{E}'$. We have that:

$$\mathbb{E} \sup_{f \in \mathscr{F}_t} [\check{\mathcal{L}}_t(f; \delta/n) - (1+\varepsilon)\hat{\mathcal{L}}_t(f; \check{\mathcal{D}}_t)]$$

$$= \mathbb{E} \sup_{f \in \mathscr{F}_t} [\check{\mathcal{L}}_t(f; \delta/n) - (1+\varepsilon)\hat{\mathcal{L}}_t(f; \check{\mathcal{D}}_t)] \mathbf{1}\{\mathcal{E}'\} + \mathbb{E} \sup_{f \in \mathscr{F}_t} [\check{\mathcal{L}}_t(f; \delta/n) - (1+\varepsilon)\hat{\mathcal{L}}_t(f; \check{\mathcal{D}}_t)] \mathbf{1}\{(\mathcal{E}')^c\}$$

$$\lesssim \tilde{O}(1)(1 + \varepsilon^{-1}) \left[ \frac{R_t^2 \check{\mu}_{t,x}^2 D + \check{\mu}_{t,q}^2}{n} \right] + \frac{1}{n^2} \sqrt{\mathbb{E} \sup_{f \in \mathscr{F}_t} [\check{\mathcal{L}}_t(f; \delta/n) - (1+\varepsilon)\hat{\mathcal{L}}_t(f; \check{\mathcal{D}}_t)]^2}.$$

To finish the proof, we observe that:

$$\check{\mathcal{L}}_t(f; \delta/n) = \mathbb{E}_{(\check{x}_0, \check{x}_t) \sim \nu_{\delta/n}} \| f(\check{x}_t) - \nabla \log q_t(\check{x}_t \mid \check{x}_0) \|^2$$
$$\leqslant 2\mathbb{E}\| f(\check{x}_t) \|^2 + 2\mathbb{E}\| \nabla \log q_t(\check{x}_t \mid \check{x}_0) \|^2$$
$$\leqslant 2R_t^2 \mathbb{E}\| \check{x}_t \|^2 + 2\mathbb{E}\| (\check{x}_t - m_t \check{x}_0)/\sigma_t^2 \|^2$$
$$\leqslant 2R_t^2 \mathbb{E}\| x_t \|^2 + 2\mathbb{E}\| (x_t - m_t x_0)/\sigma_t^2 \|^2$$
$$= 2R_t^2 \mathbb{E}\| x_t \|^2 + 2D/\sigma_t^2.$$

On the other hand, from (E.7),

$$\mathbb{E} \sup_{f \in \mathscr{F}_t} \check{\mathcal{L}}_t^2(f; \check{\mathcal{D}}_t) \lesssim R_t^4 \mathbb{E}\| x_t \|^4 + \sigma_t^{-4} D^2.$$

Hence, combining these bounds together,

$$\mathbb{E} \sup_{f \in \mathscr{F}_t} [\check{\mathcal{L}}_t(f; \delta/n) - (1 + \varepsilon)\hat{\mathcal{L}}_t(f; \check{\mathcal{D}}_t)]$$
$$\lesssim \tilde{O}(1)(1 + \varepsilon^{-1}) \left[ \frac{R_t^2 \check{\mu}_{t,x}^2 D + \check{\mu}_{t,q}^2}{n} \right] + \frac{R_t^2 \| x_t \|_{L_4(p_t)}^2 + D/\sigma_t^2}{n^2}.$$

$\square$

## F  ANALYSIS OF SUBSPACE STRUCTURE (SECTION 3.1)

We now specialize the previous approximation and uniform convergence results to the subspace structure setting.

**Proposition F.1.** *Fix an $M \geqslant 1$. For any $\varepsilon \in (0, \bar{L}M/2)$, there exists an $f_\varepsilon : \mathbb{R}^d \mapsto \mathbb{R}^d$ such that $\sup_{z \in B_2(d,M)} \| f_\varepsilon(z) - \nabla \log \pi_t(z) \| \leqslant \varepsilon$, and*

$$\| f_\varepsilon \|_{\mathcal{F}_1} \leqslant R_{\mathrm{lin}}(\varepsilon, M) := O_d(1)(\bar{L}M)^{(d+3)/2} \varepsilon^{-(d+1)/2} \log^{(d+1)/2}(\bar{L}M/\varepsilon). \tag{F.1}$$

*Proof.* We will invoke Lemma A.4. To do this, we first observe for any $z \in B_2(d, M)$, using Proposition C.4,

$$\| \nabla \log \pi_t(z) \| \leqslant \bar{L}(1 + \| z \|) \leqslant 2\bar{L}M.$$

On the other hand, we know that $\nabla \log \pi_t$ is $L$-Lipschitz. The claim now follows from Lemma A.4.

$\square$

Our next task is to upgrade the previous result to an approximation result for the ambient score $\nabla \log p_t$, using Proposition D.1.

**Proposition F.2.** *Fix an $\varepsilon \in (0, 1)$. There exists an $\hat{s} : \mathbb{R}^D \mapsto \mathbb{R}^D$ such that:*

$$\| \hat{s} \|_{\mathcal{F}_1} \leqslant \tilde{O}_d(1)(\bar{L}(\mu_{t,z} \vee \beta))^{(d+3)/2} \varepsilon^{-(d+1)/2} + 2(D - d)/\sigma_t^2, \quad \| \hat{s} - \nabla \log p_t \|_{L_2(p_t)} \leqslant \varepsilon. \tag{F.2}$$

*Proof.* Define

$$M(\delta) := \left\{ \| U^\mathsf{T} x_t \| \leqslant A_\delta \right\}, \quad A_\delta := c_0(\mu_{t,z} + \beta \sqrt{\log(1/\delta)}).$$

We note that the condition $\mathbb{P}\{ x_t \in M(\delta) \} \geqslant 1 - \delta$ holds for an appropriate choice of $c_0$.

Now, given $\varepsilon, \delta \in (0, 1)$, from Proposition F.1 there exists $\hat{h} : \mathbb{R}^d \mapsto \mathbb{R}^d$ such that:

$$\| \hat{h} \|_{\mathcal{F}_1} \leqslant R_{\mathrm{lin}}(\varepsilon, A_\delta), \quad \sup_{z \in B_2(d, A_\delta)} \| \hat{h}(z) - \nabla \log \pi_t(z) \| \leqslant \varepsilon.$$

Embed $\hat{h}$ to a function $\hat{s} : \mathbb{R}^D \mapsto \mathbb{R}^D$ by:

$$\hat{s}(x) = U\hat{h}(U^\mathsf{T} x) - \frac{1}{\sigma_t^2}(I - UU^\mathsf{T})x,$$

and observe that (cf. Proposition A.1):

$$
\begin{aligned}
\sup_{x \in M(\delta)} \|\hat{s}(x) - s_\star(x)\| &= \sup_{x \in M(\delta)} \|U\hat{h}(U^\mathsf{T}x) - U\nabla \log \pi_t(U^\mathsf{T}x)\| \\
&\leqslant \sup_{x \in M(\delta)} \|\hat{h}(U^\mathsf{T}x) - \nabla \log \pi_t(U^\mathsf{T}x)\| \\
&\leqslant \sup_{z \in B_2(d, A_\delta)} \|\hat{h}(z) - \nabla \log \pi_t(z)\| \\
&\leqslant \varepsilon.
\end{aligned}
$$

Next, we bound the $\mathcal{F}_1$-norm of $\hat{s}$. To do this, we first bound the $\mathcal{F}_1$-norm of $x \mapsto (I - UU^\mathsf{T})x$ by representing it by the following sum of Dirac masses

$$
\sum_{i=1}^{D-d} \delta_{(u_i, u_i)} + \delta_{(-u_i, -u_i)},
$$

and hence $\|x \mapsto (I - UU^\mathsf{T})x\|_{\mathcal{F}_1} \leqslant 2(D - d)$. Next, recall by Fact A.3 that $\|x \mapsto U\hat{h}(U^\mathsf{T}x)\|_{\mathcal{F}_1} = \|h\|_{\mathcal{F}_1}$. Combining these results,

$$
\begin{aligned}
\|\hat{s}\|_{\mathcal{F}_1} &\leqslant \|x \mapsto U\hat{h}(U^\mathsf{T}x)\|_{\mathcal{F}_1} + \frac{1}{\sigma_t^2}\|x \mapsto (I - UU^\mathsf{T})x\|_{\mathcal{F}_1} \\
&\leqslant \|\hat{h}\|_{\mathcal{F}_1} + \frac{2}{\sigma_t^2}(D - d) \leqslant R_{\mathrm{lin}}(\varepsilon, A_\delta) + \frac{2}{\sigma_t^2}(D - d) =: R(\varepsilon, \delta).
\end{aligned} \tag{F.3}
$$

That is, we have shown that for $\varepsilon > 0$ and $\delta \in (0, 1)$, there exists a $\hat{s} : \mathbb{R}^D \mapsto \mathbb{R}^D$ such that:

$$
\|\hat{s}\|_{\mathcal{F}_1} \leqslant R(\varepsilon, \delta), \quad \sup_{x \in M(\delta)} \|\hat{s}(x) - \nabla \log p_t(x)\| \leqslant \varepsilon.
$$

This verifies condition (D.1) of Proposition D.1. We now need to solve for a $\delta_\star$ which satisfies the conditions listed in (D.2). By several applications of Proposition C.7, the conditions listed (D.2) are satisfied with a $\delta_\star \in (0, 1)$ satisfying:

$$
\log(1/\delta_\star) \leqslant O_d(1) \log\left(\frac{\bar{L}D\mu_{t,x}\beta}{\varepsilon\sigma_t}\right).
$$

Since we do not track the exact form of the leading $O_d(1)$ constant, we skip the specific calculations. The result now follows from Proposition D.1 after estimating $R(\varepsilon/2, \delta_\star)$. First, we bound,

$$
A_{\delta_\star} = c_0(\mu_{t,z} + \beta\sqrt{\log(1/\delta_\star)}) = \tilde{O}_d(1)(\mu_{t,z} \vee \beta).
$$

Hence,

$$
R(\varepsilon/2, \delta_\star) \leqslant \tilde{O}_d(1)(\bar{L}(\mu_{t,z} \vee \beta))^{(d+3)/2}\varepsilon^{-(d+1)/2}.
$$

Therefore, the result follows. □

Now we have the tools in place to prove Theorem 3.3, our main score estimation result for this section.

**Theorem 3.3.** *Suppose that $p_0$ follows the latent structure (3.8), and that both Assumption 3.1 and Assumption 3.2 hold. Fix a $t > 0$ and define*

$$
\mathscr{F}_t := \{s : \mathbb{R}^D \mapsto \mathbb{R}^D \mid \|s\|_{\mathcal{F}_1} \leqslant R_t\}, \quad R_t := \bar{R}_t n^{\frac{d+1}{2(d+5)}} + \frac{D}{\sigma_t^2}, \tag{3.9}
$$

*where $\bar{R}_t$ does not depend on $n$.[10] Suppose that $n$ satisfies*

$$
n \geqslant n_0(t) := \mathrm{poly}(D, 1/\sigma_t, \mu_{t,x} \vee \beta) \cdot \mathrm{poly}_d(\bar{L}, \mu_{t,z} \vee \beta). \tag{3.10}
$$

*Then, the empirical risk minimizer $\hat{s}_t \in \arg\min_{s \in \mathscr{F}_t} \hat{\mathcal{L}}_t(s)$ satisfies:*

$$
\mathbb{E}_{\mathcal{D}_t}[\mathcal{R}_t(\hat{s}_t)] \leqslant \tilde{O}_d(1)\left[\frac{D^2}{\sigma_t^2 n}(\bar{L}(\mu_{t,z} \vee \beta))^{d+3}(\mu_{t,x} \vee \beta)^2\right]^{\frac{2}{d+5}} + \tilde{O}_d(1)\sqrt{\frac{D^3}{\sigma_t^6 n}(\mu_{t,x} \vee \beta)^2}.
$$

---

[10] The explicit dependence of $\bar{R}_t$ on the other problem parameters is detailed in the proof.

*Proof.* First, by Proposition F.2, we know if we set $R_t$ to be

$$R_t = \tilde{O}_d(1)(\bar{L}(\mu_{t,z} \vee \beta))^{(d+3)/2}\varepsilon^{-(d+1)/2} + 2(D-d)/\sigma_t^2,$$

then, we have $\inf_{s \in \mathscr{F}_t} \mathcal{L}_t(s) \leqslant \varepsilon^2$.

Now we need to apply Lemma E.2. To do this, we need to define our auxiliary truncated random vectors (cf. (E.1)). We choose the definition:

$$\mathcal{E}_x(\delta) := \{\|z_0\| \leqslant \mu_0 + \beta\sqrt{2\log(2/\delta)},\ \|w\| \leqslant \sqrt{D} + \sqrt{2\log(2/\delta)}\}, \tag{F.4}$$

which by sub-Gaussian concentration followed by a union bound satisfies $\mathbb{P}\{\mathcal{E}_x(\delta)\} \geqslant 1 - \delta$. Note that under this definition of $\mathcal{E}_x(\delta)$, we can take:

$$\check{\mu}_{t,x}(\delta) \lesssim \mu_{t,x} + \beta\sqrt{\log(1/\delta)}, \quad \check{\mu}_{t,q}(\delta) \lesssim \sigma_t^{-1}(\sqrt{D} + \sqrt{\log(1/\delta)}). \tag{F.5}$$

By applying Lemma E.2, we obtain for $\gamma \in (0, 1)$,

$$\mathbb{E} \sup_{f \in \mathscr{F}_t} [\mathcal{L}_t(f) - (1+\gamma)\hat{\mathcal{L}}_t(f)] \leqslant \tilde{O}(1)(1 + \gamma^{-1})\frac{D}{n}\left[R_t^2(\mu_{t,x} \vee \beta)^2 + 1/\sigma_t^2\right].$$

By the basic inequality Proposition E.1,

$$\mathbb{E}_{\mathcal{D}_t}[\mathcal{R}_t(\hat{f}_t)] \leqslant 2\varepsilon^2 + \tilde{O}_d(\gamma^{-1})\frac{D}{n}(\bar{L}(\mu_{t,z} \vee \beta))^{d+3}\varepsilon^{-(d+1)}(\mu_{t,x} \vee \beta)^2$$

$$+ \tilde{O}_d(\gamma^{-1})\frac{D^2}{n\sigma_t^4}(\mu_{t,x} \vee \beta)^2 + \gamma \cdot C_t.$$

We now optimize this expression over both $\varepsilon, \gamma \in (0, 1)$. We first optimize both expressions ignoring the constraint that $\varepsilon, \gamma < 1$. First, optimizing over $\gamma$, we set

$$\gamma = \tilde{O}_d(1)\sqrt{\frac{1}{C_t}\left[\frac{D}{n}(\bar{L}(\mu_{t,z} \vee \beta))^{d+3}\varepsilon^{-(d+1)}(\mu_{t,x} \vee \beta)^2 + \frac{D^2}{n\sigma_t^4}(\mu_{t,x} \vee \beta)^2\right]},$$

and from this we obtain:

$$\mathbb{E}_{\mathcal{D}_t}[\mathcal{R}_t(\hat{f}_t)] \leqslant 2\varepsilon^2 + \sqrt{\tilde{O}_d(1)\frac{C_t D}{n}(\bar{L}(\mu_{t,z} \vee \beta))^{d+3}\varepsilon^{-(d+1)}(\mu_{t,x} \vee \beta)^2} + \sqrt{\tilde{O}_d(1)\frac{C_t D^2}{n\sigma_t^4}(\mu_{t,x} \vee \beta)^2}.$$

Now optimizing over $\varepsilon$, we set

$$\varepsilon = \tilde{O}_d(1)\left[\frac{C_t D}{n}(\bar{L}(\mu_{t,z} \vee \beta))^{d+3}(\mu_{t,x} \vee \beta)^2\right]^{1/(d+5)},$$

and obtain:

$$\mathbb{E}_{\mathcal{D}_t}[\mathcal{R}_t(\hat{f}_t)] \leqslant \tilde{O}_d(1)\left[\frac{C_t D}{n}(\bar{L}(\mu_{t,z} \vee \beta))^{d+3}(\mu_{t,x} \vee \beta)^2\right]^{2/(d+5)} + \tilde{O}_d(1)\sqrt{\frac{C_t D^2}{n\sigma_t^4}(\mu_{t,x} \vee \beta)^2}.$$

The proof concludes by setting $n$ large enough so that both $\varepsilon, \gamma < 1$. $\square$

We now restate and prove Corollary 3.4, our main end-to-end bound for the latent subspace case.

**Corollary 3.4.** *Fix $\varepsilon, \zeta \in (0, 1)$. Suppose that $p_0$ follows the latent structure* (3.8)*, and that both Assumption 3.1 and Assumption 3.2 hold. Consider the exponential integrator* (3.4) *with:*

$$T = c_0 \log\left(\frac{\sqrt{D} \vee \mu_0}{\varepsilon}\right), \quad N = 2\left\lceil c_1 \frac{D \vee \mu_0^2}{\varepsilon^2}\left[\log^2\left(\frac{\sqrt{D} \vee \mu_0}{\varepsilon}\right) + \log^2\left(\frac{1}{\zeta}\right)\right]\right\rceil, \tag{3.11}$$

*and reverse process discretization timesteps $\{\tau_i\}_{i=0}^N$ defined as:*

$$\tau_i = \begin{cases} 2(T-1)\frac{i}{N} & \text{if } i \in \{0, \ldots, N/2\}, \\ T - \zeta^{2i/N-1} & \text{if } i \in \{N/2+1, \ldots, N\}. \end{cases} \tag{3.12}$$

*Next, define the forward process timesteps $\{t_i\}_{i=0}^{N-1}$ by $t_i := T - \tau_{N-i}$. Suppose the exponential integration scheme is run with score functions $\{\hat{s}_{t_i}\}_{i=0}^{N-1}$, where $\hat{s}_{t_i} \in \arg\min_{s \in \mathscr{F}_{t_i}} \hat{\mathcal{L}}_{t_i}(s)$ with $\mathscr{F}_t$ as defined in (3.9). Suppose furthermore that $n$ satisfies:*

$$n \geqslant \tilde{O}_d(1) \max\left\{ \frac{D^2}{\zeta}(\bar{L}(\mu_z \vee \beta))^{d+3}(\mu_x \vee \beta)^2 \cdot \varepsilon^{-(d+5)}, \frac{D^3}{\zeta^3}(\mu_x \vee \beta)^2 \cdot \varepsilon^{-4}, n_0(\zeta) \right\},$$

*where $n_0(\cdot)$ is defined in (3.10). With constant probability (over the randomness of the training datasets $\{\mathcal{D}_{t_i}\}_{i=0}^{N-1}$), we have that $\mathrm{KL}(p_\zeta \,\|\, \mathrm{Law}(\hat{y}_{T-\varsigma})) \leqslant \varepsilon^2$, where $\mathrm{Law}(\hat{y}_{T-\varsigma})$ refers to the distribution of the random vector $\hat{y}_{T-\varsigma}$.*

*Proof.* Using the bounds $\mu_{t,z} \leqslant \mu_z$ and $\mu_{t,x} \leqslant \mu_x$, from Theorem 3.3 we have that the following ERM bound holds for all $t \in [0, T]$:

$$\mathbb{E}_{\mathcal{D}_t}[\mathcal{R}_t(\hat{f}_t)] \leqslant \tilde{O}_d(1) \left[ \frac{D^2}{n\sigma_t^2}(\bar{L}(\mu_z \vee \beta))^{d+3}(\mu_x^2 \vee \beta^2) \right]^{2/(d+5)} + \tilde{O}_d(1)\sqrt{\frac{D^3}{n\sigma_t^6}(\mu_x^2 \vee \beta^2)}.$$

Furthermore, since $e^{-x} \leqslant 1 - x/2$ for $x \in [0, 1.59]$, then for $t \leqslant 0.795$ we have

$$\sigma_t^2 = 1 - \exp(-2t) \geqslant t \implies 1/\sigma_t^2 \leqslant 1/t.$$

On the other hand, for $t > 0.795$, we have the bound

$$\sigma_t^2 = 1 - \exp(-2t) \geqslant 1 - \exp(-1.59) \geqslant 0.796 \implies 1/\sigma_t^2 \lesssim 1.$$

Combining these inequalities we have that $1/\sigma_t^2 \leqslant 1/\sigma_\varsigma^2 \lesssim 1/\zeta$ for all $t \geqslant \varsigma$.

Hence, using the choice of $T$, $N$ from (3.11) and $\{t_i\}_{i=0}^{N}$ as specified in Lemma C.2, we have that:

$$\sum_{k=0}^{N-1} \gamma_k \mathbb{E}_{p_{T-t_k}} \|\hat{f}_{T-t_k} - \nabla\log p_{T-t_k}\|^2$$

$$\lesssim T \left[ \tilde{O}_d(1) \left[ \frac{D^2}{n\zeta}(\bar{L}(\mu_z \vee \beta))^{d+3}(\mu_x^2 \vee \beta^2) \right]^{2/(d+5)} + \tilde{O}_d(1)\sqrt{\frac{D^3}{n\zeta^3}(\mu_x^2 \vee \beta^2)} \right]$$

$$\lesssim \log\left(\frac{\sqrt{D} \vee \mu_0}{\varepsilon}\right) \left[ \tilde{O}_d(1) \left[ \frac{D^2}{n\zeta}(\bar{L}(\mu_z \vee \beta))^{d+3}(\mu_x^2 \vee \beta^2) \right]^{2/(d+5)} + \tilde{O}_d(1)\sqrt{\frac{D^3}{n\zeta^3}(\mu_x^2 \vee \beta^2)} \right].$$

Hence, in order to make $\varepsilon_{\mathrm{score}}^2 \leqslant \varepsilon^2$, we need to take $n$ large enough such that the following conditions hold:

$$\log\left(\frac{\sqrt{D} \vee \mu_0}{\varepsilon}\right) \tilde{O}_d(1) \left[ \frac{D^2}{n\zeta}(\bar{L}(\mu_z \vee \beta))^{d+3}(\mu_x^2 \vee \beta^2) \right]^{2/(d+5)} \lesssim \varepsilon^2,$$

$$\log\left(\frac{\sqrt{D} \vee \mu_0}{\varepsilon}\right) \tilde{O}_d(1)\sqrt{\frac{D^3}{n\zeta^3}(\mu_x^2 \vee \beta^2)} \lesssim \varepsilon^2.$$

Hence, we need to take $n$ satisfying:

$$n \geqslant \tilde{O}_d(1) \max\left\{ \frac{D^2}{\zeta}(\bar{L}(\mu_z \vee \beta))^{d+3}(\mu_x^2 \vee \beta^2) \cdot \varepsilon^{-(d+5)}, \frac{D^3}{\zeta^3}(\mu_x^2 \vee \beta^2) \cdot \varepsilon^{-4} \right\}.$$

On the other hand we also need to take $n \geqslant n_0(\zeta)$ (cf. (3.10)). The claim now follows. $\square$

## G  ANALYSIS OF INDEPENDENT COMPONENTS (SECTION 3.2)

We follow a very similar structure as in Appendix F. We first start with an approximation result.

**Proposition G.1.** *For an $M \geqslant 1$ and $i \in [K]$. For any $\varepsilon \in (0, \bar{L}_i M/2)$, there exists an $f_\varepsilon : \mathbb{R}^{d_i} \mapsto \mathbb{R}^{d_i}$ such that $\sup_{z \in B_2(d_i, M)} \|f_\varepsilon(z) - \nabla\log\pi_t^{(i)}(z)\| \leqslant \varepsilon$, and*

$$\|f_\varepsilon\|_{\mathcal{F}_1} \leqslant R_{\mathrm{ind}}^{(i)}(\varepsilon, M) := O_d(1)(\bar{L}_i M)^{(d+3)/2}\varepsilon^{-(d+1)/2}\log^{(d+1)/2}(\bar{L}_i M/\varepsilon). \qquad (\text{G.1})$$

*Proof.* The proof is nearly identical to Proposition F.1, and therefore we omit the details. □

Next, we upgrade the previous approximation result to approximation in $L_2(p_t)$.

**Proposition G.2.** *Fix $\varepsilon_1, \ldots, \varepsilon_K \in (0, 1)$. There exists an $\hat{s} : \mathbb{R}^D \mapsto \mathbb{R}^D$ satisfying:*

$$\|\hat{s}\|_{\mathcal{F}_1} \leqslant \sum_{i=1}^K \tilde{O}_{d_i}(1)(\bar{L}_i(\mu_{t,x}^{(i)} \vee \beta))^{(d_i+3)/2} \varepsilon_i^{-(d_i+1)/2}, \quad \|\hat{s} - \nabla \log p_t\|_{L_2(p_t)} \leqslant \sqrt{\sum_{i=1}^K \varepsilon_i^2}.$$

*Proof.* Recall that $P_i \in \mathbb{R}^{d_i \times D}$ selects the coordinates corresponding to the $i$-th variable group (cf. Proposition A.2). Define the sets $M^{(i)}(\delta)$ as:

$$M^{(i)}(\delta) := \left\{ x \in \mathbb{R}^D \mid \|P_i U^\mathsf{T} x\| \leqslant A_\delta^{(i)} \right\}, \quad A_\delta^{(i)} := c_0(\mu_{t,x}^{(i)} + \beta_i \sqrt{\log(1/\delta)}), \quad i \in [K].$$

With appropriate choice of $c_0$, we have that $\mathbb{P}_{x_t \sim p_t}\{x_t \in M^{(i)}(\delta)\} \geqslant 1 - \delta$.

Given $\varepsilon, \delta \in (0, 1)$, from Proposition G.1, there exists $\hat{h}_i : \mathbb{R}^d \mapsto \mathbb{R}^d$ for $i \in [K]$ such that:

$$\|\hat{h}_i\|_{\mathcal{F}_1} \leqslant R_{\text{ind}}^{(i)}(\varepsilon, A_\delta^{(i)}), \quad \sup_{z \in B_2(d, A_\delta^{(i)})} \|\hat{h}_i(z) - \nabla \log \pi_t^{(i)}(z)\| \leqslant \varepsilon.$$

Now define $\hat{s}_i := U P_i^\mathsf{T} \hat{h}_i(P_i U^\mathsf{T} x)$. Observe that:

$$\sup_{x \in M^{(i)}(\delta)} \|\hat{s}_i(x) - U P_i^\mathsf{T} \nabla \log \pi_t^{(i)}(P_i U^\mathsf{T} x)\| \leqslant \sup_{x \in M^{(i)}(\delta)} \|\hat{h}_i(P_i U^\mathsf{T} x) - \nabla \log \pi_t^{(i)}(P_i U^\mathsf{T} x)\|$$

$$\leqslant \sup_{z \in B_2(d_i, A_\delta^{(i)})} \|\hat{h}_i(z) - \nabla \log \pi_t^{(i)}(z)\|$$

$$\leqslant \varepsilon.$$

Next, observe that $\|\hat{s}_i\|_{\mathcal{F}_1} = \|\hat{h}_i\|_{\mathcal{F}_1}$ by Fact A.3. Invoking Proposition D.1 as is done in the proof of Proposition F.2, we have that for all $i \in [K]$, $\|\hat{s}_i - U P_i^\mathsf{T} \nabla \log \pi_t^{(i)}(P_i U^\mathsf{T} \cdot)\|_{L_2(p_t)} \leqslant \varepsilon_i$ and

$$\|\hat{s}_i\|_{\mathcal{F}_1} \leqslant \tilde{O}_{d_i}(1)(\bar{L}_i(\mu_{t,z}^{(i)} \vee \beta_i))^{(d_i+3)/2} \varepsilon_i^{-(d+1)/2}.$$

Recall by Proposition A.2 we have:

$$\nabla \log p_t(x) = \sum_{i=1}^K U P_i^\mathsf{T} \nabla \log \pi_t^{(i)}(P_i U^\mathsf{T} x).$$

Hence, setting $\hat{s} = \sum_{i=1}^K \hat{s}_i$, we have that

$$\|\hat{s} - \nabla \log p_t\|_{L_2(p_t)}^2$$

$$= \mathbb{E} \left\| \sum_{i=1}^K U P_i^\mathsf{T} (\hat{h}_i(P_i U^\mathsf{T} x_t) - \nabla \log \pi_t^{(i)}(P_i U^\mathsf{T} x_t)) \right\|^2$$

$$= \sum_{i=1}^K \mathbb{E} \| U P_i^\mathsf{T} (\hat{h}_i(P_i U^\mathsf{T} x_t) - \nabla \log \pi_t^{(i)}(P_i U^\mathsf{T} x_t)) \|^2 \qquad \text{since } P_j P_i^\mathsf{T} = 0 \text{ for } i \neq j$$

$$= \sum_{i=1}^K \| \hat{s}_i - U P_i^\mathsf{T} \nabla \log \pi_t^{(i)}(P_i U^\mathsf{T} \cdot) \|_{L_2(p_t)}^2$$

$$\leqslant \sum_{i=1}^K \varepsilon_i^2.$$

Furthermore,

$$\|\hat{s}\|_{\mathcal{F}_1} \leqslant \sum_{i=1}^K \|\hat{s}_i\|_{\mathcal{F}_1} \leqslant \sum_{i=1}^K \tilde{O}_{d_i}(1)(\bar{L}_i(\mu_{t,z}^{(i)} \vee \beta_i))^{(d_i+3)/2} \varepsilon_i^{-(d+1)/2}.$$

□

We now prove Theorem 3.7, our score estimation result for the independent components setting.

**Theorem 3.7.** *Suppose that $p_0$ follows the latent structure (3.13), and that both Assumption 3.5 and Assumption 3.6 hold. Fix a $t > 0$ and define*

$$\mathscr{F}_t := \{s : \mathbb{R}^D \mapsto \mathbb{R}^D \mid \|s\|_{\mathcal{F}_1} \leqslant R_t\}, \quad R_t := \sum_{i=1}^K \bar{R}_t^{(i)} n^{\frac{d_i+1}{2(d_i+5)}}, \tag{3.14}$$

*where $\bar{R}_t^{(i)}$ does not depend on $n$. Suppose that $n$ satisfies*

$$n \geqslant n_0(t) := \mathrm{poly}(D, 1/\sigma_t, \mu_{t,x} \vee \beta) \cdot \max_{i \in [K]} \mathrm{poly}_{d_i}(\bar{L}_i, \mu_{t,x}^{(i)} \vee \beta^{(i)}). \tag{3.15}$$

*Then, the empirical risk minimizer $\hat{s}_t \in \arg\min_{s \in \mathscr{F}_t} \hat{\mathcal{L}}_t(s)$ satisfies:*

$$\mathbb{E}_{\mathcal{D}_t}[\mathcal{R}_t(\hat{s}_t)] \leqslant \sum_{i=1}^K \tilde{O}_{d_i}(1) \left[ \frac{D^2 K}{\sigma_t^2 n} (\bar{L}_i(\mu_{t,x}^{(i)} \vee \beta_i))^{d_i+3} (\mu_{t,x} \vee \beta)^2 \right]^{\frac{2}{d_i+5}} + \tilde{O}(1)\sqrt{\frac{D^2}{\sigma_t^4 n}(\mu_{t,x} \vee \beta)^2}.$$

*Proof.* Here we minic the proof of Theorem 3.3. First, by Proposition G.2, we know if we set $R_t$ as

$$R_t = \sum_{i=1}^K \tilde{O}_{d_i}(1)(\bar{L}_i(\mu_{t,x}^{(i)} \vee \beta_i))^{(d_i+3)/2}\varepsilon_i^{-(d_i+1)/2},$$

then, we have $\inf_{s \in \mathscr{F}_t} \mathcal{L}_t(s) \leqslant \sum_{i=1}^K \varepsilon_i^2$.

Our next step is to apply Lemma E.2. To do this we need to define auxiliary truncated random vectors (cf. (E.1)). In this case, we use the definition:

$$\mathcal{E}_x(\delta) := \bigcap_{i \in [K]} \{\|z_0^{(i)}\| \leqslant \mu_0^{(i)} + \beta_i\sqrt{2\log(2K/\delta)}\} \cap \{\|w\| \leqslant \sqrt{D} + \sqrt{2\log(2/\delta)}\}, \tag{G.2}$$

which satisfies $\mathbb{P}\{\mathcal{E}_x(\delta)\} \geqslant 1 - \delta$. We have

$$\check{\mu}_{t,x}(\delta) \lesssim \mu_{t,x} + \beta\sqrt{\log(K/\delta)}, \quad \check{\mu}_{t,q}(\delta) \lesssim \sigma_t^{-1}(\sqrt{D} + \sqrt{\log(1/\delta)}).$$

By applying Lemma E.2, we obtain for $\gamma \in (0, 1)$,

$$\mathbb{E} \sup_{f \in \mathscr{F}_t} [\mathcal{L}_t(f) - (1 + \gamma)\hat{\mathcal{L}}_t(f)]$$

$$\leqslant \tilde{O}(1)\gamma^{-1}\frac{D}{n} \left[ R_t^2(\mu_{t,x} \vee \beta)^2 + 1/\sigma_t^2 \right]$$

$$\leqslant \gamma^{-1}\frac{DK}{n} \sum_{i=1}^K \tilde{O}_{d_i}(1)(\bar{L}_i(\mu_{t,x}^{(i)} \vee \beta_i))^{d_i+3}\varepsilon_i^{-(d_i+1)}(\mu_{t,x} \vee \beta)^2 + \tilde{O}(1)\frac{D}{\gamma n \sigma_t^2}$$

By the basic inequality Proposition E.1,

$$\mathbb{E}_{\mathcal{D}_t}[\mathcal{R}_t(\hat{f}_t)] \leqslant 2\sum_{i=1}^K \varepsilon_i^2 + \gamma^{-1}\frac{DK}{n} \sum_{i=1}^K \tilde{O}_{d_i}(1)(\bar{L}_i(\mu_{t,x}^{(i)} \vee \beta_i))^{d_i+3}\varepsilon_i^{-(d_i+1)}(\mu_{t,x} \vee \beta)^2$$

$$+ \tilde{O}(1)\frac{D}{\gamma n \sigma_t^2} + \gamma \cdot C_t.$$

We now need to optimize over both $\varepsilon_i, \gamma \in (0, 1)$. We first set $\gamma$ as:

$$\gamma = \sqrt{\frac{1}{C_t} \cdot \left[ \frac{DK}{n} \sum_{i=1}^K \tilde{O}_{d_i}(1)(\bar{L}_i(\mu_{t,x}^{(i)} \vee \beta_i))^{d_i+3}\varepsilon_i^{-(d_i+1)}(\mu_{t,x} \vee \beta)^2 + \tilde{O}(1)\frac{D}{n\sigma_t^2} \right]},$$

from which we obtain:

$$\mathbb{E}_{\mathcal{D}_t}[\mathcal{R}_t(\hat{f}_t)] \leqslant 2\sum_{i=1}^K \varepsilon_i^2 + \sum_{i=1}^K \sqrt{\tilde{O}_{d_i}(1)\frac{C_t DK}{n}(\bar{L}_i(\mu_{t,x}^{(i)} \vee \beta_i))^{d_i+3}\varepsilon_i^{-(d_i+1)}(\mu_{t,x} \vee \beta)^2} + \sqrt{\tilde{O}(1)\frac{C_t D}{n\sigma_t^2}}.$$

We now set $\varepsilon_i$ as:

$$\varepsilon_i = \tilde{O}_{d_i}(1) \left[ \frac{C_t D K}{n} (\bar{L}_i(\mu_{t,x}^{(i)} \vee \beta_i))^{d_i+3} (\mu_{t,x} \vee \beta)^2 \right]^{1/(d_i+5)},$$

and obtain:

$$\mathbb{E}_{\mathcal{D}_t}[\mathcal{R}_t(\hat{f}_t)] \leqslant \sum_{i=1}^{K} \tilde{O}_{d_i}(1) \left[ \frac{C_t D K}{n} (\bar{L}_i(\mu_{t,x}^{(i)} \vee \beta_i))^{d_i+3} (\mu_{t,x} \vee \beta)^2 \right]^{2/(d_i+5)} + \sqrt{\tilde{O}(1) \frac{C_t D}{n \sigma_t^2}}.$$

The proof concludes by setting $n$ large enough so that all of $\varepsilon_i, \gamma < 1$. $\qquad \square$

Finally, we conclude with Corollary 3.8, which provides an end-to-end sampling bound.

**Corollary 3.8.** *Fix $\varepsilon, \zeta \in (0,1)$. Suppose that $p_0$ follows the latent structure (3.13), and that both Assumption 3.5 and Assumption 3.6 hold. Consider the exponential integrator (3.4) with $(N, T)$ as in (3.11) and reverse process discretization timesteps $\{\tau_i\}_{i=0}^{N}$ defined as in (3.12). Next, define the forward process timesteps $\{t_i\}_{i=0}^{N-1}$ by $t_i := T - \tau_{N-i}$. Suppose the exponential integration scheme is run with score functions $\{\hat{s}_{t_i}\}_{i=0}^{N-1}$, where $\hat{s}_{t_i} \in \arg\min_{s \in \mathscr{F}_{t_i}} \hat{\mathcal{L}}_{t_i}(s)$ with $\mathscr{F}_t$ as defined in (3.14). Suppose that $n \geqslant n_0(\zeta)$ satisfies:*

$$n \geqslant (\mu_x \vee \beta)^2 \max \left\{ \max_{i \in [K]} \left\{ \frac{\tilde{O}_{d_i}(1) D^2}{\zeta} K^{(d_i+7)/2} (\bar{L}_i(\mu_x^{(i)} \vee \beta))^{d_i+3} \cdot \varepsilon^{-(d_i+5)} \right\}, \frac{\tilde{O}(1) D^2}{\zeta^2} \cdot \varepsilon^{-4} \right\}.$$

*where $n_0(\cdot)$ is defined in (3.15). With constant probability (over the randomness of the training datasets $\{\mathcal{D}_{t_i}\}_{i=0}^{N-1}$), we have that $\mathrm{KL}(p_\zeta \| \mathrm{Law}(\hat{y}_{T-\zeta})) \leqslant \varepsilon^2$.*

*Proof.* We follow the proof of Corollary 3.4. Using the bounds $\mu_{t,x}^{(i)} \leqslant \mu_x^{(i)}$ and $\mu_{t,x} \leqslant \mu_x$, from Theorem 3.7 we have that the following ERM bound holds for all $t \in [0, T]$:

$$\mathbb{E}_{\mathcal{D}_t}[\mathcal{R}_t(\hat{s}_t)] \leqslant \sum_{i=1}^{K} \tilde{O}_{d_i}(1) \left[ \frac{D^2 K}{\sigma_t^2 n} (\bar{L}_i(\mu_x^{(i)} \vee \beta_i))^{d_i+3} (\mu_x \vee \beta)^2 \right]^{\frac{2}{d_i+5}} + \tilde{O}(1) \sqrt{\frac{D^2}{\sigma_t^4 n} (\mu_x \vee \beta)^2}.$$

Recalling that $1/\sigma_t^{-2} \lesssim 1/\zeta$ for all $t \geqslant \zeta$, using the choice of $T, N$ from (3.11) and $\{t_i\}_{i=0}^{N}$ we have that:

$$\sum_{k=0}^{N-1} \gamma_k \mathbb{E}_{p_{T-t_k}} \|\hat{f}_{T-t_k} - \nabla \log p_{T-t_k}\|^2$$

$$\lesssim \log \left( \frac{\sqrt{D} \vee \mu_0}{\varepsilon} \right) \sum_{i=1}^{K} \tilde{O}_{d_i}(1) \left[ \frac{D^2 K}{\zeta n} (\bar{L}_i(\mu_x^{(i)} \vee \beta_i))^{d_i+3} (\mu_x \vee \beta)^2 \right]^{\frac{2}{d_i+5}}$$

$$+ \log \left( \frac{\sqrt{D} \vee \mu_0}{\varepsilon} \right) \tilde{O}(1) \sqrt{\frac{D^2}{\zeta^2 n} (\mu_x \vee \beta)^2}.$$

Hence, in order to make $\varepsilon_{\mathrm{score}}^2 \leqslant \varepsilon^2$, we need to take $n$ large enough such that the following conditions hold:

$$\log \left( \frac{\sqrt{D} \vee \mu_0}{\varepsilon} \right) \sum_{i=1}^{K} \tilde{O}_{d_i}(1) \left[ \frac{D^2 K}{\zeta n} (\bar{L}_i(\mu_x^{(i)} \vee \beta_i))^{d_i+3} (\mu_x \vee \beta)^2 \right]^{\frac{2}{d_i+5}} \lesssim \varepsilon^2,$$

$$\log \left( \frac{\sqrt{D} \vee \mu_0}{\varepsilon} \right) \tilde{O}(1) \sqrt{\frac{D^2}{\zeta^2 n} (\mu_x \vee \beta)^2} \lesssim \varepsilon^2.$$

Hence, we need to take $n$ satisfying:

$$n \geqslant (\mu_x \vee \beta)^2 \max \left\{ \max_{i \in [K]} \left\{ \frac{\tilde{O}_{d_i}(1) D^2}{\zeta} K^{(d_i+7)/2} (\bar{L}_i(\mu_x^{(i)} \vee \beta))^{d_i+3} \cdot \varepsilon^{-(d_i+5)} \right\}, \frac{\tilde{O}(1) D^2}{\zeta^2} \cdot \varepsilon^{-4} \right\}.$$

On the other hand we also need to take $n \geqslant n_0(\zeta)$ (cf. (3.15)). The claim now follows. $\qquad \square$

