# OpenReview forum: "Shallow diffusion networks provably learn hidden low-dimensional structure"
_ICLR.cc/2025/Conference — ICLR 2025 Poster_

### Official Review · Reviewer_kfcb · 2024-10-24

**Soundness:** 3
**Presentation:** 2
**Contribution:** 3
**Rating:** 8
**Confidence:** 3

**Summary:**

This theoretical work shows that for a neural network which is infinitely wide, and where neuron parameters come from the hypersphere,
and there is one of two hidden latent structures in the data, then there is a bound on the emprical risk minimizer which only
depends on the dimension of the latent structure and not the dimension of the ambient space.

If we denote the ambient dimension as D, the two hidden structures considered are 1) a linear combination of lower dimensional
vectors (of dimension d << D) and 2) a linear combination of independent components all of lower dimension than the ambient
space, but but where the sum of the dimensions of the components sum to D.

**Strengths:**

S1: Diffusion models are videly used, so theoretical work on them is highly relevant.

S2: Going from needing a specific architecture to "only" needing the network to be infinitely wide to get a bound, is a step forward.

S3: Corollary 3.4 and 3.8 give bounds on the KL-divergence between the true data distribution and that of the diffusion model,
given enough training samples (and the extra assumptions) which is interesting.

**Weaknesses:**

W1: I found this paper difficult to follow. I count this as a weakness, since I am a machine learning researcher who is interested in
the theory of machine learning, so I should be the target audience. See my questions and suggestions for places where clarification is
needed.

**Questions:**

**Questions:**

Q1: On line 42-44 you write: "these works leave open the difficulty of statistical estimation, and therefore raise
the possibility that the sampling problem’s true difficulty is hidden in the complexity of learning."
Can you explain what you mean by this?
Especially, "the difficulty of statistical estimation", do you mean how close the estimation can get to the true distribution?
And "the sampling problem’s true difficulty is hidden in the complexity of learning.", what do you mean by this?


Q2: Line 165: W.r.t. risk for score function being asymptotic to $n^{-2/(D+4)}$. Do you mean asymptotic as $n \to \infty$?
(If yes, this could also be written in words, then you could avoid introducing the extra notation.)


Q3: Line 175-179: Can you give some intuition of the F_1-norm? Is it right that F_1 is the functions for which the integral is bounded
for all their basis functions? And does that mean the F_1-norm is the smallest integral for the "largest" basis function?


Q4: In line 180, you say that you only consider neuron parameters from the hypersphere. Is there a reason why this
generalizes to neuron parameters chosen over the entire space? If there is, I would like that argument added.
If there is not, I think you should add this assumption in the part of the introduction where you state your contribution.


Q5: In line 231: In your shorthand notation, you define $\mu_{t,x} = \mu_0 \lor \sigma_t\sqrt{D}$. I am used to $\lor$ being
the logical "or", but that cannot be what you mean in this context. So what does it mean here?


Q6: In your remarks under theorem 3.3, you say that you leave showing a $n^{2/(d+4)}$ to future work, but in 3.3 you give an upper
bound, and $n^{2/(d+5)} < n^{2/(d+4)}$ for $n > 1$. So how is $n^{2/(d+4)}$ a better bound?


Q7: I don't recognize the denpendence on $\sigma_t$ which you mention in the remarks after theorem 3.3. Where do you have something
bounded by $\sigma_t^{-4/(d+5)}$?


Q8: In line 491 you mention your "truncation arguments", could you say what you mean by this? And maybe give a reference to where
you make these arguments?


**Suggestions:**

U1: Please make sure to use the full name and not an abbriviation the first time you mention a concept.
	For example: DDPM in line 66, ERM in line 74, DNN in line 90 and GD in line 97.


U2: Please make sure to explain all the notation you use before you use it. If you feel it would take up too much space
in the main paper, you can make a list of used notation in the appendix.
Specific notation, where I cannot find an explanation in the paper:
$\mathcal{F}_t$ in equation 3.7 (I am guessing this is the space of score functions), $\asymp$ in line 165,
g cts in line 176 in the definition of the Total-Variation norm, the $\lor$ in line 231 and equation 3.10 and
Law(.) in last line of corollary 3.4.


U3: I feel the title over-promises. "SHALLOW DIFFUSION NETWORKS PROVABLY LEARN HIDDEN LOW-DIMENSIONAL STRUCTURE"
sounds like it is _all_ shallow networks and _all_ hidden low-dimensional structures. Of course I understand
if you feel the title would get too long if you add all the assumptions, so I would suggest finding a new title
which is as short or shorter, and which does not make the result sound more general than it is.


U4: Typo in line 189: Should be $B_2(r, d)$

---

> ### Author Response · Authors · 2024-11-28
>
> ### Questions
>
> *> Q1: On line 42-44 you write: "these works leave open the difficulty of statistical estimation, and therefore raise the possibility that the sampling problem’s true difficulty is hidden in the complexity of learning." Can you explain what you mean by this? Especially, "the difficulty of statistical estimation", do you mean how close the estimation can get to the true distribution? And "the sampling problem’s true difficulty is hidden in the complexity of learning.", what do you mean by this?*
>
> We thank the reviewer for asking this important question, which we now clarify. There has been substantial progress in the theory of sampling with diffusions in recent years, where it has been shown that the quality of the diffusion sampler is primarily governed by the $L_2$ score estimation error (Lee et al., 2022; Chen et al., 2023d; Lee et al., 2023; Chen et al., 2023a; Benton et al., 2024). While this prior work represents a significant step forward, it still leaves open the critical question of how difficult it is to learn the score function. Unfortunately, there are several results (Wibisono et al. (2024); Zhang et al. (2024); Oko et al. (2023); Dou et al. (2024)) that appear pessimistic: they argue that in the worst (minimax) case, with no latent assumptions on the underlying data distribution, learning the score function can take a number of samples that is *exponential* in the ambient dimension $D$. That is, score learning exhibits a *curse of dimensionality*. If these results represent the actual state of affairs, then they also imply a curse of dimensionality for sampling with diffusion models. This is what we mean by "the sampling problem’s true difficulty is hidden in the complexity of learning": even if we can sample efficiently with a given, accurate score model, it is not clear we can actually learn accurate score models in a sample efficient way.
>
> In this paper, we attack this problem and show how to learn score models efficiently for certain latent structures with simple model architectures. This allows us to derive end-to-end bounds for learning to sample using diffusion models, which do not suffer from the curse of dimensionality.
>
> *> Q2: Line 165: W.r.t. risk for score function being asymptotic to $n^{-2/(D+4)}. Do you mean asymptotic as $n\to\infty$? (If yes, this could also be written in words, then you could avoid introducing the extra notation.)*
>
> This notation is meant to indicate that the rate $R_t$ *scales* as $n^{-2/(D+4)}$, but there are some leading constants that we suppress to lighten notation. In other words, we are emphasizing the scaling with respect to $n$, but not with respect to the other problem parameters. We note that this is *not* an asymptotic rate, as it holds for all $n$. The motivation for pointing out the explicit dependence on $n$ is so we have a baseline rate to compare with later on.
>
> *> Q3: Line 175-179: Can you give some intuition of the $\mathcal F_1$-norm? Is it right that $\mathcal F_1$ is the functions for which the integral is bounded for all their basis functions? And does that mean the $\mathcal F_1$-norm is the smallest integral for the "largest" basis function?*
>
> The intuition is as follows. Let us consider for simplicity the case where the output of the network is scalar. Each neuron corresponds to a (rescaled) basis function $\phi$, a two-layer network with $N$ neurons is a weighted sum of basis functions with weights $\\{a_i\\}_{i=1}^N$, and we define the parameter norm $\Vert \hat{f}_N\Vert$ of the network $\hat{f}_N$ as the sum of the absolute values of the weights (i.e., the $\ell_1$ norm of the weights in the sum). That is,
>
> $$\hat{f}\_N(x) = \frac{1}{N} \sum\_{i=1}^N \phi(w\_i^\top x)a\_i,\qquad \Vert \hat{f}\_N \Vert = \sum\_{i=1}^N |a\_i|.$$
>
> It is natural to extend this idea to the infinite width limit, replacing the weighted sum by an integration over a signed measure on the space of basis functions,
>
> $$\lim_{N\to\infty}\hat{f}_N(x) \to \hat{f}(x) = \int \phi(w^\top x)d\mu(w), \qquad \Vert \hat{f}\Vert = \\| \mu \\|\_{\mathsf{TV}}.$$
>
> Above, $d\mu(w)$ plays the same role for $w$ that $a_i$ plays for $w_i$ in the finite-width case. The analogue of the parameter norm is then the total variation of the measure. The $\mathcal F_1$ norm $\\|f\\|_{\mathcal F_1}$ is the minimal parameter norm required to represent a  function $f$ in the infinite width limit. That is, it is the minimal total variation of a measure that recovers $f$.
>
> More concretely, for a function $f$ such that there exists a $\mu$ with $f(x) = \int \phi(w^\top x)d\mu(w)$ for all $x$, we take
>
> $$\Vert f \Vert_{\mathcal F_1} = \inf_{\mu} \\| \mu \\|\_{\mathsf{TV}} \quad s.t.\quad  f(x) = \int \phi(w^\top x)d\mu(x).$$
>
> In words, in the case where the measure $\mu$ is unique, we take $\Vert f\Vert_{\mathcal F_1} = \\| \mu \\|\_{\mathsf{TV}}$. If there is more than one measure that can reproduce the function $f$, we take the measure with smallest total variation norm.

---

> ### Author Response · Authors · 2024-11-28
>
> *> Q4: In line 180, you say that you only consider neuron parameters from the hypersphere. Is there a reason why this generalizes to neuron parameters chosen over the entire space? If there is, I would like that argument added. If there is not, I think you should add this assumption in the part of the introduction where you state your contribution.*
>
> Because we consider the ReLU activation which is homogeneous, the contribution $v\sigma(w^Tx)$ of a neuron with general weights $v,w$ is always equal to a rescaling of a neuron with unit weights $v\sigma(w^Tx)=\\|v\\|\\|w\\| \bar{v}\sigma(\bar{w}^Tx)$, where $\bar{v}=\frac{v}{\\|v\\|}$ and $\bar{v}=\frac{v}{\\|v\\|}$. This is why we can restrict ourselves to unit neurons without loss of generality.
>
> *> Q5: In line 231: In your shorthand notation, you define $\vee$. I am used to  being the logical "or", but that cannot be what you mean in this context. So what does it mean here?*
>
> The $\vee$ notation is shorthand notation for $\max$, i.e., $a \vee b = \max\\{a, b\\}$. We have updated the notation section of our paper to include this notation.
>
> *> Q6: In your remarks under theorem 3.3, you say that you leave showing a $n^{2/(d+4)}$ to future work, but in 3.3 you give an upper bound, and $n^{2/(d+5)} < n^{2/(d+4)}$ for $n > 1$. So how is $n^{2/(d+4)}$ a better bound?*
>
> Our rates are of the form $\mathrm{Risk} \leq \frac{\textrm{numerator terms}}{n^{2/(d+5)}}$. What we mean is that, we want in future work to show rates of the form
> $\mathrm{Risk}' \leq \frac{\textrm{numerator terms}}{n^{2/(d+4)}}$.
> As you pointed out, since $n^{2/(d+5)} < n^{2/(d+4)}$, then we have $\mathrm{Risk}' < \mathrm{Risk}$, and hence the latter is a better rate. We have updated the manuscript to change the inline equation from $n^{2/(d+4)}$ to $n^{-2/(d+4)}$ to avoid this confusion.
>
> *> Q7: I don't recognize the dependence on $\sigma_t$ which you mention in the remarks after theorem 3.3. Where do you have something bounded by $\sigma_t^{-4/(d+5)}$?*
>
> Equation (3.12) takes on the form $\mathrm{Risk} \leq \left[ \frac{1}{\sigma_t^2} \cdot \textrm{terms} \right]^{\frac{2}{d+5}} + \textrm{Term2}$. Hence, the dependence on the first term on the RHS is $\sigma_t^{-4/(d+5)}$. We ignore the second term in this discussion, since its dependence on $\sigma_t$ is $\sigma_t^{-3}$.
>
> *> Q8: In line 491 you mention your "truncation arguments", could you say what you mean by this? And maybe give a reference to where you make these arguments?*
>
> When considering diffusion models, the data $x_t$ at time $t$ of the forward process is never almost surely bounded, since $x_t$ is given by adding Gaussian noise to the original data. However, in many places in the analysis, it is helpful and/or necessary to assume that the data is bounded. A standard way of working around this is to define an auxiliary random variable $\bar{x}_t$ defined as $\bar{x}_t := x_t \mathbf{1}\\{ \\| x_t \\| \leq B_t \\}$. If $B_t$ is chosen properly (by using concentration of measure), then we have that $\bar{x}_t = x_t$ with high probability, and we can proceed with our analysis using $\bar{x}_t$ in place of $x_t$. This type of argument is commonly referred to as a *truncation argument*, since we are defining a new random variable ($\bar{x}_t$) that "truncates" the original random variable $x_t$. We hope this clarifies things. For a reference, have at look at the high dimensional statistics book (Wainwright, 2019) cited in our work, which contains plenty of examples of truncation arguments.

---

> ### Author Response · Authors · 2024-11-28
>
> ### Suggestions
>
> *> U1: Please make sure to use the full name and not an abbriviation the first time you mention a concept. For example: DDPM in line 66, ERM in line 74, DNN in line 90 and GD in line 97.*
>
> Thanks for pointing this out, and we have made sure to do this in our manuscript to the full extent possible.
>
> *> U2: Please make sure to explain all the notation you use before you use it.*
>
> We have adding the missing pieces of notation to the Notation paragraph in Section 3.
>
> *> U3: I feel the title over-promises. "SHALLOW DIFFUSION NETWORKS PROVABLY LEARN HIDDEN LOW-DIMENSIONAL STRUCTURE" sounds like it is all shallow networks and all hidden low-dimensional structures. Of course I understand if you feel the title would get too long if you add all the assumptions, so I would suggest finding a new title which is as short or shorter, and which does not make the result sound more general than it is.*
>
> We have modified the title of our paper according to these recommendations.
>
> *> Typo in line 189: Should be $B_2(r, d)$*
>
> Fixed.

---

> > ### Comment · Reviewer_kfcb · 2024-11-28
> > **Confidence increased**
> >
> > Thank you for your answers and the additions to the paper, which I feel let me understand the work better.
> > I have kept my rating as accept, and increased my confidence.

---

### Official Review · Reviewer_USAA · 2024-11-03

**Soundness:** 4
**Presentation:** 3
**Contribution:** 3
**Rating:** 5
**Confidence:** 3

**Summary:**

The paper shows that learning diffusion model in Barron spaces (spanned by single-layer neural networks) avoids curse of dimensionality by adapting to low dimensional subspace.

**Strengths:**

Solid paper with concrete mathematical analysis. The results in the paper could be of general mathematical interests for related fields.

The paper makes a meaningful step towards understanding the gap between curse of dimensionality in theory and no curse in reality.

The paper has nice connection with recent progress in diffusion sampling process.

**Weaknesses:**

I have problems with calling results on "single-layer neural networks" by "results on shallow networks". Neural networks with a few layers are also shallow and the authors don't prove for them here. I'm not going to start a lecture on logic, but a paper should always try avoiding unnecessary confusions, especially those that make people think more favorably than it actually deserves.

The setting of single-layer networks is way too simple. The industry of deep learning scales to using 100k H100s, yet theories still are struggling with analyzing single-layer networks.

The formula in the theorems are complicated. If a simple setting leads to such convoluted formula, what would the more general cases be like?  Usually simpler things are more useful. It would benefit the presentation if a neat bound could be given the main body and a more detailed bound given in the appendix.

Lack of analysis of the gradient descent training process of neural networks.

Lack of direct usable implications. What could practitioners benefit from the theories? Or what future development of the theories could lead to some algorithmic innovations that is beyond imagination of practitioners? Machine learning is rather noisy compared with things like Physics. It's the best if we can get something useful from theoretical understandings.

Lack of experiments. The authors have proposed a set of assumptions which they believe is meaningful and proved results under these assumptions. It would be extremely beneficial to prove that these assumptions are relevant for the real world tasks and experiments on either simulation or real world data support the claim of the theorem. In theories, one inevitably chooses many simplification to make things elegant, which is quite understandable. However, experiments are needed to show that these simplification doesn't make the theoretical results irrelevant.

**Questions:**

Included in weakness.

---

> ### Author Response · Authors · 2024-11-28
>
> *> I have problems with calling results on "single-layer neural networks" by "results on shallow networks". Neural networks with a few layers are also shallow and the authors don't prove for them here. I'm not going to start a lecture on logic, but a paper should always try avoiding unnecessary confusions, especially those that make people think more favorably than it actually deserves.*
>
> The reviewer's point is noted, and we have modified the title of our paper to ``Single hidden layer diffusion models provably learn simple low-dimensional structure``. We hope that this addresses the reviewer's concern, and we thank them for the clarification.
>
> *> The setting of single-layer networks is way too simple. The industry of deep learning scales to using 100k H100s, yet theories still are struggling with analyzing single-layer networks.*
>
> **ML theory and experiment.** We hear the reviewer's concern regarding the relative simplicity of the setups generally considered in the deep learning theory literature, particularly when compared with the massive scale that powers the latest state of the art models. Indeed, the current trend in the machine learning community is that experimental progress occurs at a much faster pace than the corresponding theoretical analysis. Nevertheless, we believe that it is unlikely that we will understand industry-scale neural networks if we do not first understand more simplified settings. For this reason, we believe that characterizing the performance of simple network architectures in modern algorithms such as diffusions is both well-motivated and insightful.
>
> **Scale and complexity.** We would like to point out that compute scale (e.g., making the model larger by adding more parameters) is not necessarily the same thing as architectural complexity. That is, it is possible to take relatively simple architectures and scale them up significantly. Indeed, many state of the art models (e.g., transformers) are built from fairly simple building blocks, but scaled up to parameter counts in the billions. As a result, we believe that understanding these simple building blocks helps elucidate the performance of larger-scale models.
>
> **Idealized settings yield insight.** We believe that it is important to mathematically analyze simplified settings. In particular, these settings can, and often do, reveal insights about practice. As a concrete example, before our work, it was not known if network depth is actually needed to learn simple subspace structure: all prior work required the use of deep networks. Our analysis shows that depth is not necessary, but that a single hidden layer is enough. These kinds of results are difficult to ascertain experimentally, even at industry scale, but are readily obtained through the kind of mathematical analysis we have performed here.
>
> For further details on this point, please also see our response to **Q1**.
>
>
> *> The formula in the theorems are complicated. If a simple setting leads to such convoluted formula, what would the more general cases be like? Usually simpler things are more useful. It would benefit the presentation if a neat bound could be given the main body and a more detailed bound given in the appendix*
>
> Thanks for bringing this up. We have made sure to follow the presentation of each theorem/corollary with a simplified version of the statement, where we keep only the most important factors in the bound. We hope this helps to give more intuition about the results.
>
> *> Lack of analysis of the gradient descent training process of neural networks.*
>
> In this work, our main focus is on the sample complexity of learning a score function, which has been a key missing step in the sampling theory literature for diffusion models. While we agree that analyzing the gradient descent training process is important, we believe it is just as important to first establish that learning is *statistically possible* in the first place. Indeed, this gives researchers a target benchmark to shoot for when analyzing the training procedure. Please also see our answer to **Q2** above.

---

> > ### Comment · Reviewer_USAA · 2024-11-28
> > **Concern addressed.**
> >
> > The authors' responses are well written and addresses my concerns. I fully appreciate the authors' effort of building meaningful theories for deep learning. I would say it's a concrete and solid and meaningful work, but still far from addressing the huge between theory and practice in deep learning. I shall maintain the score but I would recommend accept if AC sees that we should encourage people more towards this direction, despite the huge difficulties.

---

> ### Author Response · Authors · 2024-11-28
>
> *> Lack of direct usable implications. What could practitioners benefit from the theories? Or what future development of the theories could lead to some algorithmic innovations that is beyond imagination of practitioners? Machine learning is rather noisy compared with things like Physics. It's the best if we can get something useful from theoretical understandings.*
>
> Certainly, our theory does not provide any concrete prescrptions such as ``use learning rate X`` or ``use optimization algorithm Y``. However, we believe that such a view of theory is a bit constraining.
>
> Our main focus is to shed light on a fairly remarkable empirical phenomenon: why it is possible to learn to sample with diffusion models? While the latest theoretical results in this area show that the sample complexity of learning to sample is effectively the same as the sample complexity of learning the score function, there are relatively few guarantees on the difficulty of learning the score. Those that do exist mostly require the number of data examples to scale exponentially in the ambient dimension.
>
> Clearly, this is *not* the data regime that modern diffusion models operate in. The key theoretical question then becomes: "Do diffusion models really generalize, or do they just memorize the training data, and thereby provide an illusion of generalization?"
>
> Our work sheds light on this phenomenon by highlighting that a substantially more benign data requirement is sufficient to learn from high-dimensional distributions that contain simple latent structures. The direct implication for practice is that it is indeed possible for diffusion models to generalize for high-dimensional datasets with reasonable sample complexity. Our work gives one of the first theoretical explanations for how and when this occurs, and thereby helps explain the puzzling performance of modern large-scale image generation models.
>
>
> *> Lack of experiments. The authors have proposed a set of assumptions which they believe is meaningful and proved results under these assumptions. It would be extremely beneficial to prove that these assumptions are relevant for the real world tasks and experiments on either simulation or real world data support the claim of the theorem. In theories, one inevitably chooses many simplification to make things elegant, which is quite understandable. However, experiments are needed to show that these simplification doesn't make the theoretical results irrelevant.*
>
> As requested, we have included a new set of experiments in Section 4. Please see our answer above to **Q3** for more details about the set of experiments we include.

---

### Official Review · Reviewer_NgbV · 2024-11-04

**Soundness:** 2
**Presentation:** 2
**Contribution:** 2
**Rating:** 6
**Confidence:** 1

**Summary:**

The paper explores the effectiveness of shallow diffusion networks in learning distributions with low-dimensional structure, challenging the traditional belief that high-dimensional data inherently suffers from the curse of dimensionality. The study provides sample complexity bounds for these models, showing that they depend more on intrinsic latent dimensions rather than the ambient space, thus offering insights into the strengths of diffusion models for structured data generation.

**Strengths:**

The research presents a theoretically grounded approach to understanding the success of shallow diffusion networks, especially for low-dimensional structures in data.

**Weaknesses:**

The reliance on Barron spaces, while insightful, could pose challenges for scalability and model tuning in larger, more complex network architectures, which are crucial to modern success of diffusion-based generative models.

**Questions:**

- Can the analysis be adopted to flow models/matching, which instead regresses a network onto the vector field of probability ODE (rather than the score)?
- Can the authors give comments on how the analysis can be extended to discrete state spaces, for applications such as discrete diffusion models for language modeling?

---

> ### Author Response · Authors · 2024-11-27
>
> ### Weaknesses
> *> The reliance on Barron spaces, while insightful, could pose challenges for scalability and model tuning in larger, more complex network architectures, which are crucial to modern success of diffusion-based generative models.*
>
> Please see our response to **Q3** above.
>
> ### Questions
> *> Can the analysis be adopted to flow models/matching, which instead regresses a network onto the vector field of probability ODE (rather than the score)?*
>
> **ODE sampling.** Our results directly apply to models based on the probability flow ODE, since the vector field of this ODE is a known affine transform of the score function. As a result, we can learn the probability flow ODE with nearly identical sample complexity. The primary difference is that the sampling guarantee will need to change from those that apply for SDE sampling (e.g., (Benton et al., 2024) which we use in our paper) to ODE sampling guarantees (e.g., (Chen et al., 2023c)).
>
> **Interpolants and flow matching.** Our results also apply to learning the drift term of stochastic interpolants/rectified flows (Albergo et al., 2023). We have updated our manuscript to contain a brief discussion in Appendix B about how to modify our proofs to bound the Barron norm of the drift term in stochastic interpolants for the latent structures we consider in the paper, where the resulting bound does not depend exponentially on the ambient dimension.
>
> *> Can the authors give comments on how the analysis can be extended to discrete state spaces, for applications such as discrete diffusion models for language modeling*
>
> This is a very interesting question! Unfortunately, we do not have enough familiarity with discrete diffusion models at this point to make an informed comment. We imagine that there is low-dimensional structure to be taken advantage of in the language modeling setting, but we leave this to future work.

---

### Official Review · Reviewer_iFA5 · 2024-11-04

**Soundness:** 3
**Presentation:** 4
**Contribution:** 3
**Rating:** 6
**Confidence:** 3

**Summary:**

This paper investigates why diffusion models, specifically those utilizing shallow neural networks within Barron spaces, can effectively learn and sample from high-dimensional data distributions that possess hidden low-dimensional structures. By focusing on Barron spaces—the function space of single-layer neural networks—the authors demonstrate that diffusion models can adapt to simple forms of low-dimensional structure without the need for specialized architectures. They provide theoretical results showing that the sample complexity for learning these models depends polynomially on the intrinsic latent dimensionality rather than exponentially on the ambient dimension. The analysis includes end-to-end sample complexity bounds for learning to sample from structured distributions, highlighting how shallow diffusion networks can circumvent the classical curse of dimensionality by leveraging the low-index structure of Barron classes.

**Strengths:**

- The paper provides a rigorous theoretical framework that explains how diffusion models can overcome the curse of dimensionality by adapting to low-dimensional latent structures within high-dimensional data.
- By leveraging Barron spaces to model shallow neural networks, the authors bridge the gap between theoretical tractability and practical relevance, as these spaces capture essential features of networks used in practice.
- The work offers comprehensive sample complexity bounds that depend on the intrinsic latent dimensionality, providing valuable insights into the efficiency of learning diffusion models for structured distributions.
- The results do not rely on specialized network architectures tailored to specific latent structures, emphasizing the general applicability of shallow diffusion networks in learning hidden low-dimensional structures.

**Weaknesses:**

- The analysis is restricted to shallow (single-layer) neural networks, which may not capture the complexities and representational power of deep neural networks commonly employed in state-of-the-art diffusion models.
- The theoretical results are derived under idealized conditions, such as target distributions concentrated on low-dimensional linear manifolds or composed of independent components, which may not fully reflect real-world data complexities.
- Optimizing over Barron spaces can be computationally challenging due to the infinite-dimensional nature of these function spaces, raising questions about the practicality of implementing the proposed methods.
- Minor typo: Line 315 (difference -> different?)

**Questions:**

- To what extent can the assumptions of Lipschitz continuity and sub-Gaussianity be relaxed? Are there alternative conditions under which similar results could be obtained, possibly broadening the applicability of the theory?
- Can the authors elaborate on the practical aspects of optimizing over Barron spaces?
- In practical scenarios, data often contain noise and may deviate from idealized models. How robust is the proposed approach to such imperfections, and what modifications, if any, are needed to handle real-world datasets?
- Given that non-linear manifolds are noted as future work, could you share any preliminary insights on how extending to non-linear latent structures might impact sample complexity or theoretical guarantees?"

---

> ### Author Response · Authors · 2024-11-27
>
> ### Weaknesses
>
> *> The analysis is restricted to shallow (single-layer) neural networks, which may not capture the complexities and representational power of deep neural networks commonly employed in state-of-the-art diffusion models.*
>
> Please see our response to **Q1** above.
>
>
> *> The theoretical results are derived under idealized conditions, such as target distributions concentrated on low-dimensional linear manifolds or composed of independent components, which may not fully reflect real-world data complexities.*
>
> Please see our response to **Q3** above.
>
> *> Optimizing over Barron spaces can be computationally challenging due to the infinite-dimensional nature of these function spaces, raising questions about the practicality of implementing the proposed methods.*
>
> Please see our response to **Q2** above.
>
> ### Questions
>
> *>To what extent can the assumptions of Lipschitz continuity and sub-Gaussianity be relaxed? Are there alternative conditions under which similar results could be obtained, possibly broadening the applicability of the theory?*
>
> **Tail behavior.** The assumption that the data is sub-Gaussian can easily be relaxed, say to heavier tailed data (e.g., sub-Weibull) distributions. We choose to use sub-Gaussanity as the main assumption to keep the proofs as simple as possible, and to not distract from the main innovations. We also note that the assumption of sub-Gaussian data is fairly standard in the literature (see e.g., (Chen et al., 2023b)).
>
> **Lipschitz.** Our assumption on Lipschitz score function can also be relaxed to Holder continuous functions. The approximations results of (Bach, 2017) that we rely upon have already been generalized to Sobolev functions in (Jacot et al., 2024), and extending them to Holder functions should not lead to any particular problem.
>
>
> *> Can the authors elaborate on the practical aspects of optimizing over Barron spaces?*
>
> Please see our response to **Q2** above. In practice, we use a finite-width approximation of Barron, which we then optimize with gradient descent. Experimentally, we find that this works very well, as we show in our new numerical results. Understanding the optimization dynamics for diffusion models is a fascinating problem, but we feel that it is out of scope for this work, which focuses on the statistical properties of learning. We therefore leave their analysis to future work.
>
>
> *> In practical scenarios, data often contain noise and may deviate from idealized models. How robust is the proposed approach to such imperfections, and what modifications, if any, are needed to handle real-world datasets?*
>
> Please see our response to **Q3** above.
>
>
> *> Given that non-linear manifolds are noted as future work, could you share any preliminary insights on how extending to non-linear latent structures might impact sample complexity or theoretical guarantees?"*
>
> As noted in our response to **Q3** above, analyzing general non-linear manifolds will most likely require us to go beyond the function class $\mathcal{F}\_1$. The intuition is that one wants to represent the score as the composition of three functions: a projection to the manifold, the on-manifold score, and then an embedding back into the full space. $\mathcal F\_1$ only adapts to compositionality if at most one of these three functions is linear $\\|U\circ f\circ V\\|\_{\mathcal F\_1}=\\|f\\|\_{\mathcal F\_1}$ (for orthonormal $U,V$). This is compatible with linear low-dimensional structure because the projection and embedding are linear, and because only the low-dimensional score is nonlinear. However, with nonlinear structures, the projection and embedding are both nonlinear functions, so adaptivity to compositions of multiple nonlinear functions is required.

---

> ### Comment · Reviewer_iFA5 · 2024-11-29
>
> Most of my questions were solved and I greatly appreciate their detailed explanation of why the simplicity of their network is a strength. Additionally, I acknowledge the value of the extra experiments presented in Section 4—while I don't find them strictly necessary for this line of research, they might be helpful for some readers. Based on this, I am inclined to raise my score to 7. However, since the system does not allow selecting a score of 7, I will leave it as 6 and notify the AC that my intended score is 7.

---

### Author Response · Authors · 2024-11-27
**General Response**

We thank the reviewers for their helpful comments regarding our work. We have incorporated the comments into our revised manuscript, which has significantly improved the clarity of our presentation. Furthermore, we have also added a new experimental section (Section 4 in the revised manuscript) to validate our theory as requested. Here, we first respond to a few questions that were shared by several reviewers.

***Q1:*** *Given that Barron networks are simplified compared with the architectures used in SoTA diffusion models, what is the relevance for studying them?*

We understand that this is a point of concern for the reviewers. While we agree that Barron networks are far from the networks used at scale in industry, we believe that core insights can be gained by studying their properties. In particular, earlier theoretical work on diffusion models (Chen et al., 2023b), (Oko et al., 2023) required more complex, deeper networks to learn the score of a data distribution supported on a low-dimensional subspace. Our results elucidate the role of depth in generative modeling by highlighting that more than one hidden layer is not necessary for this kind of structure. For this reason, we view our focus on simple networks as a **strength** of our work, as it precisely clarifies the required number of hidden layers for a neural network to identify linear structure in a generative modeling problem. In addition, despite their more complex architectures, we remark that these earlier papers do not apply to case of independent components that we study in Section 3.2, as the latent structure we exploit there is *probablistic*, not geometric (e.g., the data we consider here does *not* live on a low-dimensional manifold).
Hence, our results are more general despite the simpler network architecture.

More generally, the role of depth in deep learning is a fundamental and challenging question that has been of central interest in recent years. While several results exist that demonstrate the role of (a few layers of) depth in standard regression and approximation-theoretic contexts (Eldan and Shamir, 2016; Telgarsky, 2016), few works have tackled the generative modeling setting. Moreover, even these state of the art theoretical works study simplified fully-connected network architectures, and are far from understanding production-ready networks that are engineered in industry.

Here, we take a first step towards understanding the role of depth in generative modeling by clarifying the simplest setting, which compares kernel methods to a single layer of nonlinearity. While we hope to tackle more complex and realistic architectures in the future, such architectures are poorly understood even in basic problem settings such as supervised classification/regression. It is our belief that they should first be understood in these basic settings before considering more complex generative modeling setting; therefore, these questions are out of scope for this work.


***Q2:*** *What about the computational challenges associated with optimizing Barron networks? What does our work have to say about these issues?*

There are two primary issues that arise when implementing Barron networks in practice.

The first issue originates from the fact that Barron networks arise in the infinite-width limit, and hence cannot be implemented exactly on a computer. However, it is shown in (Bach, 2017, Section 2) that any $f \in \\mathcal{F}\_1$ can be approximated to $\varepsilon$ accuracy in $L_2$ by a finite-width network using only $O(\\| f\\|_{\\mathcal{F}_1}^2/\\varepsilon^2)$ neurons. In this case, because the $\\mathcal{F}_1$ norms of the target scores we consider scale polynomially in the ambient dimension, we can easily extend our results to finite-width networks. In particular, the required width does not require a number of neurons that is exponential in the ambient dimension. We have updated the manuscript to make this point clear.

The second issue, which is more challenging to handle, is that it is currently unknown if gradient-based optimization algorithms converge to minimum Barron norm solutions in the finite-width setting. In particular, (Bach, 2017) shows that optimization over Barron networks can be NP-hard in general. Similar to our answer to the previous question, showing that gradient-based optimization works outside of the NTK regime is an open question in all of deep learning theory, including in simpler supervised classification and regression settings. For this reason, resolving this issue is out of scope for this work. On the other hand, the complexity of analyzing gradient descent further illustrates the importance of deriving sample complexity results for single hidden layer networks. For such networks, the gradient dynamics are substantially simpler to analyze compared with deeper networks.

---

> ### Author Response · Authors · 2024-11-27
> **Continuation of General Response**
>
> ***Q3:*** *How robust are the results to perturbations of the hidden latent structures, and does our work extend to more complex latent structure such as general manifolds?*
>
> We expect our results to be robust to perturbations of the hidden latent structure. For example, suppose that we perturb data living on a low-dimensional subspace with Gaussian noise. This is mathematically equivalent to running a diffusion process for a fixed amount of time and taking the output as the new data distribution, and our results therefore easily extend to this type of perturbation. We expect our results to be robust to other classes of noise perturbations with sufficient tail decay.
>
> On the other hand, we expect there to be some limitations on the kind of hidden structure that Barron networks can efficiently capture. In particular, we expect that some amount of depth will be necessary to learn general nonlinear manifolds in a sample efficient manner; this goes beyond what $\\mathcal{F}\_1$ networks can model. Intuition for this claim is that the score can be viewed as a composition of three functions: a manifold projection, an on-manifold score, and then an embedding back into the full ambient space. $\\mathcal F\_1$ only adapts to compositionality if at most one of these three functions is linear, $\\|U\circ f \circ V\\|\_{\mathcal F\_1}=\\|f\\|\_{\mathcal F\_1}$ for orthogonal $U$ and $V$. In the linear structure case that we consider, the projection and embedding are both linear, and only the low-dimensional score is nonlinear. With nonlinear manifold data, both the projection and embedding become nonlinear functions. We leave exploring the role of depth for these more complex structures to future work.
>
> We would also like to bring attention to Section 4 in revised manuscript, which presents new empirical results as requested by Reviewer USAA, comparing both finite-width Barron $\\mathcal{F}_1$ networks optimized with gradient descent, to both two hidden layer layer MLPs and $\\mathcal{F}_2$ networks, which are single hidden layer networks where the first layer is held fixed. In particular, we would like to highlight that (a) Barron networks exhibit very similar performance to two hidden layer MLPs, and and (b) there is a non-trivial gap between $\\mathcal{F}_2$ networks and the $\\mathcal{F}_1$ networks that can be seen even on data of moderate dimensions.

---

### Meta-Review · Area_Chair_FHXz · 2024-12-21

**Metareview:**

The paper is a solid mathematical analysis of an interesting phenomenon and will be of interest to much of the community. Some complexities, such as very deep models and algorithmic aspects, have been simplified for the sake of theoretical results, but even without them, the contribution is well recognized. In addition, clarifications of the presentation made in the rebuttal should be appropriately reflected in order to appeal to a wider audience of the machine learning community.

**Additional Comments On Reviewer Discussion:**

iFA5 commended the mathematical results of this paper. He also made some algorithmic challenges and problems with the shallow neural network. USAA pointed out similar setups. ngbV pointed out the extensibility to other problems. kfcb pointed out the readability of the theoretical part of the paper. iFA5 noted the paper's mathematical presentation. With the exception of a few points, the overall rating is positive.

---

### Decision · Program_Chairs · 2025-01-22

Accept (Poster)